# Faster Accelerated First-order Methods for Convex Optimization with Strongly Convex Function Constraints

**Zhenwei Lin**
Shanghai University of Finance and Economics
zhenweilin@163.sufe.edu.cn

**Qi Deng**[*]
Antai College of Economics and Management
Shanghai Jiao Tong University
qdeng24@sjtu.edu.cn

## Abstract

In this paper, we introduce faster accelerated primal-dual algorithms for minimizing a convex function subject to strongly convex function constraints. Prior to our work, the best complexity bound was $\mathcal{O}(1/\varepsilon)$, regardless of the strong convexity of the constraint function. It is unclear whether the strong convexity assumption can enable even better convergence results. To address this issue, we have developed novel techniques to progressively estimate the strong convexity of the Lagrangian function. Our approach, for the first time, effectively leverages the constraint strong convexity, obtaining an improved complexity of $\mathcal{O}(1/\sqrt{\varepsilon})$. This rate matches the complexity lower bound for strongly-convex-concave saddle point optimization and is therefore order-optimal. We show the superior performance of our methods in sparsity-inducing constrained optimization, notably Google's personalized PageRank problem. Furthermore, we show that a restarted version of the proposed methods can effectively identify the optimal solution's sparsity pattern within a finite number of steps, a result that appears to have independent significance.

## 1 Introduction

In this paper, we are interested in the following convex function-constrained problem:

$$\min_{\mathbf{x}\in\mathbb{R}^n} \quad f(\mathbf{x}) \quad \text{s.t.} \quad g_i(\mathbf{x}) \leq 0, \ 1 \leq i \leq m, \tag{1}$$

where $f : \mathbb{R}^n \to \mathbb{R}$ is a convex continuous function and bounded from below and $g_i : \mathbb{R}^n \to \mathbb{R}$, $i = 1, 2, \ldots, m$, are strongly convex continuous functions. An important application of this problem, commonly encountered in statistics and engineering, involves the objective $f(\mathbf{x})$ as a proximal-friendly regularizer and $g_i(\mathbf{x})$ as a data-driven loss function used to gauge model fidelity.

To apply first-order methods for the above function-constrained problems, a common strategy involves a double-loop procedure that repeatedly employs fast first-order methods, such as Nesterov's accelerated method, to solve specific strongly convex proximal subproblems. Popular methods among this category include Augmented Lagrangian methods [18, 33], level-set methods [21], penalty methods [17]. When both $f(\mathbf{x})$ and $g_i(\mathbf{x})$ are convex and smooth (or composite), it has been found that these double-loop algorithms can attain an iteration complexity of $\mathcal{O}(1/\varepsilon)$ to achieve an $\varepsilon$-error

---

[*]Corresponding author

38th Conference on Neural Information Processing Systems (NeurIPS 2024).

in both the optimality gap and constraint violation. When the objective is strongly convex, the complexity can be further improved to $\mathcal{O}(1/\sqrt{\varepsilon})$ ([33, 21]).

In contrast to these double-loop algorithms, single-loop algorithms remain popular due to their simplicity in implementation. Along this research line, [32] developed a first-order algorithm based on linearizing the augmented Lagrangian function, which obtains an iteration complexity of $\mathcal{O}(1/\varepsilon)$. [34] extended the augmented Lagrangian method to stochastic function-constrained problems where both objective and constraint exhibit an expectation form. Viewing (1) as a special case of the min-max problem:

$$\min_{\mathbf{x}\in\mathbb{R}^n} \max_{\mathbf{y}\in\mathbb{R}^m} \ \mathcal{L}(\mathbf{x}, \mathbf{y}) := f(\mathbf{x}) + \sum_{i=1}^m y_i g_i(\mathbf{x}), \quad \text{s.t. } y_i \geq 0, \ i = 1, 2, \ldots, m, \quad (2)$$

[11] proposed to solve (1) and (2) by an accelerated primal-dual method (APD), which generalizes the primal-dual hybrid gradient method [6] initially developed for saddle point optimization with bilinear coupling term. Under mild conditions, APD achieves the best iteration complexity of $\mathcal{O}(1/\varepsilon)$ for general convex constrained problem and a further improved complexity of $\mathcal{O}(1/\sqrt{\varepsilon})$ when $f(\mathbf{x})$ is strongly convex. [4] proposed a unified constrained extrapolation method that can be applied to both deterministic and stochastic constrained optimization problems.

Despite these recent progresses, to the best of our knowledge, all available algorithms are suboptimal in the presence of strongly convex function constraints (1). Specifically, direct applications of previously discussed algorithms yield an $\mathcal{O}(1/\varepsilon)$ complexity, which is inferior to the $\mathcal{O}(1/\sqrt{\varepsilon})$ optimal bound for the strongly-convex-concave saddle point problem [22]. It is somewhat unsatisfactory that the strong convexity of $g(\mathbf{x})$ has not been found helpful in further algorithmic acceleration. The core underlying issue arises from the dynamics of saddle point optimization: it is the strong convexity of $\mathcal{L}(\cdot, \mathbf{y})$ that offers more potential acceleration advantages, yet the strong convexity of $\mathcal{L}(\cdot, \mathbf{y})$ is substantially harder to estimate than that of $g(\mathbf{x})$. This difficulty is compounded by the interplay between $g(\mathbf{x})$ and the varying dual sequence $\{\mathbf{y}_k\}$. The challenge naturally leads us to question: *Is it possible to further improve the convergence rate of first-order methods for solving the strongly convex constrained problem* (1)*?*

**Key intuitions**   We make an assumption that the minimizer of $f(\mathbf{x})$ is infeasible for the function constraint $g_i(\mathbf{x}) \leq 0$, $1 \leq i \leq m$. If this assumption were not made, we would be dealing with an unconstrained optimization problem that would not depend on $g(\mathbf{x})$. This assumption also implies that the optimal dual variables are non-zero, and as a result, the Lagrangian function is strongly convex with respect to $\mathbf{x}$. By leveraging the strong convexity, we can use more aggressive step sizes and achieve faster convergence rates compared to other state-of-the-art algorithms.

**Applications in sparsity-constrained optimization**   We consider the constrained Lasso-type problem, which minimizes a sparsity-inducing regularizer while explicitly ensuring data-driven error remains controlled:

$$\min_{\mathbf{x}\in\mathbb{R}^n} \ \|\mathbf{x}\|_1 \quad \text{s.t. } g(\mathbf{x}) \leq 0, \quad (3)$$

where $g(\cdot)$ is a convex smooth loss term. A motivating application is the approximate personalized PageRank problem [8], where $g(\mathbf{x}) = \frac{1}{2}\langle \mathbf{x}, Q\mathbf{x} \rangle - \langle \mathbf{b}, \mathbf{x} \rangle$ is strongly convex quadratic and $Q$ integrates the graph Laplacian with an identity matrix. Compared to the standard Lasso problem [30], $\min_{\mathbf{x}\in\mathbb{R}^n} g(\mathbf{x}) + \lambda\|\mathbf{x}\|_1$, the constrained problem (3) offers enhanced control over the data fitting error. This advantage, however, is counterbalanced by the challenge of dealing with a nonlinear constraint. Besides concerns about the efficiency in solving (3), it is often desired to show the active set (or sparsity) identification, namely, the nonzero patterns of the optimal solution $\mathbf{x}^*$ can be identified by the solution sequence $\{\mathbf{x}_k\}$ in a finite number of iterations. Identifying the embedded solution structure within a broader context is referred to as the manifold identification problem [31, 12]. Exploiting the sparsity pattern is particularly desirable in large-scale PageRank problems, as it could result in significant runtime savings. For the regularized Lasso-type problem, it has been known that proximal gradient methods (e.g. [14, 19, 24]) possess the finite active-set identification property. Specifically, [24] introduced "active set complexity", which is defined as the number of iterations required before an algorithm is guaranteed to have reached the optimal manifold, and they proved the proximal gradient method with constant stepsize can identify the optimal manifold in a finite number of iterations. However, for the problem (3), it remains unclear whether first-order methods can identify the sparsity pattern in finite time.

**Contributions**  We address the theoretical questions about strongly convex constrained optimization and the application of sparse optimization. Our contributions are summarized as follows.

First, we present a new accelerated primal-dual algorithm with progressive strong convexity estimation (APDPro) for solving problem (1). APDPro employs a novel strategy to estimate the lower bound of the dual variables, which leads to a gradually refined estimated strong convexity modulus of $\mathcal{L}(\cdot, \mathbf{y})$. With additional cut constraints on the dual update, APDPro is able to separate the dual search space from the origin point, which is critical for maintaining the desired strong convexity over the entire solution path. With these two important ingredients, APDPro exhibits an $\mathcal{O}\big((\|\mathbf{x}_0 - \mathbf{x}^*\| + D_Y)/\sqrt{\varepsilon}\big)$ complexity bound to obtain an $\varepsilon$-error on the function value gap and constraint violation, where $D_Y$ is a known upper-bound of $\|\mathbf{y}_0 - \mathbf{y}^*\|$. Moreover, we show that for the last iterate to have an $\varepsilon$ error (i.e., $\|\mathbf{x}_K - \mathbf{x}^*\|^2 \le \varepsilon$), APDPro requires a total iteration of $\mathcal{O}\big((\|\mathbf{x}_0 - \mathbf{x}^*\| + \|\mathbf{y}_0 - \mathbf{y}^*\|)/\sqrt{\varepsilon}\big)$. Both complexity results appear new in the literature for strongly convex-constrained optimization.

Second, we present a new restart algorithm (rAPDPro) which calls APDPro repeatedly with the input parameters properly changing over time. Different from APDPro, rAPDPro dynamically adjusts the iteration number of APDPro in each epoch based on the progressive strong convexity estimation. We show that rAPDPro exhibits a complexity of $\mathcal{O}\big(\log(D_X/\sqrt{\varepsilon}) + D_Y/\sqrt{\varepsilon}\big)$ to ensure $\varepsilon$-error in the last iterate convergence where $D_X$ is the estimated diameter of the primal feasible domain. While it is difficult to improve the overall $\mathcal{O}(1/\sqrt{\varepsilon})$ bound, rAPDPro appears to be more advantageous when $D_X$ and $D_Y$ are the same order of $\|\mathbf{x}_0 - \mathbf{x}^*\|$ and $\|\mathbf{y}_0 - \mathbf{y}^*\|$, respectively, and $D_X \gg D_Y$. In addition, we show that a similar restart strategy can further accelerate the standard APD. The multistage-accelerated primal dual method (msAPD) obtains a comparable $\mathcal{O}(1/\sqrt{\varepsilon})$ complexity of APDPro without introducing additional cut constraint.

Third, we apply our proposed methods to the sparse learning problem (3). In view of the theoretical analysis, all our methods converge at an $\mathcal{O}(1/\sqrt{\varepsilon})$ rate, which is substantially better than the rates of state-of-the-art first-order algorithms. Moreover, we conduct a new analysis to show that the restart algorithm rAPDPro has the favorable feature of identifying the optimal sparsity pattern. Note that such active-set/manifold identification is substantially more challenging to prove due to the coupling of dual variables and constraint functions. To establish the desired property, we develop asymptotic convergence of the dual sequence to the optimal solution, which can be of independent interest.

**Outline**  Section 2 sets notations and assumptions for the later analysis. Section 3 presents the APDPro algorithm and develops its stepsize rule and complexity rate. Section 4 presents the restart APDPro (rAPDPro) algorithm. Section 5 applies our proposed methods for sparsity-inducing optimization and shows the sparsity identification result for rAPDPro. Section 6 empirically examine the convergence performance and sparsity identification of our proposed algorithms. Finally, we draw the conclusion in Section 7. All the missing proofs are provided in the appendix sections.

## 2   Preliminaries

We use bold letters like $\mathbf{x}$ to represent vectors. Suppose $\mathbf{x} \in \mathbb{R}^n$, $q \ge 1$, we use $\|\mathbf{x}\|_q = \big(\sum_{i=1}^n |\mathbf{x}_{(i)}|^q\big)^{1/q}$ to represent the $l_q$-norm, where $\mathbf{x}_{(i)}$ is the $i$-th element of $\mathbf{x}$. For brevity, $\|\mathbf{x}\|$ stands for $l_2$-norm. For a matrix $A$, we denote the matrix norm induced by 2-norm as $\|A\| = \sup_{\|\mathbf{x}\| \le 1} \|A\mathbf{x}\|$. The normal cone of $\mathcal{U}$ at $\mathbf{u}$ is denoted as $\mathcal{N}_{\mathcal{U}}(\mathbf{u}) := \{\mathbf{v} \mid \langle \mathbf{v}, \mathbf{x} - \mathbf{u} \rangle \le 0, \forall \mathbf{x} \in \mathcal{U}\}$. Let $\mathcal{B}(\mathbf{x}, r)$ be the closed ball centered at $\mathbf{x}$ with radius $r > 0$, i.e., $\mathcal{B}(\mathbf{x}, r) = \{\mathbf{y} \mid \|\mathbf{y} - \mathbf{x}\| \le r\}$. We denote the set of feasible solutions by $\mathcal{X}_G := \{\mathbf{x} \mid g_i(\mathbf{x}) \le 0, \forall i \in [m]\}$ and write the constraint function as $G(\mathbf{x}) := [g_1(\mathbf{x}), \ldots, g_m(\mathbf{x})]^\top$. We assume each $g_i(\mathbf{x})$ is a $\mu_i$ strongly convex function, and denote $\boldsymbol{\mu} := [\mu_1, \ldots, \mu_m]^\top$. Let $[m] := \{1, \ldots, m\}$ for integer $m$. We denote minimum and maximum strongly convexity $\underline{\mu} := \min_{j \in [m]}\{\mu_j\}$, and $\bar{\mu} := \max_{j \in [m]}\{\mu_j\}$ and the vector of elements 0 by $\mathbf{0}$. The Lagrangian function of problem (1) is given by $\mathcal{L}(\mathbf{x}, \mathbf{y}) := f(\mathbf{x}) + \langle \mathbf{y}, G(\mathbf{x}) \rangle$ where $\mathbf{y} \in \mathbb{R}_+^m$.

**Definition 1** (KKT condition). *We say that $\mathbf{x}^*$ satisfies the KKT condition of (1) if there exists a Lagrangian multiplier vector $\mathbf{y}^* \in \mathbb{R}_+^m$ such that $\mathbf{0} \in \partial_x \mathcal{L}(\mathbf{x}^*, \mathbf{y}^*)$ and $\langle \mathbf{y}^*, G(\mathbf{x}^*) \rangle = 0$.*

The KKT condition is necessary for optimality when a constraint qualification (CQ) holds at $\mathbf{x}^*$. We assume Slater's CQ (Assumption 1) holds, which guarantees that an optimal solution is also a KKT point [3].

**Assumption 1.** *There exists a strictly feasible point $\widetilde{\mathbf{x}} \in \mathbb{R}^n$ such that $G(\widetilde{\mathbf{x}}) < \mathbf{0}$.*

We use $\tilde{\mathbf{x}}$ to denote a strictly feasible point throughout the paper. Moreover, we require Assumption 2 to circumvent any trivial solution.

**Assumption 2.** *For any $\mathbf{x}_0^* \in \operatorname{argmin}_{\mathbf{x} \in \mathbb{R}^n} f(\mathbf{x})$, there exists an $i \in [m]$ such that $g_i(\mathbf{x}_0^*) > 0$.*

**Remark 1.** *Assumption 2 is essential for our analysis. While verifying Assumption 2 can be indeed challenging, it is achievable for the sparsity-inducing problem considered in our paper. In this example, the solution $\mathbf{x}_0^* = \mathbf{0}$ is the single minimizer of the sparsity penalty.*

Next, we give several useful properties about the optimal solutions of problem (1). Please refer to Appendix D.1 for the proof of Proposition 1 and Appendix D.2 for the proof of Proposition 2.

**Proposition 1.** *Suppose Assumption 1 holds. Then, for any optimal solution $\mathbf{x}^*$ of problem (1), there exists $\mathbf{y}^* \in \mathbb{R}^m$ such that KKT condition holds. Moreover, $\mathbf{y}^*$ falls into set $\mathcal{Y} := \{\mathbf{y} \mid \|\mathbf{y}\|_1 \leq \bar{c}\}$, where $\bar{c} := \frac{f(\tilde{\mathbf{x}}) - \min_{\mathbf{x} \in \mathbb{R}^n} f(\mathbf{x})}{\min_{i \in [m]}\{-g_i(\tilde{\mathbf{x}})\}}$.*

**Proposition 2.** *Under Assumption 2, $\mathbf{x}^*$ is the unique solution of (1). Furthermore, set $\mathcal{Y}^* = \operatorname{argmax}_{\mathbf{y} \in \mathbb{R}_+^m} \mathcal{L}(\mathbf{x}^*, \mathbf{y})$ is convex and bounded.*

In view of Assumption 2, Proposition 2, and closedness of the subdifferential set of proper convex functions [2, Theorem 3.9], [27, Chapter 23], we know that $\mathbf{dist}(\partial f(\mathbf{x}^*), \mathbf{0}) > 0$, where $\mathbf{dist}(\partial f(\mathbf{x}^*), \mathbf{0}) := \min_{\xi \in \partial f(\mathbf{x}^*)} \|\xi\|$. Furthermore, we make the following assumption:

**Assumption 3.** *Throughout the paper, suppose that a constant $r$ satisfying*

$$\mathbf{dist}(\partial f(\mathbf{x}^*), \mathbf{0}) \geq r > 0, \tag{4}$$

*is known.*

We give some important examples for which the lower bound $r$ can be estimated. Suppose $f(\mathbf{x})$ is a Lasso regularizer, i.e., $f(\mathbf{x}) = \|\mathbf{x}\|_1$, then $r = 1$ satisfies (4). More general, consider the group Lasso regularizer, i.e., $f(\mathbf{x}) = \sum_{i=1}^B p_i \|\mathbf{x}_{(i)}\|$, where $\mathbf{x}_{(i)} \in \mathbb{R}^{b_i}$ and $\sum_{i=1}^B b_i = n$, $B$ is the number of blocks, then $r = \min_{i \in [B]}\{p_i\}$ when $\mathbf{x}^* \neq \mathbf{0}$. Another example is $f(\mathbf{x}) = \mathbf{c}^\top \mathbf{x}$, then we have $r = \|\mathbf{c}\|$.

**Remark 2.** *Condition (4) is similar to the bounded gradient assumption that has been used for accelerating the convergence of the Frank-Wolfe algorithm. See Appendix B for more discussions.*

When considering the Lipschitz continuity of function in $\mathbb{R}^n$, even quadratic functions are not Lipschitz continuous. However, the Lipschitz continuity of $g_i(x)$ is crucial for algorithm convergence. Therefore, we define the bounded feasible region in the following proposition, with its proof provided in Appendix D.3.

**Proposition 3.** *Let $\mathcal{X} := \mathcal{B}\big(\tilde{\mathbf{x}}, \min_{i \in [m]} 2\sqrt{\frac{-2g_i(\mathbf{x}_i^*)}{\mu_i}}\big)$, where $\mathbf{x}_i^* = \operatorname{argmin}_{\mathbf{x} \in \mathbb{R}^n} g_i(\mathbf{x})$. Then under Assumptions 1 and 2, we have $\mathbf{x}^* \in \mathbf{int}\,\mathcal{X}$.*

**Assumption 4.** *There exist $L_X, L_G > 0$ such that*

$$\|\nabla G(\mathbf{x}) - \nabla G(\bar{\mathbf{x}})\| \leq L_X \|\mathbf{x} - \bar{\mathbf{x}}\|, \quad \forall \mathbf{x}, \bar{\mathbf{x}} \in \mathcal{X}, \tag{5}$$

$$\|G(\mathbf{x}) - G(\bar{\mathbf{x}})\| \leq L_G \|\mathbf{x} - \bar{\mathbf{x}}\|, \quad \forall \mathbf{x}, \bar{\mathbf{x}} \in \mathcal{X}, \tag{6}$$

*where $\nabla G(\mathbf{x}) := [\nabla g_1(\mathbf{x}), \cdots, \nabla g_m(\mathbf{x})] \in \mathbb{R}^{n \times m}$ and $\mathcal{X}$ is defined in Proposition 3.*

The Lipschitz smoothness of the Lagrangian function with respect to the primal variable $\mathbf{x}$ is crucial for the convergence of algorithms. Given that the dual variable $\mathbf{y}$ is bounded from above, and considering the smoothness of the constraint functions, we can derive the smoothness of the Lagrangian function. Combining (5) and the fact $\|\mathbf{y}\| \leq \|\mathbf{y}\|_1 \leq \bar{c}, \forall \mathbf{y} \in \mathcal{Y}$, we obtain that

$$\|\nabla G(\mathbf{x})\mathbf{y} - \nabla G(\bar{\mathbf{x}})\mathbf{y}\| \leq L_{XY}\|\mathbf{x} - \bar{\mathbf{x}}\| \quad \forall \mathbf{x}, \bar{\mathbf{x}} \in \mathcal{X}, \ \forall \mathbf{y} \in \mathcal{Y}, \tag{7}$$

where $L_{XY} = \bar{c}L_X$. For set $\mathcal{X}$, $\mathcal{Y}$, we use $D_X$ and $D_Y$ to denote their diameters, respectively, i.e., $D_X := \max_{\mathbf{x}_1, \mathbf{x}_2 \in \mathcal{X}} \|\mathbf{x}_1 - \mathbf{x}_2\|$ and $D_Y := \max_{\mathbf{y}_1, \mathbf{y}_2 \in \mathcal{Y}} \|\mathbf{y}_1 - \mathbf{y}_2\|$.

**Algorithm 1** Accelerated Primal-Dual Algorithm with Progressive Strong Convexity Estimation (APDPro)

---

**Require:** $\tau_0 > 0, \sigma_0 > 0, \mathbf{x}_0 \in \mathcal{X}, \mathbf{y}_0 \in \mathcal{Y}, \rho_0 \geq 0, N > 0$

1: **Initialize:** $(\mathbf{x}_{-1}, \mathbf{y}_{-1}) \leftarrow (\mathbf{x}_0, \mathbf{y}_0), \bar{\mathbf{x}}_0 \leftarrow \mathbf{x}_0, \sigma_{-1} \leftarrow \sigma_0, T_0 = 0$
2: Set $\Delta_{XY} = \frac{1}{2\tau_0} D_X^2 + \frac{1}{2\sigma_0} D_Y^2$
3: **for** $k = 0, 1, \ldots, N$ **do**
4: $\quad \mathcal{Y}_k \leftarrow \left\{ \mathbf{y} \in \mathbb{R}_+^m \mid \|\mathbf{y}\|_1 \cdot \underline{\mu} \geq \rho_k \right\} \bigcap \mathcal{Y},$
5: $\quad \mathbf{z}_k \leftarrow (1 + \sigma_{k-1}/\sigma_k)G(\mathbf{x}_k) - (\sigma_{k-1}/\sigma_k)G(\mathbf{x}_{k-1})$
6: $\quad \mathbf{y}_{k+1} \leftarrow \operatorname{argmin}_{\mathbf{y} \in \mathcal{Y}_k} \|\mathbf{y} - (\mathbf{y}_k + \sigma_k \mathbf{z}_k)\|^2$
7: $\quad \mathbf{x}_{k+1} \leftarrow \operatorname{prox}_{f,\mathcal{X}}(\mathbf{x}_k - \tau_k \nabla G(\mathbf{x}_k)\mathbf{y}_{k+1}, \tau_k)$
8: $\quad$ Compute $t_k, \quad \bar{\mathbf{x}}_{k+1} \leftarrow (T_k \bar{\mathbf{x}}_k + t_k \mathbf{x}_{k+1})/(T_k + t_k), \quad T_{k+1} \leftarrow T_k + t_k$
9: $\quad$ Update $\rho_{k+1} \leftarrow \text{IMPROVE}(\mathbf{x}_k, \bar{\mathbf{x}}_k, \frac{\sigma_0 \tau_{k-1} \Delta_{XY}}{\sigma_{k-1}}, \frac{\Delta_{XY}}{T_k}, \rho_k)$
10: $\quad$ Update $\tau_{k+1}$ and $\sigma_{k+1}$ depending on $\rho_{k+1}$
11: **end for**
12: **Output:** $\mathbf{x}_{N+1}, \mathbf{y}_{N+1}$
13: **procedure** IMPROVE($\mathbf{x}, \bar{\mathbf{x}}, \beta, \bar{\beta}, \rho_{\text{old}}$)
14: $\quad$ Compute $\rho = \underline{\mu} \cdot \max \left\{ r \left[ \|\nabla G(\mathbf{x})\| + L_X \sqrt{2\beta} \right]^{-1}, \left[ \frac{L_X}{r} \sqrt{\frac{\bar{\beta}}{2\underline{\mu}}} + \sqrt{\frac{L_X^2 \bar{\beta}}{2\underline{\mu} r^2} + \frac{\|\nabla G(\bar{\mathbf{x}})\|}{r}} \right]^{-2} \right\}$
15: $\quad$ Set $\rho_{\text{new}} = \max\{\rho_{\text{old}}, \rho\}$
16: $\quad$ **return** $\rho_{\text{new}}$
17: **end procedure**

---

## 3 APD with progressive strong convexity estimation

We present the Accelerated Primal-Dual Algorithm with Progressive Strong Convexity Estimation (APDPro) to solve problem (1). For problem (1), APDPro achieves the improved convergence rate $\mathcal{O}(1/\sqrt{\varepsilon})$ without relying on the uniform strong convexity assumption [11, 22]. For the rest of this paper, we denote $\operatorname{prox}_{f,\mathcal{X}}(\mathbf{x} - \eta \mathbf{z}, \eta) := \operatorname{argmin}_{\hat{\mathbf{x}} \in \mathcal{X}} f(\hat{\mathbf{x}}) + \langle \mathbf{z}, \hat{\mathbf{x}} \rangle + \frac{1}{2\eta} \|\hat{\mathbf{x}} - \mathbf{x}\|^2$ as the proximal mapping.

We describe APDPro in Algorithm 1. The main component of APDPro contains a dual ascent step to update $\mathbf{y}_k$ based on the extrapolated gradient, followed by a primal proximal step to update $\mathbf{x}_k$. Compared with standard APD [11], APDPro has two more steps. First, line 4 of Algorithm 1 applies a novel cut constraint to separate the dual sequence $\{\mathbf{y}_k\}$ from the origin, which allows us to leverage the strong convexity of the Lagrangian function and hence obtain a faster rate of convergence than APD. Second, to use the strong convexity more effectively, in line 9, we perform a progressive estimation of the strong convexity by using the latest iterates $\mathbf{x}_k$ and $\bar{\mathbf{x}}_k$. Throughout the algorithm process, we use a routine IMPROVE to construct a non-decreasing sequence $\{\rho_k\}$, which provides increasingly refined lower bounds of the strong convexity of the Lagrangian function.

**The IMPROVE step** In order to estimate the strong convexity of the Lagrangian function, we rely on the subdifferential separation (eq. (4)) to bound the dual variables. From the first-order optimality condition in minimizing $\mathcal{L}(\mathbf{x}, \mathbf{y}^*)$ and the fact that $\mathbf{x}^* \in \operatorname{int} \mathcal{X}$ (Proposition 3), we have $\mathbf{0} \in \partial f(\mathbf{x}^*) + \nabla G(\mathbf{x}^*)\mathbf{y}^* + \mathcal{N}_{\mathcal{X}}(\mathbf{x}^*) = \partial f(\mathbf{x}^*) + \nabla G(\mathbf{x}^*)\mathbf{y}^*$. It follows from (4) that

$$r \leq \|\nabla G(\mathbf{x}^*)\mathbf{y}^*\| \leq \|\nabla G(\mathbf{x}^*)\| \cdot \|\mathbf{y}^*\| \leq \|\mathbf{y}^*\|_1 \|\nabla G(\mathbf{x}^*)\|, \tag{8}$$

where the last inequality use the fact that $\|\cdot\| \leq \|\cdot\|_1$. Note that the bound $\|\mathbf{y}^*\|_1 \geq r/\|\nabla G(\mathbf{x}^*)\|$ can not be readily used in the algorithm implementation because $\mathbf{x}^*$ is generally unknown. To resolve this issue, we develop more concrete dual lower bounds by using the generated solution $\hat{\mathbf{x}}$ in the proximity of $\mathbf{x}^*$. As we will show in the analysis, APDPro keeps track of two primal sequences $\{\mathbf{x}_k\}$ and $\{\bar{\mathbf{x}}_k\}$, for which we can establish bounds on $\|\mathbf{x}_k - \mathbf{x}^*\|^2$ and $(\mathbf{y}^*)^\top \boldsymbol{\mu} \cdot \|\hat{\mathbf{x}} - \mathbf{x}^*\|^2/2$, respectively. This drives us to develop the following lower bound property, with the proof provided in Appendix E.1.

**Proposition 4.** *Suppose Assumption 4 holds. Let* $\mathbf{y}^* \in \mathcal{Y}^*$ *be a dual optimal solution.*

*1. Suppose that* $\|\hat{\mathbf{x}} - \mathbf{x}^*\|^2 \leq 2\beta$, *then we have*

$$\|\mathbf{y}^*\|_1 \geq h_1(\hat{\mathbf{x}}, \beta) := r \left[ \|\nabla G(\hat{\mathbf{x}})\| + L_X \sqrt{2\beta} \right]^{-1}. \tag{9}$$

2. *Suppose* $(\mathbf{y}^*)^\top \boldsymbol{\mu} \cdot \|\hat{\mathbf{x}} - \mathbf{x}^*\|^2 \leq 2\beta$, *then we have*

$$\|\mathbf{y}^*\|_1 \geq h_2(\hat{\mathbf{x}}, \beta) := \left[ \frac{L_X}{r} \sqrt{\frac{\beta}{2\underline{\mu}}} + \sqrt{\frac{L_X^2 \beta}{2\underline{\mu} r^2} + \frac{\|\nabla G(\hat{\mathbf{x}})\|}{r}} \right]^{-2}. \tag{10}$$

Our next goal is to conduct the convergence analysis for APDPro in Theorem 1 and Corollary 1. Complete proof details are provided in Appendix E.2 and E.3.

**Theorem 1.** *Suppose for any* $\mathbf{y}^* \in \mathcal{Y}^*$, $(\mathbf{y}^*)^\top \boldsymbol{\mu} \geq \rho_0$ *holds, and let the sequence* $\{\tau_k, \sigma_k, t_k, \rho_{k+1}\}$ *generated by Algorithm 1 satisfy:*

$$t_{k+1}(\tau_{k+1}^{-1} - \rho_{k+1}) \leq t_k \tau_k^{-1}, \quad t_{k+1}\sigma_{k+1}^{-1} \leq t_k \sigma_k^{-1}, \quad L_{XY} + L_G^2 \sigma_k \leq \tau_k^{-1}. \tag{11}$$

*Then, the set* $\mathcal{Y}_k$ *is nonempty and* $\mathcal{Y}^* \subseteq \mathcal{Y}_k$. *Let* $\Delta(\mathbf{x}, \mathbf{y}) := \frac{1}{2\tau_0}\|\mathbf{x} - \mathbf{x}_0\|^2 + \frac{1}{2\sigma_0}\|\mathbf{y} - \mathbf{y}_0\|^2$, $\bar{\mathbf{y}}_K = T_K^{-1} \sum_{s=0}^{K-1} t_s \mathbf{y}_s$. *The sequence* $\{\bar{\mathbf{x}}_k, \mathbf{x}_k, \bar{\mathbf{y}}_k\}$ *generated by* APDPro *satisfies*

$$\frac{t_{K-1}\tau_{K-1}^{-1}}{2T_K}\|\mathbf{x}^* - \mathbf{x}_K\|^2 + \mathcal{L}(\bar{\mathbf{x}}_K, \mathbf{y}^*) - \mathcal{L}(\mathbf{x}^*, \bar{\mathbf{y}}_K) \leq \frac{1}{T_K}\Delta(\mathbf{x}^*, \mathbf{y}^*). \tag{12}$$

Next, we develop more concrete complexity results in Corollary 1.

**Corollary 1.** *Suppose that* $\sigma_k, \tau_k, t_k$ *satisfy:*

$$\tau_0^{-1} \geq L_{XY} + L_G^2 \sigma_0, \ t_k = \sigma_k/\sigma_0,$$
$$\tau_{k+1} = \tau_k/\sqrt{1 + \rho_{k+1}\tau_k}, \ \sigma_{k+1} = \sigma_k \tau_k/\tau_{k+1} \tag{13}$$

*Then we have*

$$f(\bar{\mathbf{x}}_K) - f(\mathbf{x}^*) \leq \frac{6}{6 + \tau_0 \tilde{\rho}_K (K+1)K}\left(\frac{1}{2\tau_0}\|\mathbf{x}_0 - \mathbf{x}^*\|^2 + \frac{D_Y^2}{2\sigma_0}\right),$$
$$\|[G(\bar{\mathbf{x}}_K)]_+\| \leq \frac{6}{c^*(6 + \tau_0 \tilde{\rho}_K (K+1)K)}\left(\frac{1}{2\tau_0}\|\mathbf{x}_0 - \mathbf{x}^*\|^2 + \frac{D_Y^2}{2\sigma_0}\right), \tag{14}$$
$$\frac{1}{2}\|\mathbf{x}_K - \mathbf{x}^*\|^2 \leq \frac{3\sigma_0}{\hat{\rho}_K^2 \tau_0^2 K^2 + 9(\sigma_0/\tau_0)}\Delta(\mathbf{x}^*, \mathbf{y}^*).$$

*where* $c^* := \left(f(\mathbf{x}^*) - \min_\mathbf{x} f(\mathbf{x})\right)/\min_{i \in [m]}\{-g_i(\tilde{\mathbf{x}})\} > 0$, $\tilde{\rho}_k = 2\sum_{s=0}^k \hat{\rho}_s s/\left(k(k+1)\right)$ *and* $\tilde{\rho}_k$ *satisfy the following condition,* $\hat{\rho}_{k+1} := \sqrt{\hat{\rho}_k^2 k^2 + (3\rho_{k+1}\hat{\rho}_k)k}/(k+1), \hat{\rho}_1 = 3\sqrt{\rho_1/\tau_0}$.

**Remark 3.** *In view of Corollary 1,* APDPro *obtains an iteration complexity of* $\mathcal{O}(1/\sqrt{\tilde{\rho}_K \varepsilon})$, *which is substantially better than the* $\mathcal{O}(1/\varepsilon)$ *bound of APD [11] and ConEx [4] when the strong convexity parameter* $\tilde{\rho}_K$ *is relatively large compared with* $\varepsilon$.

**Remark 4.** *Additionally, we argue that even when* $\tilde{\rho}_K = O(\varepsilon)$, APDPro *can obtain the matching* $\mathcal{O}(1/\varepsilon)$ *bound of the state-of-the-art algorithms. Specifically, using the definition of* $\sigma_k, \tau_k$, *we can easily derive the monotonicity of* $\{\sigma_k\}$. *It follows from* $\sigma_{k+1} = \tau_k \sigma_k/\tau_{k+1} = \tau_k \sigma_k/\left(\tau_k/\sqrt{1 + \rho_{k+1}\tau_k}\right) \geq \sigma_k$, *that* $T_k = \sum_{s=0}^{k-1} t_k = \sigma_0^{-1}\sum_{s=0}^{k-1} \sigma_k \geq k$. *Using a similar argument to that of Corollary 1, we obtain the bound* $f(\bar{\mathbf{x}}_K) - f(\mathbf{x}^*) \leq \mathcal{O}(1/K)$ *and* $\|[G(\bar{\mathbf{x}}_K)]_+\| \leq \mathcal{O}(1/K)$.

**Remark 5.** *The implementation of* APDPro *requires knowing an upper bound on* $\|\mathbf{y}^*\|$. *When the bound is unavailable, [11] developed an adaptive* APD *which still ensures the boundedness of dual sequence via line search. Since our main goal of this paper is to exploit the* lower-bound *rather than the* upper bound *of* $\|\mathbf{y}^*\|$, *we leave the extension for the future work.*

## 4 APDPro with a restart scheme

Note that in the worst case, APDPro exhibits an iteration complexity of $\mathcal{O}\left((D_X + D_Y)/\sqrt{\varepsilon}\right)$, which has a linear dependence on the diameter. While the $\mathcal{O}(1/\sqrt{\varepsilon})$ is optimal [25], it is possible to improve the complexity with respect to the primal part from $\mathcal{O}\left(D_X/\sqrt{\varepsilon}\right)$ to $\mathcal{O}\left(\log\left(D_X/\sqrt{\varepsilon}\right)\right)$. To achieve this goal, we propose a restart scheme (rAPDPro) that calls APDPro repeatedly and present the details in Algorithm 2. Inspired by [16], we set the iteration number as a function of the estimated strong convexity, detailed in the TERMINATEITER procedure. For convenience in describing a double-loop algorithm, we use superscripts for the number of epochs and subscripts for the number of

---

**Algorithm 2** Restarted APDPro (rAPDPro)

---

**Require:** $\rho_{N_{-1}}^{-1} \geq 0, \bar{\sigma} > 0, \nu_0 \in (0,1), \delta \in (0,1), \mathbf{x}_{N_{-1}}^{-1}, \mathbf{y}_{N_{-1}}^{-1}, S$

1: Compute $\bar{\tau} = (1 - \nu_0)\left(L_{XY} + L_G^2 \bar{\sigma}/\delta\right)^{-1}$
2: **for** $s = 0, 1, \ldots, S$ **do**
3:      $\tau_0^s = \bar{\tau}, \sigma_0^s = \bar{\sigma}, (\mathbf{x}_{-1}^s, \mathbf{y}_{-1}^s) \leftarrow (\mathbf{x}_{N_{s-1}}^{s-1}, \mathbf{y}_{N_{s-1}}^{s-1}), (\mathbf{x}_0^s, \mathbf{y}_0^s) \leftarrow (\mathbf{x}_{N_{s-1}}^{s-1}, \mathbf{y}_{N_{s-1}}^{s-1}), \rho_0^s = \rho_{N_{s-1}}^{s-1}$
4:      Set $\Delta_{XY} = \frac{1}{\tau_0^s}D_X^2 + \frac{1}{2\sigma_0^s}D_Y^2, \sigma_{-1}^s \leftarrow \sigma_0^s, T_0^s = 0, k = 0, \hat{\rho}_0^s = 1, N_s = \infty$
5:      **while** $k < N_s$ **do**
6:          Run line 4-10 of APDPro with index set $(s, k)$
7:          Update $N_s, \hat{\rho}_{k+1}^s \leftarrow \text{TERMINATEITER}(\hat{\rho}_k^s, \rho_{k+1}^s, s, k), k \leftarrow k + 1$
8:      **end while**
9: **end for**
10: **Output:** $\mathbf{x}_{N_S}^S, \mathbf{y}_{N_S}^S$
11: **procedure** $\text{TERMINATEITER}(\hat{\rho}_{\text{old}}, \rho, s, k)$
12:      Compute $\hat{\rho}_{\text{new}} = \begin{cases} \frac{1}{k+1}\sqrt{\hat{\rho}_{\text{old}}^2 k^2 + 3\rho\hat{\rho}_{\text{old}}k} & k > 1 \\ 3\sqrt{\rho/\tau_0} & k = 1 \end{cases}$
13:      Compute $N = \lceil \max\{6(\hat{\rho}_{\text{new}}\tau_0^s)^{-1}, \sqrt{2}^s \cdot 3\sqrt{2}D_Y/(\hat{\rho}_{\text{new}}D_X\sqrt{\tau_0^s\sigma_0^s})\} \rceil$
14:      **return** $N, \hat{\rho}_{\text{new}}$
15: **end procedure**

---

sub-iterations in parameters $\mathbf{x}, \mathbf{y}, \tau, \sigma$, e.g., $\mathbf{x}_1^S$ meaning the $\mathbf{x}$ output of first iterations at $S$-th epoch. To avoid redundancy in the Algorithm 2, we call the APDPro iteration directly. Note that the notation system here is identical to that of APDPro, with the only difference being the use of superscripts to distinguish the number of epochs.

In Theorem 2, we show the overall convergence complexity of rAPDPro with the proof provided in Appendix F.1.

**Theorem 2.** *Let $\{\mathbf{x}_0^s\}_{s \geq 0}$ be the sequence generated by* rAPDPro*, then we have*

$$\|\mathbf{x}_0^s - \mathbf{x}^*\|^2 \leq \Delta_s \equiv D_X^2 \cdot 2^{-s}, \quad \forall s \geq 0. \tag{15}$$

*As a consequence,* rAPDPro *will find a solution $\mathbf{x}_0^S$ such that $\|\mathbf{x}_0^S - \mathbf{x}^*\|^2 \leq \varepsilon$ for any $\varepsilon \in (0, D_X^2)$ in at most $S := \lceil \log_2(D_X^2/\varepsilon) \rceil$ epochs. Moreover, The iteration number of* rAPDPro *to find $\mathbf{x}_0^S$ such that $\|\mathbf{x}_0^S - \mathbf{x}^*\|^2 \leq \varepsilon$ is bounded by*

$$T_\varepsilon := \left(\frac{12}{\varpi_1 \tau_0^s} + 2\right)\left\lceil \log_2 \frac{D_X}{\sqrt{\varepsilon}} + 1 \right\rceil + \left(\frac{6(\sqrt{2}+2)}{\varpi_2\sqrt{\tau_0^s\sigma_0^s}}\right) \cdot \left(\frac{D_Y}{\sqrt{\varepsilon}}\right), \tag{16}$$

*where $\varpi_1$ and $\varpi_2$ satisfy $\sum_{s=0}^{S}(\hat{\rho}_{N_s}^s)^{-1} = (\varpi_1)^{-1}(S + 1)$ and $\sum_{s=0}^{S}\sqrt{2}^s/\hat{\rho}_{N_s}^s = (\varpi_2)^{-1}\sum_{s=0}^{S}\sqrt{2}^s$, respectively.*

**Remark 6.** *The bound $T_\varepsilon$ depends on $\varepsilon$, $\varpi_1$ and $\varpi_2$. If $\varpi_1 = O\left((-\log_2\sqrt{\varepsilon})^{-1}\right)$ or $\varpi_2 = O(\sqrt{\varepsilon})$, then we have $T_\varepsilon = \infty$, which implies that we can not guarantee $\|\mathbf{x}_0^s - \mathbf{x}^*\| \leq \varepsilon$ at finite iterations. $T_\varepsilon = \infty$ implies that there exists an epoch with infinite sub-iterations. Hence,* rAPDPro *is reduced to* APDPro *if we only consider that epoch.*

**Remark 7.** *Comparison of* rAPDPro *and* APDPro *involves a number of factors. In particular,* rAPDPro *compares favorably against* APDPro *if $\|\mathbf{x}_0 - \mathbf{x}^*\| = \tilde{\Omega}(\sqrt{\varepsilon}\log D_X)$. Moreover, the complexity (16) can be slightly improved if $D_X$ is replaced by any tighter upper bound of $\|\mathbf{x}_0^s - \mathbf{x}^*\|$. However, it is still unknown whether we can directly replace $D_X$ with $\|\mathbf{x}_0^s - \mathbf{x}^*\|$ in (16).*

**Dual Convergence** For dual variables, we establish asymptotic convergence to the optimal solution, a key condition for developing the active-set identification in the later section. For ease in notation, it is more convenient to label the generated solution as a whole sequence using a single subscript index: $\mathbf{x}_1, \mathbf{x}_2, \ldots, \mathbf{x}_N; \mathbf{y}_1, \mathbf{y}_2, \ldots, \mathbf{y}_N$. Hence, we use the index system $j$ and $(s, k)$ interchangeably. Note that $\{\mathbf{x}_0^{s+1}, \mathbf{y}_0^{s+1}\}$ and $\{\mathbf{x}_{N_s+1}^s, \mathbf{y}_{N_s+1}^s\}$ correspond to the same pair of points. We present the dual asymptotic result in the following theorem, with the proof provided in Appendix F.2.

**Theorem 3.** *Assume $\bar{\tau}^{-1} > \bar{\rho}$ and choose $\nu_0 > 0$ such that $1 > \inf_{j \geq 0}\{\sigma_{j-1}/\sigma_j\} \geq \delta + \nu_0$. We have $(\mathbf{x}^*, \mathbf{y}^*)$ satisfy the KKT condition, where $\mathbf{y}^*$ is any limit point of $\{\mathbf{y}_j\}$ generated by* rAPDPro.

**Remark 8.** *To establish the asymptotic convergence of the dual variable, we introduce an additional constant $\delta \in (0, 1)$, which implies that the initial step size must meet a stricter requirement than the convergence condition specified in Corollary 1. Since $\sigma_k^s/\sigma_{k-1}^s = \sqrt{1 + \rho_k^s \tau_k^s}$, $\{\rho_k^s\}$ is bounded due to the boundedness of the dual variable, $\{\tau_k^s\}$ is monotonically decreasing, then $\inf_{0 \leq k \leq N_s}\{\sigma_{k-1}^s/\sigma_k^s\} \geq (1 + \overline{\rho}\overline{\tau})^{-1/2}$. Hence, inequality, $1 > \inf_{j\geq 0}\{\sigma_{j-1}/\sigma_j\} \geq \delta + \nu_0$, is always satisfiable if we choose proper $\delta, \nu_0$ such that $(1 + \overline{\rho}\overline{\tau})^{-1/2} \geq \delta + \nu_0$. Furthermore, Assumption $(\overline{\tau})^{-1} > \overline{\rho}$ is mild. Since we always choose $\overline{\sigma}$ large enough in rAPDPro, $\overline{\tau}$ can be sufficiently small.*

**Remark 9.** *Both algorithms proposed previously require solving quadratic optimization with linear constraints when updating dual variables, which may introduce implementation overheads when the constraint number is high. Inspired by the multi-stage algorithm, we additionally propose an algorithm (Multi-Stage APD, msAPD) that uses different step sizes in different stages and dynamically adjusts the number of iterations in each stage by leveraging strong convexity, as detailed in Appendix H.*

## 5 Active-set identification in sparsity-inducing optimization

In this section, we apply our proposed algorithms to the aforementioned sparse learning problem:

$$\min f(\mathbf{x}), \text{ s.t. } g(\mathbf{x}) \leq 0, \ \mathbf{x} = \mathbf{x}_{(1)} \times \ldots \times \mathbf{x}_{(B)}, \ \mathbf{x}_{(i)} \in \mathbb{R}^{n_i}, 1 \leq i \leq B, \tag{17}$$

where $f(\mathbf{x}) = \sum_{i=1}^B p_i \|\mathbf{x}_{(i)}\|$ is the group Lasso regularizer and $g(\mathbf{x})$ is a strongly convex function. We use $\mathbf{x}_{(i)}$ to express the $i$-th block coordinates of $\mathbf{x}$. The goal of this section is to show that rAPDPro can identify the sparsity pattern of the optimal solution of (17) in a finite number of iterations.

In general, suppose that $f(\mathbf{x})$ has a separable structure $f(\mathbf{x}) = \sum_{i=1}^B f_i(\mathbf{x}_{(i)})$, we define the active set $\mathcal{A}(\mathbf{x})$ for $f(\mathbf{x})$ by $\mathcal{A}(\mathbf{x}) := \{i : \partial f_i(\mathbf{x}_{(i)}) \text{ is not a singleton}\}$. For $f(\mathbf{x}) = \sum_{i=1}^B p_i \|\mathbf{x}_{(i)}\|$, it is easy to see that $\mathcal{A}(\mathbf{x})$ is the index set of the zero blocks: $\mathcal{A}(\mathbf{x}^*) = \{i : \mathbf{x}_{(i)}^* = \mathbf{0}\}$. Next, we describe one property for the optimal solution of (17) in Proposition 5 with the proof provided in Appendix G.1.

**Proposition 5.** *Under Assumptions 1 and 2, the KKT point for* (17) *is unique.*

To identify the sparsity pattern (active set) of the optimal solution, it is common to assume the existence of a non-degenerate optimal solution, which is stronger than the standard optimality condition [24, 29]. We say that $\mathbf{x}^*$ is non-degenerate if $\mathbf{0} \in \mathbf{ri}\, \partial \mathcal{L}(\mathbf{x}^*, \mathbf{y}^*) = \mathbf{ri}(\partial f(\mathbf{x}^*) + \nabla g(\mathbf{x}^*)\mathbf{y}^*)$ for the Lagrangian multiplier $\mathbf{y}^*$, where $\mathbf{ri}$ stands for the relative interior. More specifically, $(\mathbf{x}^*, \mathbf{y}^*)$ satisfies the block-wise optimality condition

$$\begin{cases} -[\nabla g(\mathbf{x}^*)\mathbf{y}^*]_{(i)} = \nabla f_i(\mathbf{x}_{(i)}^*), & \text{if } i \notin \mathcal{A}(\mathbf{x}^*), \\ -[\nabla g(\mathbf{x}^*)\mathbf{y}^*]_{(i)} \in \mathbf{int}\left(\partial f_i(\mathbf{x}_{(i)}^*)\right), & \text{if } i \in \mathcal{A}(\mathbf{x}^*). \end{cases}$$

Inspired by [24], we use the radius $\eta := \min_{i \in \mathcal{A}(\mathbf{x}^*)}\{p_i - \|[\nabla g(\mathbf{x}^*)\mathbf{y}^*]_{(i)}\|\}$, which describes the certain distance between the gradient and "subdifferential boundary" of the active set. We demonstrate in the following theorem that the optimal sparsity pattern is identified when the iterates fall in a neighborhood dependent on $\eta$, with the proof provided in Appendix G.2.

**Theorem 4.** *Set $\mathcal{X} := \mathcal{B}\left(\tilde{\mathbf{x}}, \min_{i \in [m]} 2\sqrt{\frac{-2g_i(\mathbf{x}_i^*)}{\mu_i}} + \zeta\right)$ with $\zeta > 0$ and $3L_{XY} \cdot (\overline{\tau} + (2L_{XY})^{-1}) \cdot \zeta > \eta\overline{\tau}$ in rAPDPro, then we have there exists a epoch $\hat{S}_0$ such that $\mathbf{x}_{(i)}^* = \mathbf{x}_{k(i)}^s, s \geq \hat{S}_0, \forall k \in [N_s], \forall i \in \mathcal{A}(\mathbf{x}^*)$.*

**Remark 10.** *The active-set identification result is achieved using the optimality condition at the next iterate $\mathbf{x}_i^{k+1}$. To ensure $\mathbf{x}_i^{k+1} \in \mathbf{int}\, \mathcal{X}$, we define an expanded region, which prevents cases where the normal cone differs from $\{\mathbf{0}\}$.*

## 6 Numerical study

In this section, we examine the empirical performance of our proposed algorithms for solving the sparse Personalized PageRank [8, 9, 23]. The constrained form of Personalized PageRank can be

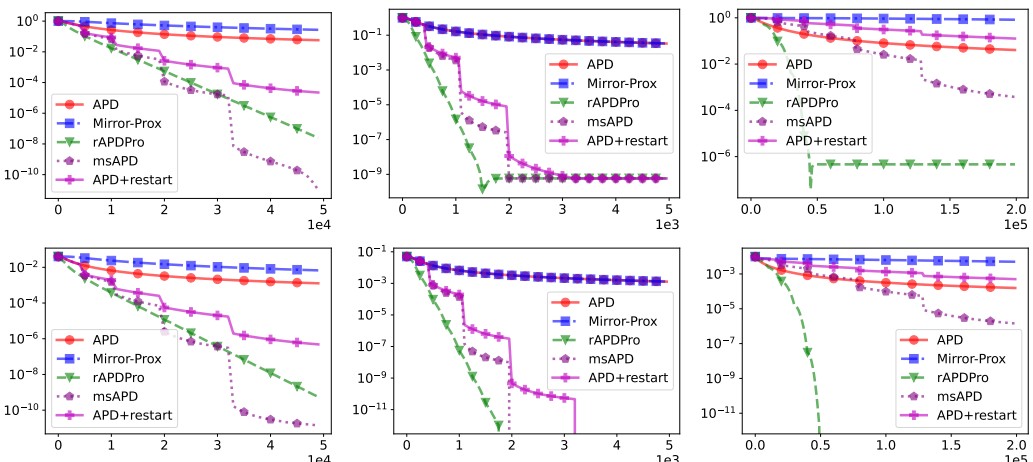

Figure 1: The first row describes the convergence to optimum, where the $y$-axis reports $\log_{10}((\|D^{1/2}\mathbf{x}_k\|_1 - \|D^{1/2}\mathbf{x}^*\|_1)/\|D^{1/2}\mathbf{x}^*\|_1)$ for rAPDPro, and $\log_{10}((\|D^{1/2}\bar{\mathbf{x}}_k\|_1 - \|D^{1/2}\mathbf{x}^*\|_1)/\|D^{1/2}\mathbf{x}^*\|_1)$ for APD, APD+restart, msAPD and Mirror-Prox ($\mathbf{x}^*$ is computed by MOSEK [1]). The second row describes feasibility violation, where $y$-axis reports the feasibility gap $\log_{10}(\max\{0, G(\mathbf{x}_k)\})$ for rAPDPro, and $\log_{10}(\max\{0, G(\bar{\mathbf{x}}_k)\})$ for APD, msAPD and Mirror-Prox. Datasets (Left-Right order) correspond to bio-CE-HT, bio-CE-LC and econ-beaflw.

written as follows: $\min_{\mathbf{x}\in\mathbb{R}^n} \quad \|D^{1/2}\mathbf{x}\|_1$ s.t. $\frac{1}{2}\langle \mathbf{x}, Q\mathbf{x}\rangle - \alpha\langle \mathbf{s}, D^{-1/2}\mathbf{x}\rangle \leq b$, where $Q, D$ and $\mathbf{s}$ are generated by graph. We implement both rAPDPro and msAPD. We skip APDPro as we observe that the restart strategy consistently improves the algorithm performance. For comparison, we consider the state-of-the-art accelerated primal-dual (APD) method [11], APD with restart mechanism at fixed iterations (APD+restart) and Mirror-Prox [13]. 6 small to medium-scale datasets from various domains in the Network Datasets [28] are selected in our experiments. All experiments are implemented on Mac mini M2 Pro, 32GB. Due to the page limit, we only report results on three datasets and leave more details in the last Appendix I.

We plot the relative function value gap $|f(\mathbf{x}) - f(\mathbf{x}^*)|/|f(\mathbf{x}^*)|$ and the feasibility violation $\max\{G(\mathbf{x}), 0\}$ over the iteration number in Figure 1, respectively. Firstly, in terms of both optimality gap and constraint violation, the performance of rAPDPro and msAPD is significantly better than that of APD, APD+restart and Mirror-Prox. Additionally, rAPDPro and msAPD often converge to high-precision solutions. Secondly, based on the experimental results, it is indeed observed that msAPD exhibits a periodic variation in convergence performance, which aligns with our algorithm theory.

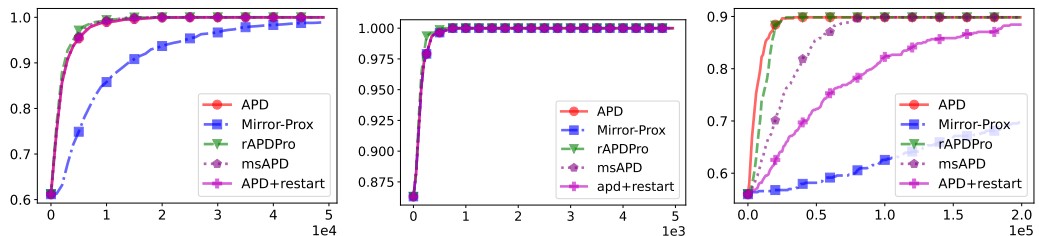

Figure 2: The experimental results on active-set identification. Datasets (Left-Right order) correspond to bio-CE-HT, bio-CE-LC and econ-beaflw. The $x$-axis reports the iteration number and the $y$-axis reports accuracy in active-set identification.

Next, we examine the algorithm's effectiveness in identifying sparsity patterns. We computed a nearly optimal solution $\mathbf{x}^*$ from MOSEK. Note that $\mathbf{x}^*$ is a dense vector. For numerical consideration, we truncate the coordinate values of $\mathbf{x}^*$ to zero if the absolute value is below $10^{-8}$ and perform the same truncation to all the generated solutions of the compared algorithms. Then we use $(|\mathcal{A}(\mathbf{x}) \cap$

$\mathcal{A}(\mathbf{x}^*)| + |\mathcal{A}^c(\mathbf{x}) \cap \mathcal{A}^c(\mathbf{x}^*)|)/n$ to measure the accuracy of identifying the active set, where $|\cdot|$ denotes the set cardinality. For rAPDPro, we consider the last iterate $\mathbf{x}_k$ while for APD, msAPD and Mirror-Prox, we plot the result on $\bar{\mathbf{x}}_k$, as these are the solutions where the convergence rates are established. Figure 2 plots the experiment result, from which we observe that rAPDPro and msAPD are highly effective in identifying the active set. Often, they are able to recognize the structure of the active set within a small number of iterations. Overall, the experimental results show the great potential of our proposed algorithms in identifying the sparsity structure and are consistent with our theoretical analysis.

## 7    Conclusion

The key contribution of this paper is that we develop several new first-order primal-dual algorithms for convex optimization with strongly convex constraints. Using some novel strategies to exploit the strong convexity of the Lagrangian function, we substantially improve the best convergence rate from $\mathcal{O}(1/\varepsilon)$ to $\mathcal{O}(1/\sqrt{\varepsilon})$. In the application of constrained sparse learning problems, the experimental study confirms the advantage of our proposed algorithms against state-of-the-art first-order methods for constrained optimization. Moreover, we show that one of our proposed algorithms rAPDPro has the favorable feature of identifying the sparsity pattern in the optimal solution. For future work, one direction is to apply the adaptive strategy, such as line search, to our framework to deal with cases when the dual bound is unavailable. Another interesting direction is to further exploit the active set identification property in a general setting. For example, it would be interesting to incorporate our algorithm with active constraint identification, which could be highly desirable when there are a large number of constraints. It would also be interesting to consider a more general convex objective when the proximal operator is not easy to compute.

## Acknowledgement

This research is partially supported by the Major Program of National Natural Science Foundation of China (Grant 72394360, 72394364), Natural Science Foundation of Shanghai (Grant No. 24ZR1421300). We sincerely thank all the reviewers for their valuable suggestions, which have significantly improved the quality of our article.

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

# Appendix

## Structure of the Appendix

The appendix is structured as follows: Appendix A introduces some limitations of our methods, primarily concerning the application scenarios of our algorithm. Appendix B includes comparisons between ours and some related Frank-Wolfe methods. We give some auxiliary lemmas in Appendix C, which are very important for the proofs presented later. Appendix D, E, F and G present the proof of conclusion in Section 2, 3, 4 and 5, respectively. Furthermore, Appendix H introduces a new algorithm to obtain a convergence rate without complicated dual updating. Finally, Appendix I offers more extensive details on our experiments.

## A  Limitations

In this paper, we focus on the theoretical analysis of convex optimization. Although our proposed algorithms for the convex optimization with strongly convex constraints can theoretically improve the existing results from $\mathcal{O}(1/\varepsilon)$ to $\mathcal{O}(1/\sqrt{\varepsilon})$. However, we still need to point out that our optimization algorithm has the following limitations. One is the algorithm needs a lower bound on the norm of sub-gradients of the objective function in the optimal solution, which may not be satisfied for all functions. On the other hand, we require consistent smoothness of the constraints to ensure convergence, and how to use the line search method to ensure convergence is a future direction.

## B  Comparison with Frank-Wolfe

We note that the strongly convex function constraint in (1) is a special case of a strongly convex set constraint, as demonstrated in [15]. Over the strongly convex set, it has been shown that Frank-Wolfe Algorithm (FW) can obtain convergence rates substantially better than the worst-case $\mathcal{O}(1/\varepsilon)$ rate. Under the bounded gradient assumption, [7, 20] show that FW obtains linear convergence over a strongly convex set. Nevertheless, the uniform bounded gradient assumption appears to be stronger than ours, as we only impose the lower boundedness assumption on the optimal solution $\mathbf{x}^*$ and allow the objective to be non-differentiable. More recently, [10] shows that FW obtains an $\mathcal{O}(1/\sqrt{\varepsilon})$ rate when the gradient is the order of the square root of the function value gap. For more recent progress, please refer to [5]. Despite the attractive convergence property, FW exhibits certain limitations when applied to the general function constraints (1) addressed in this paper. Specifically, FW involves a sequence of linear optimization problems throughout the iterations. While linear optimization over certain strongly convex sets, such as $\ell_p$-ball, admits a closed-form solution, there exists no efficient routine to handle general function constraints explored in this paper.

## C  Auxiliary lemmas

The following three-point property is important in the convergence analysis.

**Lemma 1.** *Let $f : \mathbb{R}^n \to \mathbb{R}\cup\{+\infty\}$ be a closed strongly convex function with modulus $\mu \geq 0$. Give $\bar{\mathbf{x}} \in \mathcal{X}$, where $\mathcal{X}$ is a compact convex set and $t \geq 0$, let $\mathbf{x}^+ = \operatorname{argmin}_{x \in \mathcal{X}} f(\mathbf{x}) + \frac{t}{2}\|\mathbf{x} - \bar{\mathbf{x}}\|^2$, then for all $\mathbf{x} \in \mathcal{X}$, we have*

$$f(\mathbf{x}) + \tfrac{t}{2}\|\mathbf{x} - \bar{\mathbf{x}}\|^2 \geq f(\mathbf{x}^+) + \tfrac{t+\mu}{2}\|\mathbf{x}^+ - \mathbf{x}\|^2 + \tfrac{t}{2}\|\mathbf{x}^+ - \bar{\mathbf{x}}\|^2.$$

*Proof.* Since $\mathcal{X}$ is a convex compact set, $\phi(x) := I_{\mathcal{X}}(\mathbf{x}) + f(\mathbf{x}) + \frac{t}{2}\|\mathbf{x} - \bar{\mathbf{x}}\|^2$ is lower-semi-continuous and $(\mu + t)$-strongly convex, where $I_{\mathcal{X}}(\mathbf{x}) = \begin{cases} 0 & \mathbf{x} \in \mathcal{X} \\ \infty & \mathbf{x} \notin \mathcal{X} \end{cases}$. Using the optimality $(\mathbf{0} \in \phi(\mathbf{x}^+))$ and strong convexity, we have $\phi(\mathbf{x}) \geq \phi(\mathbf{x}^+) + \langle \mathbf{0}, \mathbf{x} - \mathbf{x}^+ \rangle + \frac{t+\mu}{2}\|\mathbf{x}^+ - \mathbf{x}\|^2$, for any $\mathbf{x} \in \mathcal{X}$. This immediately gives the desired relation. □

The following result is adjusted from the classic supermartingale convergence theorem [26, Theorem 1]. We give proof for completeness.

**Lemma 2.** *Let $(\Omega, \mathcal{F}, \mathbb{P})$ be a probability space and $\mathcal{F}_1 \subset \mathcal{F}_2 \subset \cdots$ be a sequence of sub-$\sigma$-algebras of $\mathcal{F}$. For each $j = 1, 2, \cdots$, let $a_j, b_j$ and $c_j$ be non-negative $\mathcal{F}_n$-measure random variables such $\mathbb{E}[a_{j+1} \mid \mathcal{F}_j] \leq a_j - b_j + c_j$, then we have $\lim_{j \to \infty} a_j < \infty$ exists and $\sum_{j=1}^{\infty} b_j < \infty$ a.s. when $\sum_{j=1}^{\infty} c_j < \infty$.*

*Proof.* Define $d_j = a_j - \sum_{l=1}^{j-1}(c_l - b_l)$ and for any $\bar{a} > 0$, define $t = \inf\{t : \sum_{l=1}^{t} c_l > \bar{a}\}$. If $j < t$, we have

$$\mathbb{E}[d_{j+1} \mid \mathcal{F}_j] = \mathbb{E}[a_{j+1} - \sum_{l=1}^{j}(c_l - b_l) \mid \mathcal{F}_j] \overset{(a)}{\leq} a_j - \sum_{l=1}^{j-1}(c_l - b_l) =: d_j, \qquad (18)$$

where $(a)$ holds by $\mathbb{E}[a_{j+1} \mid \mathcal{F}_j] \leq a_j + c_j - b_j$, and hence

$$\mathbb{E}[d_{\min\{t,(j+1)\}} \mid \mathcal{F}_j] = d_t \mathbb{I}_{\{t \leq j\}} + \mathbb{E}[d_{j+1} \mid \mathcal{F}_j]\mathbb{I}_{\{t > j\}} \overset{(a)}{\leq} d_{\min\{t,j\}},$$

where $(a)$ holds by (18). Therefore, we have $\{d_{\min\{t,(j+1)\}}, \mathcal{F}_j, 1 \le j \le \infty\}$ is a supermartingale. Since

$$d_{\min\{t,j\}} = a_{\min\{t,j\}} - \sum_{l=1}^{\min\{t,j\}-1}(c_l - b_l) \overset{(a)}{\ge} - \sum_{l=1}^{\min\{t,(j-1)\}} c_l \ge -\bar{a},$$

holds for all $j$, where $(a)$ holds by $a_{\min\{t,j\}}, b_l \ge 0$. Then it follows from the martingale convergence theorem that $\lim_{j\to\infty} d_{\min\{t,j\}}$ exists and is finite a.s., i.e., $\lim_{j\to\infty} d_j$ exists and is finite on $\{t = \infty\} = \{\sum_{j=1}^{\infty} c_j \le \bar{a}\}$. Since $\bar{a}$ is arbitrary, we see that $\lim_{j\to\infty} d_j$ exists and is finite a.s. on $\{\sum_{j=1}^{\infty} c_j < \infty\}$. By $d_j = a_j - \sum_{l=1}^{j-1}(c_l - b_l)$, we have $\lim_{j\to\infty} a_j$ exists and is finite and $\sum_{j=1}^{\infty} b_j < \infty$ when $\{\sum_{j=1}^{\infty} c_j < \infty\}$. $\qquad \square$

# D  Proof details in Section 2

## D.1  Proof of Proposition 1

*Proof.* Under Slater's CQ, it is standard to show that any optimal solution $\mathbf{x}^*$ will also satisfy the KKT condition. For example, one can refer to [3]. For any $\mathbf{x} \in \mathcal{X}_G$, we have

$$f(\mathbf{x}) + \langle \mathbf{y}^*, G(\mathbf{x}) \rangle \ge f(\mathbf{x}^*) + \langle \mathbf{y}^*, G(\mathbf{x}^*) \rangle = f(\mathbf{x}^*),$$

where the equality is from the complementary slackness. In view of the above result and the Slater's condition (i.e., $G(\tilde{\mathbf{x}}) < \mathbf{0}$), we have

$$f(\tilde{\mathbf{x}}) > f(\tilde{\mathbf{x}}) + \langle \mathbf{y}^*, G(\tilde{\mathbf{x}}) \rangle \ge f(\mathbf{x}^*). \tag{19}$$

Combining with fact $\|\mathbf{y}^*\|_1 \min_{i \in [m]}\{-g_i(\tilde{\mathbf{x}})\} \le -\langle \mathbf{y}^*, G(\tilde{\mathbf{x}}) \rangle$, then we have

$$\|\mathbf{y}^*\| \le \|\mathbf{y}^*\|_1 \le \frac{f(\tilde{\mathbf{x}}) - f(\mathbf{x}^*)}{\min_{i \in [m]}\{-g_i(\tilde{\mathbf{x}})\}} = \bar{c}, \tag{20}$$

where the last inequality is by $f(\mathbf{x}^*) \ge \min_{\mathbf{x} \in \mathbb{R}^n} f(\mathbf{x})$. $\qquad \square$

## D.2  Proof of Proposition 2

*Proof.* We prove the uniqueness property by contradiction. Suppose that there exist $(\mathbf{x}^*, \mathbf{y}^*), (\tilde{\mathbf{x}}^*, \tilde{\mathbf{y}}^*)$ satisfying the KKT condition, then from the complementary slackness, optimality of $\mathbf{x}^*$ and $\tilde{\mathbf{x}}^*$, we have

$$\mathcal{L}(\mathbf{x}^*, \mathbf{y}^*) = f(\mathbf{x}^*) = f(\tilde{\mathbf{x}}^*) = \mathcal{L}(\tilde{\mathbf{x}}^*, \tilde{\mathbf{y}}^*).$$

Moreover, we have $\mathcal{L}(\tilde{\mathbf{x}}^*, \tilde{\mathbf{y}}^*) \le \mathcal{L}(\mathbf{x}^*, \tilde{\mathbf{y}}^*) \le \mathcal{L}(\mathbf{x}^*, \mathbf{y}^*)$. Hence, we must have $\mathcal{L}(\tilde{\mathbf{x}}^*, \tilde{\mathbf{y}}^*) = \mathcal{L}(\mathbf{x}^*, \tilde{\mathbf{y}}^*)$. However, since Assumption 2 implies $\tilde{\mathbf{y}}^* \ne \mathbf{0}$, the strongly convex function $\mathcal{L}(\cdot, \tilde{\mathbf{y}}^*)$ has a unique optimizer. Therefore, we conclude that $\mathbf{x}^* = \tilde{\mathbf{x}}^*$.

Next, we show that the set of optimal dual variables for problem (1) is convex. Suppose that there exist two optimal dual variables $\mathbf{y}_1^*$ and $\mathbf{y}_2^*$ for the unique primal variable $\mathbf{x}^*$, both satisfying the KKT condition, then we have $\langle \mathbf{y}_1^*, G(\mathbf{x}^*) \rangle = \langle \mathbf{y}_2^*, G(\mathbf{x}^*) \rangle = 0$. This implies that any linear combination of $\mathbf{y}_1^*$ and $\mathbf{y}_2^*$ satisfy KKT condition, i.e., $\langle a\mathbf{y}_1^* + b\mathbf{y}_2^*, G(\mathbf{x}^*) \rangle = 0, \forall a, b$. From Proposition 1, we know any optimal dual variable falls into a bounded convex set $\mathcal{Y}$. The intersection of two convex sets is also a convex set. Hence, we complete our proof. $\qquad \square$

## D.3  Proof of Proposition 3

*Proof.* From the strong convexity of $g_i(\mathbf{x})$, we have $g_i(\mathbf{x}) \ge g_i(\mathbf{x}_i^*) + \frac{\mu_i}{2}\|\mathbf{x} - \mathbf{x}_i^*\|^2$, which implies

$$\begin{aligned}
\|\tilde{\mathbf{x}} - \mathbf{x}_i^*\|^2 &\le (g_i(\tilde{\mathbf{x}}) - g_i(\mathbf{x}_i^*))\frac{2}{\mu_i} \overset{(a)}{<} \frac{-2g_i(\mathbf{x}_i^*)}{\mu_i}, \\
\|\mathbf{x}^* - \mathbf{x}_i^*\|^2 &\le (g_i(\mathbf{x}^*) - g_i(\mathbf{x}_i^*))\frac{2}{\mu_i} \le \frac{-2g_i(\mathbf{x}_i^*)}{\mu_i},
\end{aligned} \tag{21}$$

where $(a)$ holds by $g_i(\tilde{\mathbf{x}}) < 0$. In view of the triangle inequality and the above result, we have

$$\|\tilde{\mathbf{x}} - \mathbf{x}^*\| \le \|\mathbf{x}_i^* - \mathbf{x}^*\| + \|\tilde{\mathbf{x}} - \mathbf{x}_i^*\| < 2\sqrt{\frac{-2g_i(\mathbf{x}_i^*)}{\mu_i}}.$$

Hence, $\mathbf{x}^* \in \text{int } \mathcal{B}\left(\tilde{\mathbf{x}}, \min_{i \in [m]} 2\sqrt{\frac{-2g_i(\mathbf{x}_i^*)}{\mu_i}}\right)$. $\qquad \square$

# E   Convergence analysis of APDPro

## E.1   Proof of Proposition 4

*Proof.* Using the triangle inequality and (5), we have
$$\|\nabla G(\mathbf{x}^*)\| - \|\nabla G(\hat{\mathbf{x}})\| \le \|\nabla G(\mathbf{x}^*) - \nabla G(\hat{\mathbf{x}})\| \le L_X \|\hat{\mathbf{x}} - \mathbf{x}^*\|.$$
Combining the above inequality and (8), we obtain
$$\frac{r}{\|\mathbf{y}^*\|_1} \le L_X \|\hat{\mathbf{x}} - \mathbf{x}^*\| + \|\nabla G(\hat{\mathbf{x}})\|. \tag{22}$$
Next, we develop more specific lower bounds on $\|\mathbf{y}\|_1$. i). Inequality (9) can be easily verified since we have $\|\hat{\mathbf{x}} - \mathbf{x}^*\| \le \sqrt{2\beta}$. ii). Suppose $(\mathbf{y}^*)^\top \boldsymbol{\mu} \cdot \|\hat{\mathbf{x}} - \mathbf{x}^*\|^2 \le 2\beta$, then together with (22) we have
$$\frac{r}{\|\mathbf{y}^*\|_1} \le L_X \sqrt{\frac{2\beta}{(\mathbf{y}^*)^\top \boldsymbol{\mu}}} + \|\nabla G(\hat{\mathbf{x}})\| \le L_X \sqrt{\frac{2\beta}{\underline{\mu}\|\mathbf{y}^*\|_1}} + \|\nabla G(\hat{\mathbf{x}})\|.$$
Note that the above inequality can be expressed as $at^2 - bt - c \le 0$ with $t = \|\mathbf{y}^*\|_1^{-1/2}$, $a = r$, $b = L_X \sqrt{2\beta/\underline{\mu}}$ and $c = \|\nabla G(\hat{\mathbf{x}})\|$. Standard analysis implies that $t \le (b + \sqrt{b^2 + 4ac})/2a$, which gives the desired bound (10).   $\square$

## E.2   Proof of Theorem 1

*Proof.* First, it is easy to verify by our construction that $\{\mathcal{Y}_k\}$ is a monotone sequence: $\mathcal{Y}_1 \supseteq \mathcal{Y}_2 \supseteq \ldots \supseteq \mathcal{Y}_k \ldots$. Our goal is to show $\mathcal{Y}^* \subseteq \mathcal{Y}_k$ holds for any $k \ge 0$ by induction. Note that $\mathcal{Y}^* \subseteq \mathcal{Y}_0$ immediately follows from our assumption that $(\mathbf{y}^*)^\top \boldsymbol{\mu} \ge \rho_0$, for any $\mathbf{y}^* \in \mathcal{Y}^*$. Suppose that $\mathcal{Y}^* \subseteq \mathcal{Y}_k$ holds for $k = 0, \ldots, K-1$, we claim:

1. For any $\mathbf{x} \in \mathcal{X}$ and $\mathbf{y} \in \mathcal{Y}^*$, we have
$$\mathcal{L}(\bar{\mathbf{x}}_K, \mathbf{y}) - \mathcal{L}(\mathbf{x}, \bar{\mathbf{y}}_K) \le \frac{1}{T_K} \Delta(\mathbf{x}, \mathbf{y}) - \frac{t_{K-1} \tau_{K-1}^{-1}}{2T_K} \|\mathbf{x} - \mathbf{x}_K\|^2. \tag{23}$$

2. $\mathcal{Y}^* \subseteq \mathcal{Y}_K$.

**Part 1.** For $k = 0, 1, 2, \ldots, K-1$, taking $-\langle \mathbf{z}_k, \cdot \rangle$ and $f(\cdot) + \langle \nabla G(\mathbf{x}_k) \mathbf{y}_{k+1}, \cdot \rangle$ in Lemma 1, the following relations
$$-\langle \mathbf{y}_{k+1} - \mathbf{y}, \mathbf{z}_k \rangle \le A_{k+1}, \tag{24}$$
$$f(\mathbf{x}_{k+1}) + \langle \mathbf{y}_{k+1}, \nabla G(\mathbf{x}_k)^\top (\mathbf{x}_{k+1} - \mathbf{x}) \rangle \le f(\mathbf{x}) + B_{k+1}, \tag{25}$$
where
$$A_{k+1} \triangleq \frac{1}{2\sigma_k} \left( \|\mathbf{y} - \mathbf{y}_k\|^2 - \|\mathbf{y} - \mathbf{y}_{k+1}\|^2 - \|\mathbf{y}_{k+1} - \mathbf{y}_k\|^2 \right), \tag{26}$$
$$B_{k+1} \triangleq \frac{1}{2\tau_k} \left( \|\mathbf{x} - \mathbf{x}_k\|^2 - \|\mathbf{x} - \mathbf{x}_{k+1}\|^2 - \|\mathbf{x}_{k+1} - \mathbf{x}_k\|^2 \right), \tag{27}$$
hold for any $\mathbf{x} \in \mathcal{X}$ and $\mathbf{y} \in \bigcap_{0 \le s \le k} \mathcal{Y}_s$. The existence of such $\mathbf{y}$ follows from our induction hypothesis. Since $\mathbf{y}_{k+1}^\top G(\cdot)$ is $\rho_k$-strongly convex, we have
$$\langle \mathbf{y}_{k+1}, \nabla G(\mathbf{x}_k)^\top (\mathbf{x}_{k+1} - \mathbf{x}) \rangle$$
$$\ge \langle \mathbf{y}_{k+1}, \nabla G(\mathbf{x}_k)^\top (\mathbf{x}_{k+1} - \mathbf{x}_k) \rangle$$
$$+ \langle \mathbf{y}_{k+1}, G(\mathbf{x}_{k+1}) - G(\mathbf{x}) \rangle - \langle \mathbf{y}_{k+1}, G(\mathbf{x}_{k+1}) - G(\mathbf{x}_k) \rangle + \frac{\rho_k}{2} \|\mathbf{x} - \mathbf{x}_k\|^2.$$
Combining this result and (25), we have
$$f(\mathbf{x}_{k+1}) - f(\mathbf{x}) + \langle \mathbf{y}_{k+1}, G(\mathbf{x}_{k+1}) - G(\mathbf{x}) \rangle$$
$$\le B_{k+1} - \langle \mathbf{y}_{k+1}, \nabla G(\mathbf{x}_k)^\top (\mathbf{x}_{k+1} - \mathbf{x}_k) \rangle + \langle \mathbf{y}_{k+1}, G(\mathbf{x}_{k+1}) - G(\mathbf{x}_k) \rangle - \frac{\rho_k}{2} \|\mathbf{x} - \mathbf{x}_k\|^2. \tag{28}$$
On the other hand, by the definition of $\mathbf{z}_k$, we have
$$\langle \mathbf{y} - \mathbf{y}_{k+1}, \mathbf{z}_k \rangle$$
$$= \langle \mathbf{y} - \mathbf{y}_{k+1}, G(\mathbf{x}_k) - G(\mathbf{x}_{k+1}) \rangle + \langle \mathbf{y} - \mathbf{y}_{k+1}, G(\mathbf{x}_{k+1}) \rangle$$
$$+ (\sigma_{k-1}/\sigma_k) \langle \mathbf{y} - \mathbf{y}_k, G(\mathbf{x}_k) - G(\mathbf{x}_{k-1}) \rangle + (\sigma_{k-1}/\sigma_k) \langle \mathbf{y}_k - \mathbf{y}_{k+1}, G(\mathbf{x}_k) - G(\mathbf{x}_{k-1}) \rangle. \tag{29}$$

Let us denote $\mathbf{q}_k = G(\mathbf{x}_k) - G(\mathbf{x}_{k-1})$ for brevity. Combining (24) and (29) yields

$$\langle \mathbf{y} - \mathbf{y}_{k+1}, G(\mathbf{x}_{k+1}) \rangle$$
$$\leq A_{k+1} + \langle \mathbf{y} - \mathbf{y}_{k+1}, G(\mathbf{x}_{k+1}) - G(\mathbf{x}_k) \rangle - (\sigma_{k-1}/\sigma_k)\langle \mathbf{y} - \mathbf{y}_k, \mathbf{q}_k \rangle - (\sigma_{k-1}/\sigma_k)\langle \mathbf{y}_k - \mathbf{y}_{k+1}, \mathbf{q}_k \rangle. \tag{30}$$

Putting (28) and (30) together, we have

$$\mathcal{L}(\mathbf{x}_{k+1}, \mathbf{y}) - \mathcal{L}(\mathbf{x}, \mathbf{y}_{k+1})$$
$$\leq A_{k+1} + B_{k+1} - \langle \mathbf{y}_{k+1}, \nabla G(\mathbf{x}_k)^\top(\mathbf{x}_{k+1} - \mathbf{x}_k) \rangle + \langle \mathbf{y}_{k+1}, G(\mathbf{x}_{k+1}) - G(\mathbf{x}_k) \rangle$$
$$+ \langle \mathbf{y} - \mathbf{y}_{k+1}, \mathbf{q}_{k+1} \rangle - (\sigma_{k-1}/\sigma_k)\langle \mathbf{y} - \mathbf{y}_k, \mathbf{q}_k \rangle + (\sigma_{k-1}/\sigma_k)\langle \mathbf{y}_{k+1} - \mathbf{y}_k, \mathbf{q}_k \rangle - \tfrac{\rho_k}{2}\|\mathbf{x} - \mathbf{x}_k\|^2$$
$$\leq A_{k+1} + B_{k+1} + \tfrac{L_{XY}}{2}\|\mathbf{x}_{k+1} - \mathbf{x}_k\|^2 - \tfrac{\rho_k}{2}\|\mathbf{x} - \mathbf{x}_k\|^2$$
$$+ \langle \mathbf{y} - \mathbf{y}_{k+1}, \mathbf{q}_{k+1} \rangle - (\sigma_{k-1}/\sigma_k)\langle \mathbf{y} - \mathbf{y}_k, \mathbf{q}_k \rangle + (\sigma_{k-1}/\sigma_k)\langle \mathbf{y}_{k+1} - \mathbf{y}_k, \mathbf{q}_k \rangle,$$

where the last inequality is by Lipschitz smoothness of $\langle \mathbf{y}_{k+1}, G(\cdot) \rangle$.

Next, we bound the term $\langle \mathbf{q}_k, \mathbf{y}_{k+1} - \mathbf{y}_k \rangle$ by Young's inequality, which gives

$$\langle \mathbf{y}_{k+1} - \mathbf{y}_k, \mathbf{q}_k \rangle \leq \tfrac{1}{2\sigma_{k-1}}\|\mathbf{y}_{k+1} - \mathbf{y}_k\|^2 + \tfrac{\sigma_{k-1}}{2}\|\mathbf{q}_k\|^2, \tag{31}$$

It follows from (31) and $\tfrac{\sigma_k}{2}\|\mathbf{q}_{k+1}\|^2 \leq \tfrac{L_G^2 \sigma_k}{2}\|\mathbf{x}_{k+1} - \mathbf{x}_k\|^2$ that

$$\mathcal{L}(\mathbf{x}_{k+1}, \mathbf{y}) - \mathcal{L}(\mathbf{x}, \mathbf{y}_{k+1})$$
$$\leq \tfrac{\tau_k^{-1} - \rho_k}{2}\|\mathbf{x} - \mathbf{x}_k\|^2 - \tfrac{\tau_k^{-1}}{2}\|\mathbf{x} - \mathbf{x}_{k+1}\|^2 + \tfrac{(\sigma_{k-1}/\sigma_k)\sigma_{k-1}}{2}\|\mathbf{q}_k\|^2 - \tfrac{\sigma_k}{2}\|\mathbf{q}_{k+1}\|^2$$
$$+ \tfrac{1}{2\sigma_k}\left(\|\mathbf{y} - \mathbf{y}_k\|^2 - \|\mathbf{y} - \mathbf{y}_{k+1}\|^2\right) + \langle \mathbf{y} - \mathbf{y}_{k+1}, \mathbf{q}_{k+1} \rangle - (\sigma_{k-1}/\sigma_k)\langle \mathbf{y} - \mathbf{y}_k, \mathbf{q}_k \rangle \tag{32}$$
$$- \tfrac{\sigma_k^{-1} - (\sigma_{k-1}/\sigma_k)/\sigma_{k-1}}{2}\|\mathbf{y}_{k+1} - \mathbf{y}_k\|^2 + \tfrac{L_G^2 \sigma_k}{2}\|\mathbf{x}_{k+1} - \mathbf{x}_k\|^2 - \tfrac{\tau_k^{-1} - L_{XY}}{2}\|\mathbf{x}_{k+1} - \mathbf{x}_k\|^2.$$

Multiply both sides of the above relation by $t_k$ and sum up the result for $k = 0, 1, \ldots, K - 1$. In view of the parameter relation (11), we have

$$\sum_{k=0}^{K-1} t_k \left[\mathcal{L}(\mathbf{x}_{k+1}, \mathbf{y}) - \mathcal{L}(\mathbf{x}, \mathbf{y}_{k+1})\right]$$
$$\overset{(a)}{\leq} \tfrac{t_0(\tau_0^{-1} - \rho_0)}{2}\|\mathbf{x} - \mathbf{x}_0\|^2 - \tfrac{t_{K-1}\tau_{K-1}^{-1}}{2}\|\mathbf{x} - \mathbf{x}_K\|^2 - \tfrac{t_{K-1}\sigma_{K-1}}{2}\|\mathbf{q}_K\|^2$$
$$+ \tfrac{t_0\sigma_0^{-1}}{2}\|\mathbf{y} - \mathbf{y}_0\|^2 - \tfrac{t_{K-1}\sigma_{K-1}^{-1}}{2}\|\mathbf{y} - \mathbf{y}_K\|^2 + t_{K-1}\langle \mathbf{y} - \mathbf{y}_K, \mathbf{q}_K \rangle - t_0\langle \mathbf{y} - \mathbf{y}_0, \mathbf{q}_0 \rangle \tag{33}$$
$$\overset{(b)}{\leq} \tfrac{1}{2\tau_0}\|\mathbf{x} - \mathbf{x}_0\|^2 + \tfrac{1}{2\sigma_0}\|\mathbf{y} - \mathbf{y}_0\|^2 - \tfrac{t_{K-1}\tau_{K-1}^{-1}}{2}\|\mathbf{x} - \mathbf{x}_K\|^2$$

where $(a)$ uses $\mathbf{q}_0 = \mathbf{0}$ and $\mathbf{x}_{-1} = \mathbf{x}_0$, and $(b)$ holds by $\rho_0 = 0$, $t_0 = 1$ and

$$t_{K-1}\langle \mathbf{y} - \mathbf{y}_K, \mathbf{q}_K \rangle \leq \tfrac{t_{K-1}}{2\sigma_{K-1}}\|\mathbf{y} - \mathbf{y}_K\|^2 + \tfrac{t_{K-1}}{2/\sigma_{K-1}}\|\mathbf{q}_K\|^2.$$

Since $\mathcal{L}(\mathbf{x}, \mathbf{y})$ is convex in $\mathbf{x}$ and linear in $\mathbf{y}$, we have

$$T_K\left[\mathcal{L}(\bar{\mathbf{x}}_K, \mathbf{y}) - \mathcal{L}(\mathbf{x}, \bar{\mathbf{y}}_K)\right] \leq \sum_{k=0}^{K-1} t_k\left[\mathcal{L}(\mathbf{x}_{k+1}, \mathbf{y}) - \mathcal{L}(\mathbf{x}, \mathbf{y}_{k+1})\right], \tag{34}$$

Combining (33) and (34), we obtain

$$T_K\left[\mathcal{L}(\bar{\mathbf{x}}_K, \mathbf{y}) - \mathcal{L}(\mathbf{x}, \bar{\mathbf{y}}_K)\right] \leq \tfrac{1}{2\tau_0}\|\mathbf{x} - \mathbf{x}_0\|^2 - \tfrac{t_{K-1}\tau_{K-1}^{-1}}{2}\|\mathbf{x} - \mathbf{x}_K\|^2 + \tfrac{1}{2\sigma_0}\|\mathbf{y} - \mathbf{y}_0\|^2. \tag{35}$$

Dividing both sides by $T_K$, we obtain the desired result (23).

Part 2. Next we show $\mathcal{Y}^* \subseteq \mathcal{Y}_K$. Let $\mathbf{y}^*$ be any point in $\mathcal{Y}^*$. Since (35) holds for any $\mathbf{x} \in \mathcal{X}$ and $\mathbf{y} \in \cap_{0 \leq k \leq K-1}\mathcal{Y}_k \supseteq \mathcal{Y}^*$, we can place $\mathbf{x} = \mathbf{x}^*, \mathbf{y} = \mathbf{y}^* \in \mathcal{Y}^*$ in (23) to obtain

$$\tfrac{t_{K-1}\tau_{K-1}^{-1}}{2T_K}\|\mathbf{x}^* - \mathbf{x}_K\|^2 + \mathcal{L}(\bar{\mathbf{x}}_K, \mathbf{y}^*) - \mathcal{L}(\mathbf{x}^*, \bar{\mathbf{y}}_K) \leq \tfrac{1}{T_K}\Delta(\mathbf{x}^*, \mathbf{y}^*).$$

Moreover, the strong convexity of $\mathcal{L}(\cdot, \mathbf{y}^*)$ implies

$$\mathcal{L}(\bar{\mathbf{x}}_K, \mathbf{y}^*) \geq \mathcal{L}(\mathbf{x}^*, \mathbf{y}^*) + \tfrac{(\mathbf{y}^*)^\top \boldsymbol{\mu}}{2}\|\bar{\mathbf{x}}_K - \mathbf{x}^*\|^2 \geq \mathcal{L}(\mathbf{x}^*, \bar{\mathbf{y}}_K) + \tfrac{(\mathbf{y}^*)^\top \boldsymbol{\mu}}{2}\|\bar{\mathbf{x}}_K - \mathbf{x}^*\|^2.$$

Applying the above two inequalities yields

$$\frac{(\mathbf{y}^*)^\top \boldsymbol{\mu}}{2}\|\bar{\mathbf{x}}_K - \mathbf{x}^*\|^2 \leq \frac{1}{T_K}\Delta(\mathbf{x}^*, \mathbf{y}^*), \ \frac{1}{2}\|\mathbf{x}_K - \mathbf{x}^*\|^2 \leq \frac{\tau_{K-1}\sigma_0}{\sigma_{K-1}}\Delta(\mathbf{x}^*, \mathbf{y}^*). \tag{36}$$

In view of (36) and Proposition 4, we have that

$$(\mathbf{y}^*)^T \boldsymbol{\mu} \geq \underline{\mu}\,\|\mathbf{y}^*\|_1 = \underline{\mu}\max\left\{h_1(\mathbf{x}_K, \tfrac{\sigma_0\tau_{K-1}\Delta_{XY}}{\sigma_{K-1}}), h_2(\bar{\mathbf{x}}_K, \tfrac{\Delta_{XY}}{T_K})\right\} := \hat{\rho}_K.$$

Moreover, since $\mathcal{Y}^* \subseteq \mathcal{Y}_{K-1}$, we have $(\mathbf{y}^*)^T \boldsymbol{\mu} \geq \rho_{K-1}$. Hence we have $(\mathbf{y}^*)^T \boldsymbol{\mu} \geq \rho_K$ where $\rho_K = \max\{\hat{\rho}_K, \rho_{K-1}\}$ is the output of the IMPROVE procedure. Due to the construction of $\mathcal{Y}_K$, we immediately see that $\mathbf{y}^* \in \mathcal{Y}_K$. This implies $\mathcal{Y}^* \subseteq \mathcal{Y}_K$ and completes our induction proof. $\qquad\square$

Next, we specify the stepsize selection in Lemma 3 and develop more concrete complexity results in Corollary 1.

**Lemma 3.** *Let* $\hat{\rho}_{k+1} := \frac{\sqrt{\hat{\rho}_k^2 k^2 + (3\rho_{k+1}\hat{\rho}_k)k}}{k+1}$ *for* $k \geq 1$ *and* $\hat{\rho}_1 = 3\sqrt{\frac{\rho_1}{\tau_0}}$. *Suppose* $\sigma_k, \tau_k$ *satisfy:*

$$\tau_0^{-1} \geq L_{XY} + L_G^2\sigma_0, \ \ \tau_{k+1} = \tau_k(1 + \rho_{k+1}\tau_k)^{-\frac{1}{2}}, \ \ \sigma_{k+1} = \frac{\tau_k\sigma_k}{\tau_{k+1}}. \tag{37}$$

*Then we have*

$$\frac{1}{\tau_k^2} \geq \frac{\hat{\rho}_k^2}{9}k^2 + \frac{1}{\tau_0^2}, \ \ T_k \geq 1 + \frac{\tau_0}{6}\tilde{\rho}_k(k+1)k, \ \ \hat{\rho}_k \geq \min\{\rho_1, \hat{\rho}_1\}, \tag{38}$$

*where* $\tilde{\rho}_k = 2\sum_{s=0}^{k}\frac{\hat{\rho}_s s}{k(k+1)}$ *for* $k \geq 1$. *Moreover, suppose* $\bar{\rho}\tau_0 \leq 2$, *where* $\bar{\rho} = \bar{c}\cdot\bar{\mu}$, *then we have*

$$\sigma_k^2 \leq \sigma_0^2(k+1)^2. \tag{39}$$

*Proof.* We first use induction to show that $\frac{1}{\tau_k^2} \geq \frac{\hat{\rho}_k^2}{9}k^2 + \frac{1}{\tau_0^2}$. It is easy to see that $\frac{1}{\tau_k^2} \geq \frac{\hat{\rho}_k^2}{9}k^2 + \frac{1}{\tau_0^2}$ holds for $k = 1$ by the definition $\hat{\rho}_1 = 3\sqrt{\rho_1/\tau_0}$ and $\tau_1 = \tau_0(1+\rho_1\tau_0)^{-\frac{1}{2}}$. Assume $\frac{1}{\tau_k^2} \geq \frac{\hat{\rho}_k^2}{9}k^2 + \frac{1}{\tau_0^2}$ holds for all $k = 0, \ldots, K$, then we have

$$\begin{aligned}
\frac{1}{\tau_{K+1}^2} &= \frac{1}{\tau_K^2} + \frac{\rho_{K+1}}{\tau_K} \\
&\geq \frac{\hat{\rho}_K^2}{9}K^2 + \frac{1}{\tau_0^2} + \rho_{K+1}\sqrt{\frac{\hat{\rho}_K^2}{9}K^2 + \frac{1}{\tau_0^2}} \\
&\geq \frac{\hat{\rho}_K^2}{9}K^2 + \frac{1}{\tau_0^2} + \frac{\rho_{K+1}\hat{\rho}_K K}{3} \\
&\geq \frac{\hat{\rho}_{K+1}^2}{9}(K+1)^2 + \frac{1}{\tau_0^2},
\end{aligned} \tag{40}$$

which completes our induction. It follows from $\frac{1}{\tau_k^2} \geq \frac{\hat{\rho}_k^2}{9}k^2 + \frac{1}{\tau_0^2}$ and the relation among $T_k, t_k, \sigma_k, \tau_k$ that, for any $k \geq 1$

$$\begin{aligned}
T_k &= \sum_{s=0}^{k-1}t_s = 1 + \sum_{s=1}^{k-1}t_s \geq 1 + \sum_{s=1}^{k-1}\frac{\sigma_s}{\sigma_0} = 1 + \sum_{s=1}^{k-1}\frac{\tau_0}{\tau_s} \geq 1 + \tau_0\sum_{s=1}^{k-1}\sqrt{\frac{\hat{\rho}_s^2 s^2}{9} + \frac{1}{\tau_0^2}} \\
&> 1 + \tau_0\sum_{s=1}^{k-1}\frac{\hat{\rho}_s s}{3} = 1 + \frac{\tau_0}{6}\tilde{\rho}_k(k+1)k.
\end{aligned} \tag{41}$$

Similarly, we use induction to prove

$$\hat{\rho}_k \geq \min\{\rho_1, \hat{\rho}_1\}, \forall k \geq 1. \tag{42}$$

It is easy to find that $\hat{\rho}_1 \geq \min\{\rho_1, \hat{\rho}_1\}$. We assume that $\hat{\rho}_k \geq \min\{\rho_1, \hat{\rho}_1\}, \forall k \geq 1$ holds for any $k = 1, \ldots, K$. Considering $\hat{\rho}_{K+1}$, we have

$$\begin{aligned}
\hat{\rho}_{K+1} &\geq \frac{1}{K+1}\sqrt{\hat{\rho}_K^2 K^2 + 3\rho_1\hat{\rho}_K K} \\
&\geq \frac{1}{K+1}\sqrt{(\min\{\rho_1, \hat{\rho}_1\})^2 K^2 + 3\rho_1\cdot\min\{\rho_1, \hat{\rho}_1\}K} \geq \min\{\rho_1, \hat{\rho}_1\},
\end{aligned}$$

which completes the induction. Moreover, we use induction to show $\sigma_k^2 \le \sigma_0^2(k+1)^2$. It is obvious that the inequality holds for $k = 0$. Assume the inequality holds for all $k = 0, \ldots, K$, then we have

$$
\begin{aligned}
\sigma_{K+1}^2 &= \sigma_K^2(1 + \rho_{K+1}\tfrac{\tau_0\sigma_0}{\sigma_K}) \\
&= \sigma_K^2 + \rho_{K+1}\tau_0\sigma_0\sigma_K \\
&\le \sigma_0^2\left((K+1)^2 + \rho_{K+1}\tau_0(K+1)\right) \\
&\le \sigma_0^2(K+2)^2,
\end{aligned}
\tag{43}
$$

where the last inequality use the relation $\rho_k \le \bar{\rho}, \forall k$, and $\bar{\rho}\tau_0 \le 2$.

$\square$

### E.3 Proof of Corollary 1

*Proof.* First, we show that the sequences $\{\tau_k, \sigma_k, t_k, \rho_k\}$ generated by APDPro satisfy the relationship in (11) in Theorem 1. The first part of (11) can be derived using the monotonicity of $\{\rho_k\}$ as follows:

$$
\begin{aligned}
t_{k+1}\left(\tau_{k+1}^{-1} - \rho_{k+1}\right) &= \sigma_0^{-1}\left(\sigma_{k+1}\tau_{k+1} - \sigma_{k+1}\rho_{k+1}\right) \\
&= \sigma_0^{-1}\left(\sigma_k\tau_k\tau_{k+1}^{-2} - \sigma_{k+1}\rho_{k+1}\right) \\
&= \sigma_0^{-1}\left(\sigma_k(1 + \rho_{k+1}\tau_k)/\tau_k - \sigma_{k+1}\rho_{k+1}\right) \\
&= \sigma_0^{-1}\left(\sigma_k/\tau_k + \rho_{k+1}\sigma_k - \sigma_{k+1}\rho_{k+1}\right) \\
&\le t_k\tau_k^{-1}
\end{aligned}
$$

The second part of (11) can be easily verified using the parameters setting.

Next, we prove the last term in (11) by induction. Firstly, it easy to verify that for any $\sigma_0 > 0$, there exists $\tau_0 \in (0, (L_{XY} + L_G^2\sigma_0)^{-1}]$ such that last term of (11) holds. Hence, when $k = 0$, the last term of (11) is directly from the first term of (13). Suppose that the last term of (11) holds for $k = 0, \ldots, K-1$. From $\sigma_{K-1}/\sigma_K = \tau_K/\tau_{K-1} \le 1$, we have

$$
\frac{1}{\tau_K} = \frac{\sigma_K}{\tau_{K-1}\sigma_{K-1}} \ge \frac{L_{XY}}{\sigma_{K-1}/\sigma_K} + L_G^2\sigma_K \ge L_{XY} + L_G^2\sigma_K.
\tag{44}
$$

Without loss of generality, place $\mathbf{x} = \mathbf{x}^*$, $\mathbf{y} = \mathbf{y}^+ := (\|\mathbf{y}^*\|_1 + c^*)\frac{[G(\bar{\mathbf{x}}_K)]_+}{\|[G(\bar{\mathbf{x}}_K)]_+\|}$ in (23), and using $\|\mathbf{y}^*\|_1 \le \bar{c}$ in Proposition 1. It is easy to see $\|\mathbf{y}^+\| = \|\mathbf{y}^*\|_1 + c^* \le \bar{c}$, and $\|\mathbf{y}^+\|_1 \ge \|\mathbf{y}^+\| = \|\mathbf{y}^*\|_1 + c^* \ge \|\mathbf{y}^*\|_1$, Hence, we conclude that $\mathbf{y}^+ \in \mathcal{Y}_k, \forall k \ge 0$.

Now observe that $\mathcal{L}(\bar{\mathbf{x}}_K, \mathbf{y}^*) - \mathcal{L}(\mathbf{x}^*, \mathbf{y}^*) \ge 0$, which implies $f(\bar{\mathbf{x}}_K) + \langle \mathbf{y}^*, G(\bar{\mathbf{x}}_K)\rangle - f(\mathbf{x}^*) \ge 0$. In view of $\langle \mathbf{y}^*, G(\bar{\mathbf{x}}_K)\rangle \le \langle \mathbf{y}^*, [G(\bar{\mathbf{x}}_K)]_+\rangle \le \|\mathbf{y}^*\| \cdot \|[G(\bar{\mathbf{x}}_K)]_+\|$, then we have

$$
f(\bar{\mathbf{x}}_K) + \|\mathbf{y}^*\| \cdot \|[G(\bar{\mathbf{x}}_K)]_+\| - f(\mathbf{x}^*) \ge 0.
\tag{45}
$$

Moreover, it follows from $\|\mathbf{y}^*\|_1 \ge \|\mathbf{y}^*\|$ that

$$
\begin{aligned}
\mathcal{L}(\bar{\mathbf{x}}_K, \mathbf{y}^+) - \mathcal{L}(\mathbf{x}^*, \bar{\mathbf{y}}_K) &\ge \mathcal{L}(\bar{\mathbf{x}}_K, \mathbf{y}^+) - \mathcal{L}(\mathbf{x}^*, \mathbf{y}^*) \\
&\ge f(\bar{\mathbf{x}}_K) + (\|\mathbf{y}^*\| + c^*)\|[G(\bar{\mathbf{x}}_K)]_+\| - f(\mathbf{x}^*).
\end{aligned}
\tag{46}
$$

Combining (45), (46) and (23), we obtain

$$
\max\left\{c^*\|[G(\bar{\mathbf{x}}_K)]_+\|, f(\bar{\mathbf{x}}_K) - f(\mathbf{x}^*)\right\} \le \frac{1}{T_K}\left(\frac{1}{2\tau_0}\|\mathbf{x}_0 - \mathbf{x}^*\|^2 + \frac{D_Y^2}{2\sigma_0}\right),
\tag{47}
$$

In view of the bound in (38) and the relation between $\tau_k, \sigma_k$, we can get

$$
\frac{\tau_k}{\sigma_k} \le \frac{3}{\hat{\rho}_k^2\tau_0^2k^2 + 9\sigma_0/\tau_0}.
\tag{48}
$$

In view of (47) and (38), we have

$$
\max\left\{c^*\|[G(\bar{\mathbf{x}}_K)]_+\|, f(\bar{\mathbf{x}}_K) - f(\mathbf{x}^*)\right\} \le \frac{6}{6 + \tau_0\hat{\rho}_K(K+1)K}\left(\frac{1}{2\tau_0}\|\mathbf{x}_0 - \mathbf{x}^*\|^2 + \frac{D_Y^2}{2\sigma_0}\right).
$$

Combining (23) and (48) yields $\frac{1}{2}\|\mathbf{x}_K - \mathbf{x}^*\|^2 \le 3\sigma_0\Delta(\mathbf{x}^*, \mathbf{y}^*)/(\hat{\rho}_K^2\tau_0^2K^2 + 9\sigma_0/\tau_0)$. $\square$

# F  Convergence analysis of rAPDPro

## F.1  Proof of Theorem 2

*Proof.* First, we show that the choice of $\tau_0^s = \bar{\tau}, \sigma_0^s = \bar{\sigma}, \forall s \geq 0$ satisfy the condition (13) in Corollary 1: $(\tau_0^s)^{-1} \geq (1 - \nu_0)(\tau_0^s)^{-1} = L_{XY} + cL_G^2\sigma_0^s/\delta \geq L_{XY} + cL_G^2\sigma_0^s$.

Next, we show (15) holds by induction. Clearly, (15) holds for $s = 0$. Assume $\|\mathbf{x}_0^s - \mathbf{x}^*\|^2 \leq \Delta_s$ holds for $s = 0, \ldots, S - 1$. Then by Theorem 1, we have

$$\|\mathbf{x}_0^S - \mathbf{x}^*\|^2 \leq \frac{\sigma_0^S \tau_{N_S}^S}{\sigma_{N_S}^S}\left(\frac{2}{\tau_0^S}\Delta_S + \frac{1}{\sigma_0^S}D_Y^2\right). \tag{49}$$

In view of the first bound in (38) and the relation between $\tau_{N_s}^s, \sigma_{N_s}^s$, we can get

$$\frac{\tau_{N_s}^s}{\sigma_{N_s}^s} \leq \frac{9}{\sigma_0^s \tau_0^s (\hat{\rho}_{N_s} N_s)^2}. \tag{50}$$

Combining (49) and (50) yields

$$\|\mathbf{x}_0^S - \mathbf{x}^*\|^2 \leq \frac{18}{(\hat{\rho}_{N_s}\tau_0^s N_s)^2} + \frac{9D_Y^2}{\sigma_0^s\tau_0^s(\hat{\rho}_{N_s} N_s)^2}.$$

Since the algorithm sets $N_s = \lceil\max\{6(\hat{\rho}_{N_s}\tau_0^s)^{-1}, \sqrt{2}^s \cdot 3\sqrt{2}D_Y/(\hat{\rho}_{N_s} D_X\sqrt{\tau_0^s\sigma_0^s})\}\rceil$, it follows that

$$\frac{18}{(\hat{\rho}_{N_s}\tau_0^s N_s)^2} \leq \frac{18}{(\hat{\rho}_{N_s}\tau_0^s)^2} \cdot \frac{(\hat{\rho}_{N_s}\tau_0^s)^2}{36} = \frac{1}{2},$$

$$\frac{9D_Y^2}{\sigma_0^s\tau_0^s(\hat{\rho}_{N_s} N_s)^2} \leq \frac{9D_Y^2}{\sigma_0^s\tau_0^s\hat{\rho}_{N_s}^2} \cdot \frac{\hat{\rho}_{N_s}^2\sigma_0^s\tau_0^s D_X^2}{18D_Y^2 2^s} = \frac{1}{2}\cdot 2^{-s}D_X^2 = \frac{1}{2}\Delta_S,$$

which implies the desired result (15).

Let the algorithm run for $S = \lceil\log_2(D_X^2/\varepsilon)\rceil$ epochs, then $\|\mathbf{x}_0^S - \mathbf{x}^*\|^2 \leq D_X^2 \cdot 2^{-S} \leq \varepsilon$. The total iteration number required by Algorithm 2 for attaining a solution $\mathbf{x}_0^S$ such that $\|\mathbf{x}_0^S - \mathbf{x}^*\|^2 \leq \varepsilon$ is

$$\sum_{s=0}^S N_s \leq \sum_{s=0}^S \left\{\frac{6}{\hat{\rho}_{N_s}^s\tau_0^s} + \frac{3\sqrt{2}D_Y}{\hat{\rho}_{N_s}^s D_X\sqrt{\tau_0^s\sigma_0^s}}\sqrt{2}^s + 1\right\}$$

$$\overset{(a)}{=} \left(\frac{6}{\varpi_1\tau_0^s} + 1\right)(S + 1) + \frac{3\sqrt{2}D_Y}{\varpi_2 D_X\sqrt{\tau_0^s\sigma_0^s}}\sum_{s=0}^S\sqrt{2}^s$$

$$\leq \left(\frac{12}{\varpi_1\tau_0^s} + 2\right)\left\lceil\log_2\frac{D_X}{\sqrt{\varepsilon}} + 1\right\rceil + \frac{3\sqrt{2}D_Y}{\varpi_2 D_X\sqrt{\tau_0^s\sigma_0^s}} \cdot \frac{\sqrt{2}^{S+1}-1}{\sqrt{2}-1}$$

$$\leq \left(\frac{12}{\varpi_1\tau_0^s} + 2\right)\left\lceil\log_2\frac{D_X}{\sqrt{\varepsilon}} + 1\right\rceil + \frac{3\sqrt{2}D_Y(\sqrt{2}+1)}{\varpi_2 D_X\sqrt{\tau_0^s\sigma_0^s}} \cdot \left(\sqrt{2}^{\log_2(D_X^2/\varepsilon)+2} - 1\right)$$

$$\leq \left(\frac{12}{\varpi_1\tau_0^s} + 2\right)\left\lceil\log_2\frac{D_X}{\sqrt{\varepsilon}} + 1\right\rceil + \frac{6D_Y(\sqrt{2}+2)}{\varpi_2\sqrt{\tau_0^s\sigma_0^s}} \cdot \frac{1}{\sqrt{\varepsilon}},$$

where $(a)$ holds by $\sum_{s=0}^S(\hat{\rho}_{N_s}^s)^{-1} = (\varpi_1)^{-1}(S + 1)$ and $\sum_{s=0}^S\sqrt{2}^s/\hat{\rho}_{N_s}^s = (\varpi_2)^{-1}\sum_{s=0}^S\sqrt{2}^s$. $\square$

Now, we give some proof details in dual convergence results. Let

$$Q_j(\mathbf{x}, \mathbf{y}) := \frac{(\tau_j)^{-1}-\rho_j}{2}\|\mathbf{x} - \mathbf{x}_j\|^2 + \frac{1}{2\sigma_j}\|\mathbf{y} - \mathbf{y}_j\|^2 + (\sigma_{j-1}/\sigma_j)\langle\mathbf{y}_j - \mathbf{y}, G(\mathbf{x}_j) - G(\mathbf{x}_{j-1})\rangle$$

$$+ \frac{(\sigma_{j-1}/\sigma_j)}{2/\sigma_{j-1}}\|G(\mathbf{x}_j) - G(\mathbf{x}_{j-1})\|^2,$$

then we establish an important property about the solution sequence in the following lemma.

**Lemma 4.** *Assume $\bar{\tau}^{-1} > \bar{\rho}$ and choose $\nu_0 > 0$ such that*

$$1 > \inf_{j\geq 0}\{\sigma_{j-1}/\sigma_j\} \geq \delta + \nu_0. \tag{51}$$

*Then there exists an $\nu_1 > 0$ such that for any $j \geq 0$ and any KKT point $(\mathbf{x}^*, \tilde{\mathbf{y}}^*)$:*

$$0 \leq t_j Q_j(\mathbf{x}^*, \tilde{\mathbf{y}}^*) - t_{j+1}Q_{j+1}(\mathbf{x}^*, \tilde{\mathbf{y}}^*) - \nu_1 t_j\left[\frac{1}{2\tau_j}\|\mathbf{x}_{j+1} - \mathbf{x}_j\|^2 + \frac{1}{2\sigma_j}\|\mathbf{y}_{j+1} - \mathbf{y}_j\|^2\right],$$

$$0 < t_j Q_j(\mathbf{x}^*, \tilde{\mathbf{y}}^*).$$

*Proof.* First, we give some results that will be used repeatedly in the following. For notation simplicity, we denote $\theta_j = \sigma_{j-1}/\sigma_j$. In view of Lemma 3, and the parameter ergodic sequence generated by rAPDPro, we have $\{(\tau_k^s)^{-1}, \sigma_k^s\}$ is monotonically increasing sequence in $k$, $\bar{\tau} = \tau_0^s, \bar{\sigma} = \sigma_0^s, t_0^s = 1, \forall s \geq 0$, and there exist a $\nu_3 > 0$ such that $\bar{\sigma} + \nu_3 \leq \underline{\sigma} := \min_s\{\sigma_{N_s}^s\}$. Now, for rAPDPro, we claim that there exist $\nu_1, \nu_2 > 0$ such that the following two conditions hold

1. For any $j \geq 0$, we have

$$\min\left\{1 - \delta, (\tau_j^{-1} - L_{XY} - L_G^2\sigma_j)\tau_j\right\} \geq \nu_1 > 0, \tag{52}$$

and

$$t_j \min\left\{\tau_j^{-1} - \rho_j, \frac{1}{\sigma_j} - \frac{\delta}{\sigma_{j-1}}\right\} \geq \nu_2 > 0. \tag{53}$$

2. For any $j \geq 0$, we have

$$0 \leq t_j Q_j(\mathbf{x}^*, \tilde{\mathbf{y}}^*) - t_{j+1}Q_{j+1}(\mathbf{x}^*, \tilde{\mathbf{y}}^*) - \nu_1 t_j\left((2\tau_j)^{-1}\|\mathbf{x}_{j+1} - \mathbf{x}_j\|^2 + (2\sigma_j)^{-1}\|\mathbf{y}_{j+1} - \mathbf{y}_j\|^2\right). \tag{54}$$

**Part 1.** We first consider two subsequent points $\mathbf{x}_j$ and $\mathbf{x}_{j+1}$ within the same epoch, and assume $j \sim (s, k)$. Then, it follows from $\theta_k^s = \sigma_{k-1}^s/\sigma_k^s$ that

$$(\sigma_k^s)^{-1} - \theta_k^s\delta(\sigma_{k-1}^s)^{-1} = (\sigma_k^s)^{-1} - \delta(\sigma_k^s)^{-1} = \frac{1-\delta}{\sigma_k^s} \overset{(51)}{\geq} \frac{\nu_0}{\sigma_k^s}. \tag{55}$$

Next, we use induction to show

$$\frac{1-\nu_0}{\tau_k^s} \geq L_{XY} + L_G^2\sigma_k^s\delta^{-1}. \tag{56}$$

When $k = 0$, inequality (56) degenerates as the definition of $\tau_0^s, \sigma_0^s$. Suppose (56) holds for $k = 0, 1, \ldots, K - 1$. Then, from $\theta_K^s = \sigma_{K-1}^s/\sigma_K^s = \tau_K^s/\tau_{K-1}^s \leq 1$, we have

$$(1 - \nu_0)(\tau_K^s)^{-1} = (1 - \nu_0)(\tau_{K-1}^s\theta_K^s)^{-1} \geq \frac{L_{XY}}{\theta_K^s} + \frac{L_G^2\sigma_{K-1}^s\delta^{-1}}{\theta_K^s} \geq L_{XY} + L_G^2\sigma_K^s\delta^{-1},$$

which completes our induction proof. Hence, combining (55) and (56), we have

$$\min\left\{1 - \delta, \left((\tau_k^s)^{-1} - L_{XY} - L_G^2\sigma_k^s/\delta\right)\tau_k^s\right\} \geq \nu_0, \quad \forall k \in [N_s]. \tag{57}$$

Furthermore, when switching to the next epoch $(s \to s + 1)$, we have

$$\sigma_0^{s+1}((\sigma_0^{s+1})^{-1} - \theta_0^{s+1}\delta/\sigma_{N_s}^s) \overset{(a)}{\geq} \sigma_0^{s+1}((\sigma_0^{s+1})^{-1} - (\sigma_{N_s}^s)^{-1}) \overset{(b)}{\geq} 1 - \sigma_0^{s+1}\underline{\sigma}^{-1} = 1 - \bar{\sigma}\underline{\sigma}^{-1}$$

$$((\tau_0^{s+1})^{-1} - L_{XY} - L_G^2\delta^{-1}\sigma_0^{s+1})\tau_0^{s+1} \overset{(c)}{\geq} \nu_0\tau_0^{s+1} = \nu_0\bar{\tau}, \tag{58}$$

where $(a)$ holds by $\theta_0^s = 1$, $\delta < 1$, $(b)$ follows from $(\sigma_{N_s}^s)^{-1} \geq \underline{\sigma}^{-1}$. Hence, combining (55), (57) and (58), we completes our proof of (52) by setting $\nu_1 = \min\{1 - \bar{\sigma}\underline{\sigma}^{-1}, \nu_0\bar{\tau}, \nu_0\}$.

Since rAPDPro reset the stepsize periodically and $\{t_k^s, (\tau_k^s)^{-1}\}_{k\in[N_s]}$ are two monotonically increasing sequences, hence

$$\inf_{j\geq 0} t_j(\tau_j^{-1} - \rho_j) \geq t_0^s(\bar{\tau}^{-1} - \bar{\rho}) = \bar{\tau}^{-1} - \bar{\rho}. \tag{59}$$

Consider $\inf_{k\in[N_s]} t_k^s\sigma_k^s(1 - \delta\sigma_k^s/\sigma_{k-1}^s)$. Combining $\delta + \nu_0 \leq \inf_{k\in[N_s]}\{\theta_k^s\}$, then

$$\inf_{k\in[N_s]} t_k^s\sigma_k^s(1 - \delta\frac{\sigma_k^s}{\sigma_{k-1}^s}) = \inf_{k\in[N_s]} t_k^s\sigma_k^s(1 - \delta/\theta_k^s) \geq \nu_0\bar{\sigma}. \tag{60}$$

Furthermore, when switching to the next epoch $(s \to s + 1)$, we have

$$\inf_{s\geq 0} t_0^{s+1}\sigma_0^{s+1}(1 - \delta\sigma_0^{s+1}(\sigma_{N_s}^s)^{-1}) = \bar{\sigma}^2\inf_{s\geq 0}(\bar{\sigma}^{-1} - \delta(\sigma_{N_s}^s)^{-1}) \geq \bar{\sigma}(1 - \delta), \tag{61}$$

where the last inequality holds by $\bar{\sigma} = \sigma_0^s \leq \sigma_{N_s}^s$. Hence, it follows from (59), (60) and (61) that there exist $\nu_2 = \min\{\bar{\tau}^{-1} - \bar{\rho}, \nu_0\bar{\sigma}, \bar{\sigma}(1 - \delta)\}$ such (53) holds.

Part 2. for any $j \geq 0$, we have

$$t_{j+1}Q_{j+1}(\mathbf{x}^*, \tilde{\mathbf{y}}^*) \leq t_j \big( \tfrac{(\tau_j)^{-1}}{2} \|\mathbf{x}^* - \mathbf{x}_{j+1}\|^2 + \langle G(\mathbf{x}_{j+1}) - G(\mathbf{x}_j), \mathbf{y}_{j+1} - \tilde{\mathbf{y}}^* \rangle$$
$$+ (2\sigma_j)^{-1}\|\tilde{\mathbf{y}}^* - \mathbf{y}_{j+1}\|^2 + \tfrac{\sigma_j}{2}\|G(\mathbf{x}_{j+1}) - G(\mathbf{x}_j)\|^2 \big). \tag{62}$$

Consider $k \in \{0, 1, \ldots, N_s\}$. Inequality (51) implies (11) holds (see proof of Corollary 1 in Section E.3). Hence, for $0 \leq k \leq N_s$, we have

$$t_{j+1}Q_{j+1}(\mathbf{x}^*, \tilde{\mathbf{y}}^*) \leq t_k^s \big( \tfrac{(\tau_k^s)^{-1}}{2} \|\mathbf{x}^* - \mathbf{x}_{k+1}^s\|^2 + \langle G(\mathbf{x}_{k+1}^s) - G(\mathbf{x}_k^s), \mathbf{y}_{k+1}^s - \tilde{\mathbf{y}}^* \rangle$$
$$+ \tfrac{1}{2\sigma_k^s}\|\tilde{\mathbf{y}}^* - \mathbf{y}_{k+1}^s\|^2 + \tfrac{\sigma_k^s}{2\delta}\|G(\mathbf{x}_{k+1}^s) - G(\mathbf{x}_k^s)\|^2 \big) \tag{63}$$

where $j$ corresponds to $(s, k)$. Furthermore, consider switching to next epoch ($s \to s + 1$). Since $t_k^s(\tau_k^s)^{-1}$ is an increasing sequence in $k$, $\rho_0^{s+1} > 0, t_0^{s+1} = 1$, hence

$$t_{N_s}^s(\tau_{N_s}^s)^{-1} \geq t_0^{s+1}(\tau_0^{s+1})^{-1} - \rho_0^{s+1}t_0^{s+1}, \forall s \geq 0. \tag{64}$$

Next, we have

$$\tfrac{t_{N_s}^s}{\sigma_{N_s}^s} \overset{(a)}{=} \tfrac{t_0^{s+1}}{\sigma_0^{s+1}}, \ t_{N_s}^s \overset{(b)}{\geq} t_0^{s+1} \overset{(c)}{=} t_0^{s+1}\theta_0^{s+1}, t_{N_s}^s \sigma_{N_s}^s \overset{(b)}{\geq} t_0^{s+1}\sigma_0^{s+1} \overset{(c)}{=} t_0^{s+1}\sigma_0^{s+1}\theta_0^{s+1}, \tag{65}$$

where $(a)$ holds by the definition of $t_k^s = \tfrac{\sigma_k^s}{\sigma_0^s}$, $(b)$ holds by $\{t_k^s, \sigma_k^s\}$ is an increasing sequence in $k$, and $(c)$ holds by $\theta_0^{s+1} = 1$. Hence, by (64) and (65), we have

$$t_{j+1}Q_{j+1}(\mathbf{x}^*, \tilde{\mathbf{y}}^*) \leq t_{N_s}^s \big( \tfrac{1}{2\tau_{N_s}^s} \|\mathbf{x}^* - \mathbf{x}_0^{s+1}\|^2 + \tfrac{\sigma_{N_s}^s}{2}\|G(\mathbf{x}_0^{s+1}) - G(\mathbf{x}_{N_s}^s)\|^2$$
$$+ \tfrac{1}{2\sigma_{N_s}^s}\|\tilde{\mathbf{y}}^* - \mathbf{y}_0^{s+1}\|^2 + \langle G(\mathbf{x}_0^{s+1}) - G(\mathbf{x}_{N_s}^s), \mathbf{y}_0^{s+1} - \tilde{\mathbf{y}}^* \rangle \big) \tag{66}$$

where $j$ corresponds to $(s, N_s)$. By putting (63) and (66) together, we complete the proof of (62).

Placing $(\mathbf{x}, \mathbf{y}) = (\mathbf{x}^*, \tilde{\mathbf{y}}^*), (\mathbf{x}_{k+1}, \mathbf{y}_{k+1}) = (\mathbf{x}_{j+1}, \mathbf{y}_{j+1})$ in (32) and multiplying $t_j$ on both sides, we have

$$0 \leq t_j[\mathcal{L}(\mathbf{x}_{j+1}, \tilde{\mathbf{y}}^*) - \mathcal{L}(\mathbf{x}^*, \mathbf{y}_{j+1})]$$
$$\leq t_j \big[ \tfrac{\tau_j^{-1} - \rho_j}{2}\|\mathbf{x} - \mathbf{x}_j\|^2 - \tfrac{\tau_j^{-1}}{2}\|\mathbf{x} - \mathbf{x}_{j+1}\|^2 + \tfrac{\theta_j}{2\delta/\sigma_{j-1}}\|\mathbf{q}_j\|^2 - \tfrac{1}{2\delta/\sigma_j}\|\mathbf{q}_{j+1}\|^2$$
$$+ (2\sigma_j)^{-1}\big(\|\mathbf{y} - \mathbf{y}_j\|^2 - \|\mathbf{y} - \mathbf{y}_{j+1}\|^2\big) + \langle \mathbf{y} - \mathbf{y}_{j+1}, \mathbf{q}_{j+1} \rangle - \theta_j\langle \mathbf{y} - \mathbf{y}_j, \mathbf{q}_j \rangle$$
$$- \tfrac{\sigma_j^{-1} - \theta_j\delta/\sigma_{j-1}}{2}\|\mathbf{y}_{j+1} - \mathbf{y}_j\|^2 + \tfrac{L_G^2}{2\delta/\sigma_j}\|\mathbf{x}_{j+1} - \mathbf{x}_j\|^2 - \tfrac{\tau_j^{-1} - L_{XY}}{2}\|\mathbf{x}_{j+1} - \mathbf{x}_j\|^2 \big]$$
$$\leq t_jQ_j(\mathbf{x}^*, \tilde{\mathbf{y}}^*) - t_{j+1}Q_{j+1}(\mathbf{x}^*, \tilde{\mathbf{y}}^*) - \nu_1 t_j[(2\tau_j)^{-1}\|\mathbf{x}_{j+1} - \mathbf{x}_j\|^2 + (2\sigma_j)^{-1}\|\mathbf{y}_{j+1} - \mathbf{y}_j\|^2], \tag{67}$$

where the last inequality holds by (62) and (52). It follows from (53), $\sigma_{j-1}/\sigma_j \leq 1$ and

$$\langle \mathbf{y}_j - \tilde{\mathbf{y}}^*, \mathbf{q}_j \rangle \geq -\tfrac{\sigma_{j-1}}{2\delta}\|\mathbf{q}_j\|^2 - \tfrac{\delta/\sigma_{j-1}}{2}\|\tilde{\mathbf{y}}^* - \mathbf{y}_k\|^2$$

that

$$t_jQ_j(\mathbf{x}^*, \tilde{\mathbf{y}}^*) \geq t_j\big((2\tau_j)^{-1}\|\mathbf{x}^* - \mathbf{x}_j\|^2 + (2\sigma_j)^{-1}\|\tilde{\mathbf{y}}^* - \mathbf{y}_j\|^2 - \tfrac{\delta}{2\sigma_{j-1}}\|\mathbf{y}_j - \tilde{\mathbf{y}}^*\|^2\big)$$
$$\geq \nu_2\big(\tfrac{1}{2}\|\mathbf{x}^* - \mathbf{x}_j\|^2 + \tfrac{1}{2}\|\mathbf{y}_j - \tilde{\mathbf{y}}^*\|^2\big) > 0. \tag{68}$$

Combining (67) and (68), we complete our proof of (54). $\qquad \square$

### F.2 Proof of Theorem 3

*Proof.* Since $\{(\mathbf{x}_j, \mathbf{y}_j)\}$ located in set $\mathcal{X} \times \mathcal{Y}$ is a bounded sequence, it must have a convergent subsequence $\lim_{n \to \infty}(\mathbf{x}_{j_n}, \mathbf{y}_{j_n}) = (\mathbf{x}^*, \mathbf{y}^*)$, where $\mathbf{y}^*$ is the limit point. We claim that limit point $(\mathbf{x}^*, \mathbf{y}^*)$ satisfies the KKT condition. Placing $a_j = t_jQ_j(\mathbf{x}^*, \tilde{\mathbf{y}}^*), b_j = \nu_1 t_j[(2\tau_j)^{-1}\|\mathbf{x}_{j+1} - \mathbf{x}_j\|^2 + (2\sigma_j)^{-1}\|\mathbf{y}_{j+1} - \mathbf{y}_j\|^2]$ and $c_j = 0$ in Lemma 2. It follows from (54) in Lemma 4 that $a_j \geq 0, b_j > 0$. Hence, we have $\sum_{j=0}^{\infty} \|\mathbf{x}_{j+1} - \mathbf{x}_j\|^2 < \infty$, and $\sum_{j=0}^{\infty} \|\mathbf{y}_{j+1} - \mathbf{y}_j\|^2 < \infty$, which

implies $\lim_{n\to\infty} \|\mathbf{x}_{j_n} - \mathbf{x}_{j_n+1}\|^2 = 0$ and $\lim_{n\to\infty} \|\mathbf{y}_{j_n} - \mathbf{y}_{j_n+1}\|^2 = 0$. There are two different cases for $\tau_{j_n}$ when $j_n \to \infty$, and we discuss the value of $B_{j_n+1}$ in (25) decided by $\tau_{j_n}$ in each of the two cases below.

Case 1: $\tau_{j_n}^{-1} < \infty$. By the definition of $B_{j_n+1}$ in (27) and $\lim_{n\to\infty} \|\mathbf{x}_{j_n} - \mathbf{x}_{j_n+1}\|^2 = 0$, we have $B_{j_n+1} \le \|\mathbf{x} - \mathbf{x}_{j_n+1}\| \cdot \|\mathbf{x}_{j_n+1} - \mathbf{x}_{j_n}\| / \tau_{j_n} \overset{n\to\infty}{\longrightarrow} 0$.

Case 2: $\tau_{j_n}^{-1} = \infty$. It follows from (39) that $\tau_{j_n}^{-1}$ increases at order $\Theta(k)$, where $j_n \sim (s,k)$. By (23), we obtain $\|\mathbf{x} - \mathbf{x}_{j_n}\|$ decreases at order $\mathcal{O}(1/k)$ ($j_n \sim (s,k)$). Hence, combining $\lim_{n\to\infty} \|\mathbf{x}_{j_n} - \mathbf{x}_{j_n+1}\|^2 = 0$, we have $B_{j_n+1} \le \frac{1}{\tau_{j_n}}\big(\|\mathbf{x} - \mathbf{x}_{j_n+1}\|\|\mathbf{x}_{j_n+1} - \mathbf{x}_{j_n}\|\big) \overset{n\to\infty}{\longrightarrow} 0$. It follows from $\lim_{n\to\infty} \mathbf{x}_{j_n} = \mathbf{x}^*$, $\lim_{n\to\infty} B_{j_n+1} = 0$ and (25) that

$$f(\mathbf{x}^*) + \langle \nabla G(\mathbf{x}^*)\mathbf{y}^*, \mathbf{x}^* \rangle \le f(\mathbf{x}) + \langle \nabla G(\mathbf{x}^*)\mathbf{y}^*, \mathbf{x} \rangle, \forall \mathbf{x} \in \mathcal{X}.$$

Hence, according to the first-order optimality condition, we have

$$\mathbf{0} \in \partial f(\mathbf{x}^*) + \nabla G(\mathbf{x}^*)\mathbf{y}^* + \mathcal{N}_{\mathcal{X}}(\mathbf{x}^*). \tag{69}$$

Next, we show the complementary slackness holds for $(\mathbf{x}^*, \mathbf{y}^*)$. Since $\sigma_{j_n}^{-1}$ has an upper bound $\bar{\sigma}^{-1}$, $\|\mathbf{y} - \mathbf{y}_{j_n+1}\| \le D_Y$, $\lim_{n\to\infty} \|\mathbf{y}_{j_n} - \mathbf{y}_{j_n+1}\|^2 = 0$ and the definition of $A_{j_n+1}$ in (26), hence we obtain $A_{j_n+1} \le \frac{1}{\sigma_{j_n}}\big(\|\mathbf{y}_{j_n} - \mathbf{y}_{j_n+1}\|\|\mathbf{y} - \mathbf{y}_{j_n+1}\|\big) \overset{n\to\infty}{\longrightarrow} 0$. Combining above, $\lim_{n\to\infty} \mathbf{y}_{j_n} = \mathbf{y}^*$ and (24), we have $0 \le -\langle G(\mathbf{x}^*), \mathbf{y}^* \rangle \le -\langle G(\mathbf{x}^*), \mathbf{y} \rangle, \forall \mathbf{y} \in \mathcal{Y}$. Moreover, due to the complementary slackness, there exists an $\hat{\mathbf{y}}^* \in \mathcal{Y}^* \subseteq \mathcal{Y}$ such that $-\langle G(\mathbf{x}^*), \hat{\mathbf{y}}^* \rangle = 0$. Hence, we must have $\langle G(\mathbf{x}^*), \mathbf{y}^* \rangle = 0$, which, together with (69), implies that $(\mathbf{x}^*, \mathbf{y}^*)$ is KKT point. $\qquad\square$

## G Proof details for sparsity identification

Our proof strategy of active-set identification in rAPDPro is similar to those in unconstrained optimization [24]. Namely, we show that the optimal sparsity pattern is identified when the iterates fall in a properly defined neighborhood dependent on $\eta$. The next lemma shows that the primal and dual sequences indeed converge to the neighborhood of the optimal primal and dual solutions, respectively, in a finite number of iterations.

**Lemma 5.** *There exists an $\hat{S}_1$ such that*

$$\|\mathbf{x}_0^s - \mathbf{x}^*\| \le \|\mathbf{x}_0^{\hat{S}_1} - \mathbf{x}^*\| \ \text{ and } \ \|\mathbf{y}_0^s - \mathbf{y}^*\| \le \|\mathbf{y}_0^{\hat{S}_1} - \mathbf{y}^*\|, \forall s \ge \hat{S}_1, \tag{70}$$

*where $(\mathbf{x}^*, \mathbf{y}^*)$ is the unique solution of problem (17). Moreover, there exists an epoch $\hat{S}_0 \ge \hat{S}_1$ such that $\forall s \ge \hat{S}_0$, we have*

$$\|\mathbf{y}_k^s - \mathbf{y}^*\| \le \frac{\eta}{3\|\nabla g(\mathbf{x}^*)\|}, \ \|\mathbf{x}_k^s - \mathbf{x}^*\| \le \frac{\eta}{3L_{XY}} \frac{\tau_k^s}{\tau_k^s + (2L_{XY})^{-1}}, \ \forall k = 0, 1, \ldots N_s. \tag{71}$$

*Proof.* From Theorem 2 and 3, we have $\lim_{j\to\infty}(\mathbf{x}_j, \mathbf{y}_j) = (\mathbf{x}^*, \mathbf{y}^*)$, where $j$ corresponds to $(s,0)$. It implies that there exists an epoch $\hat{S}_1$ such that (70) holds.

It follows from (35) that $\|\mathbf{x}_1^s - \mathbf{x}^*\| \le \sqrt{\sigma_0^s \tau_0^s / \sigma_1^s (\|\mathbf{x}_0^s - \mathbf{x}^*\|^2 / \tau_0^s + \|\mathbf{y}_0^s - \mathbf{y}^*\|^2 / \sigma_0^s)}$. Hence, in order to prove $\|\mathbf{x}_1^s - \mathbf{x}^*\| \le \frac{\eta}{3L_{XY}} \cdot \frac{\tau_k^s}{\tau_k^s + (2L_{XY})^{-1}}$, we need to prove

$$\sqrt{\frac{\sigma_0^s \tau_0^s}{\sigma_1^s}\Big(\frac{1}{\tau_0^s}\|\mathbf{x}_0^s - \mathbf{x}^*\|^2 + \frac{1}{\sigma_0^s}\|\mathbf{y}_0^s - \mathbf{y}^*\|^2\Big)} \le \frac{\eta}{3L_{XY}} \frac{\tau_1^s}{\tau_1^s + (2L_{XY})^{-1}}. \tag{72}$$

From Corollary 1 and Theorem 2, 3, we know that the left hand side of (72) converges to $0$ and right hand side of (72) is a positive constant. Hence, there exist a $\hat{S}_2$ such that (72) holds, which implies (71) holds for $k = 1, s = \hat{S}_2$. Now we use induction to prove, for $\forall k \in [N_{\hat{S}_2}]$, we have

$$\Big(\frac{\sigma_0^{\hat{S}_2} \tau_k^{\hat{S}_2}}{\sigma_k^{\hat{S}_2}}\Big(\frac{1}{\tau_0^{\hat{S}_2}}\|\mathbf{x}_0^{\hat{S}_2} - \mathbf{x}^*\|^2 + \frac{1}{\sigma_0^{\hat{S}_2}}\|\mathbf{y}_0^{\hat{S}_2} - \mathbf{y}^*\|^2\Big)\Big)^{1/2} \le \frac{\eta}{3L_{XY}} \frac{\tau_k^{\hat{S}_2}}{\tau_k^{\hat{S}_2} + (2L_{XY})^{-1}}. \tag{73}$$

When $k = 1$, inequality (73) coincides with (72) with $s = \hat{S}_2$. Now, assume (73) holds for $k$, we aim to prove (73) holds for $k + 1$. It follows from (35) that

$$\|\mathbf{x}_{k+1}^{\hat{S}_2} - \mathbf{x}^*\| \overset{(a)}{\leq} \sqrt{\frac{\tau_{k+1}^{\hat{S}_2}}{\sigma_{k+1}^{\hat{S}_2}} \cdot \frac{\sigma_k^{\hat{S}_2}}{\tau_k^{\hat{S}_2}}} \cdot \frac{\eta}{3L_{XY}} \cdot \frac{\tau_k^{\hat{S}_2}}{\tau_k^{\hat{S}_2} + (2L_{XY})^{-1}} \overset{(b)}{=} \frac{\eta}{3L_{XY}} \cdot \frac{\tau_{k+1}^{\hat{S}_2}}{\tau_k^{\hat{S}_2} + (2L_{XY})^{-1}}$$

$$\overset{(c)}{\leq} \frac{\eta}{3L_{XY}} \cdot \frac{\tau_{k+1}^{\hat{S}_2}}{\tau_{k+1}^{\hat{S}_2} + (2L_{XY})^{-1}},$$

where $(a)$ follows from induction, $(b)$ holds by $\tau_k^{\hat{S}_2} \sigma_k^{\hat{S}_2} = \tau_{k+1}^{\hat{S}_2} \sigma_{k+1}^{\hat{S}_2}$ and $(c)$ holds by $\tau_{k+1}^{\hat{S}_2} \leq \tau_k^{\hat{S}_2}$. Hence, we complete our proof of (73). From Theorem 2, we have $\|\mathbf{x}_0^s - \mathbf{x}^*\|^2 \leq D_X^2 \cdot 2^{-s}$, which implies that there exists a $\hat{S}_3 = \left\lceil 2 \log_2 \left\{ D_X \left( \frac{\eta}{3L_{XY}} \cdot \frac{\bar{\tau}}{\bar{\tau} + (2L_{XY})^{-1}} \right)^{-1} \right\} \right\rceil$ such that $\|\mathbf{x}_0^{\hat{S}_3} - \mathbf{x}^*\| \leq D_X \cdot \sqrt{2}^{-\hat{S}_3} \leq \frac{\eta}{3L_{XY}} \frac{\bar{\tau}}{\bar{\tau} + (2L_{XY})^{-1}}$, which implies that $\|\mathbf{x}_0^s - \mathbf{x}^*\| \leq D_X^2 \cdot 2^{-s} \leq \frac{\eta}{3L_{XY}} \frac{\bar{\tau}}{\bar{\tau} + (2L_{XY})^{-1}}$ holds for any $s \geq \hat{S}_3$.

It follows from the definition of $\hat{S}_1$ in (70) and stepsize will be reset at different epoch, then we have (72) holds for $s \geq \max\{\hat{S}_1, \hat{S}_2\}$, which implies that (73) holds with substituting $\hat{S}_2$ as any $s \geq \max\{\hat{S}_1, \hat{S}_2\}$. Furthermore, it follows from Theorem 3 that $\lim_{j \to \infty} \mathbf{y}_j = \mathbf{y}^*$, where $j$ corresponds to $(s, k)$. Then there exists a $\hat{S}_4$ such that the first term in (71) holds. Hence, we can obtain that there exist a $\hat{S}_0 = \max\{\hat{S}_1, \hat{S}_2, \hat{S}_3, \hat{S}_4\}$ such that (71) holds. $\qquad \square$

It is worth noting that the primal neighborhood defined by the second term of (71) is a bit different from the fixed neighborhood in the standard analysis [24], which involves a constant stepsize. As APDPro sets $\tau_k^s = \mathcal{O}(1/k)$, both the point distance and neighborhood radius decay at the same $\mathcal{O}(1/k)$ rate. Hence, we use a substantially different analysis to show the sparsity identification in the constrained setting.

### G.1  Proof of Proposition 5

*Proof.* The uniqueness of primal optimal solution $\mathbf{x}^*$ follows from Proposition 2. The KKT condition (ensured by Slater's CQ) implies

$$\mathbf{0} \in \partial f(\mathbf{x}^*) + \nabla g(\mathbf{x}^*) \mathbf{y}^*. \tag{74}$$

According to Assumption 2, we have $\mathbf{x}^* \neq \mathbf{0}$, hence $\mathcal{A}^c(\mathbf{x}^*) = \{1, 2, \ldots, B\} \setminus \mathcal{A}(\mathbf{x}^*) \neq \emptyset$. In view of (74), for any $i \in \mathcal{A}^c(\mathbf{x})$, we have $p_i \mathbf{x}_{(i)}^* / \|\mathbf{x}_{(i)}^*\| = -\nabla_{(i)} g(\mathbf{x}^*) \mathbf{y}^*$, which gives a unique $\mathbf{y}^*$. $\qquad \square$

### G.2  Proof of Theorem 4

*Proof.* It follows from the Lipschitz smoothness of $g(\cdot)$ and property (71) that for any $s \geq \hat{S}_0$, we have

$$\begin{aligned}
\left\| \left[ \nabla g(\mathbf{x}_k^s) \mathbf{y}_{k+1}^s \right]_{(i)} \right\| &- \left\| \left[ \nabla g(\mathbf{x}^*) \mathbf{y}_{k+1}^s \right]_{(i)} \right\| \\
&\leq \left\| \nabla g(\mathbf{x}_k^s) \mathbf{y}_{k+1}^s - \nabla g(\mathbf{x}^*) \mathbf{y}_{k+1}^s \right\| \\
&\leq L_{XY} \left\| \mathbf{x}_k^s - \mathbf{x}^* \right\| \leq \frac{\eta}{3} \frac{\tau_k^s}{\tau_k^s + (2L_{XY})^{-1}}, \ k = 0, \ldots N_s.
\end{aligned} \tag{75}$$

Recall that the primal update has the following form

$$\mathbf{x}_{k+1}^s = \underset{\mathbf{x} \in \mathcal{X}}{\operatorname{argmin}} \left\{ \sum_{i=1}^B p_i \|\mathbf{x}_{(i)}\| + \langle \nabla g(\mathbf{x}_k^s) \mathbf{y}_{k+1}^s, \mathbf{x} \rangle + \frac{1}{2\tau_k^s} \|\mathbf{x} - \mathbf{x}_k^s\|^2 \right\}.$$

Since $\tau_k^s / (\tau_k^s + (2L_{XY})^{-1})$ is monotonically increasing with respect to $\tau_k^s$, for the strictly feasible point $\tilde{\mathbf{x}}$, we have

$$\begin{aligned}
\|\mathbf{x}_{k+1}^s - \tilde{\mathbf{x}}\| &\overset{(a)}{\leq} \frac{\eta}{3L_{XY}} \cdot \frac{\bar{\tau}}{\bar{\tau} + (2L_{XY})^{-1}} + \|\mathbf{x}^* - \tilde{\mathbf{x}}\| \\
&\overset{(b)}{<} \zeta + \min_{i \in [m]} 2 \sqrt{\frac{-2g_i(\mathbf{x}_i^*)}{\mu_i}},
\end{aligned} \tag{76}$$

where $(a)$ holds by (71), $\bar{\tau} \geq \tau_k^s$ and $(b)$ follows from the definition of $\mathbf{x}^*$, $\tilde{\mathbf{x}}$ and $\zeta$. Inequality (76) implies that $\mathbf{x}_{k+1}^s \in \mathbf{int}\,\mathcal{X}$, and hence $\mathcal{N}_{\mathcal{X}}(\mathbf{x}_{k+1}^s) = \{\mathbf{0}\}$. In view of the optimality condition, we have

$$\left[\tfrac{1}{\tau_k^s}(\mathbf{x}_k^s - \mathbf{x}_{k+1}^s) - \nabla g(\mathbf{x}_k^s)\mathbf{y}_{k+1}^s\right]_{(i)} \in p_i \partial \|[\mathbf{x}_{k+1}^s]_{(i)}\|,\ 1 \leq i \leq B. \tag{77}$$

Our next goal is to show $[\mathbf{x}_{k+1}^S]_{(i)} = \mathbf{x}_{(i)}^*$ satisfies condition (77) for $i \in \mathcal{A}(\mathbf{x}^*)$. Placing $\mathbf{x}_{(i)} = \mathbf{x}_{(i)}^*$ in $\left\|\left[\nabla g(\mathbf{x}_k^S)\mathbf{y}_{k+1}^S + \tfrac{1}{\tau_k^s}(\mathbf{x} - \mathbf{x}_k^s)\right]_{(i)}\right\|$, we have

$$\left\|\left[\nabla g(\mathbf{x}_k^s)\mathbf{y}_{k+1}^s + \tfrac{1}{\tau_k^s}(\mathbf{x}^* - \mathbf{x}_k^s)\right]_{(i)}\right\|$$

$$\leq \left\|\left[\nabla g(\mathbf{x}_k^s)\mathbf{y}_{k+1}^s\right]_{(i)}\right\| + \left\|\tfrac{1}{\tau_k^s}(\mathbf{x}_{(i)}^* - \mathbf{x}_{k(i)}^s)\right\|$$

$$\overset{(a)}{\leq} \tfrac{\eta}{3}\tfrac{\tau_k^s}{\tau_k^s + (2L_{XY})^{-1}} + \left\|\left[\nabla g(\mathbf{x}^*)\mathbf{y}_{k+1}^s\right]_{(i)}\right\| + \tfrac{\eta}{3}\tfrac{(L_{XY})^{-1}}{\tau_k^s + (2L_{XY})^{-1}} \tag{78}$$

$$\overset{(b)}{\leq} \tfrac{\eta}{3}\left[\tfrac{\tau_k^s + 2(2L_{XY})^{-1}}{\tau_k^s + (2L_{XY})^{-1}} + 1\right] + \left\|\left[\nabla g(\mathbf{x}^*)\mathbf{y}^*\right]_{(i)}\right\|$$

$$< \eta + \left\|\left[\nabla g(\mathbf{x}^*)\mathbf{y}^*\right]_{(i)}\right\| \overset{(c)}{\leq} p_i, \forall i \in \mathcal{A}(\mathbf{x}^*).$$

In above, $(a)$ follows from (71) and (75), $(b)$ follows from

$$\|[\nabla g(\mathbf{x}^*)\mathbf{y}_{k+1}^S]_{(i)}\| - \|[\nabla g(\mathbf{x}^*)\mathbf{y}^*]_{(i)}\| \leq \|\mathbf{y}_{k+1}^S - \mathbf{y}^*\| \|\nabla g(\mathbf{x}^*)\| \leq \tfrac{\eta}{3},$$

and $(c)$ holds by the definition of $\eta$. Combining (77) and (78), we have $\mathcal{A}(\mathbf{x}^*) \subseteq \mathcal{A}(\mathbf{x}_{k+1}^s), s \geq \hat{S}_0, \forall k \in [N_s]$, which completes our proof. $\qquad\square$

Table 1: Datasets description and parameter settings

| dataset | Node(n) | Edge | $b$ | $\alpha$ |
|---|---|---|---|---|
| bio-CE-HT | 2617 | 3K | -0.04 | 0.4 |
| bio-CE-LC | 1387 | 2K | -0.05 | 0.4 |
| econ-beaflw | 502 | 53K | -0.01 | 0.995 |
| DD68 | 775 | 2K | -0.005 | 0.4 |
| DD242 | 1284 | 3K | -0.05 | 0.4 |
| peking-1 | 3341 | 13.2K | -0.001 | 0.4 |

# H  A multi-stage accelerated primal-dual algorithm

Both the previous algorithms need to solve a complicated dual problem that involves a linear cut constraint, posing a potential issue: the associated sub-problem might lack a closed-form solution. To resolve this issue, we present the Multi-Stage Accelerated Primal-Dual Algorithm (msAPD) in Algorithm 3, which obtains the same $\mathcal{O}(1/\sqrt{\varepsilon})$ complexity without introducing a new cut constraint. Our new method is a double-loop procedure for which an accelerated primal-dual algorithm with a pending sub-iteration number (APDPi) is running in each stage. While both APDPi and APDPro employ the IMPROVE step to estimate the dual lower bound, APDPi only relies on the lower bound estimation to change the inner-loop iteration number adaptively, but not the stepsize selection.

We develop the convergence property of APDPi, which paves the path to proving our main theorem. For the convergence analysis, it suffices to verify that the initial stepsize parameter $\tau_0^s, \sigma_0^s$ satisfy assumptions in Theorem 5.

**Theorem 5.** *Let* $\{\bar{\mathbf{x}}_k^s, \bar{\mathbf{y}}_k^s\}$ *be the sequence generated by* APDPi, *then we have*

$$\mathcal{L}(\bar{\mathbf{x}}_K^s, \mathbf{y}^*) - \mathcal{L}(\mathbf{x}^*, \bar{\mathbf{y}}_K^s) \leq \tfrac{1}{K}\Delta^s(\mathbf{x}^*, \mathbf{y}^*),\ \ \tfrac{1}{2}\|\bar{\mathbf{x}}_K^s - \mathbf{x}^*\|^2 \leq \tfrac{1}{(\mathbf{y}^*)^\top \boldsymbol{\mu} K}\Delta^s(\mathbf{x}^*, \mathbf{y}^*), \tag{79}$$

*where* $\Delta^s(\mathbf{x}^*, \mathbf{y}^*) \triangleq \tfrac{1}{2\tau_0^s}\|\mathbf{x}_0^s - \mathbf{x}^*\|^2 + \tfrac{1}{2\sigma_0^s}\|\mathbf{y}_0^s - \mathbf{y}^*\|^2$ *and* $(\mathbf{x}^*, \mathbf{y}^*)$ *is a KKT point.*

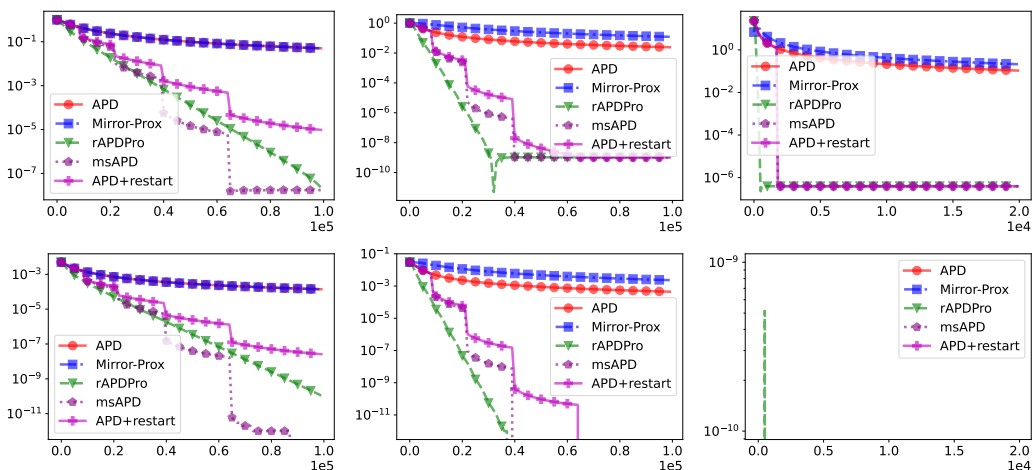

Figure 3: The first row is the results of objective convergence to optimum, where the $y$-axis reports $\log_{10}((\|D^{1/2}\mathbf{x}_k\|_1 - \|D^{1/2}\mathbf{x}^*\|_1)/\|D^{1/2}\mathbf{x}^*\|_1)$ for rAPDPro, and $\log_{10}((\|D^{1/2}\bar{\mathbf{x}}_k\|_1 - \|D^{1/2}\mathbf{x}^*\|_1)/\|D^{1/2}\mathbf{x}^*\|_1)$ for APD, msAPD and Mirror-Prox. The second row is the results of feasibility violation, where $y$-axis reports the feasibility gap $\log_{10}(\max\{0, G(\mathbf{x}_k)\})$ for rAPDPro, and $\log_{10}(\max\{0, G(\bar{\mathbf{x}}_k)\})$ for APD, APD+restart msAPD and Mirror-Prox. Datasets (Left-Right order) correspond to DD68, DD242 and peking-1.

---

**Algorithm 3** Multi-Stage APD (msAPD)

---

**Require:** $\bar{\mathbf{x}}^0 \in \mathcal{X}, \bar{\mathbf{y}}^0 \in \mathcal{Y}, \tilde{\sigma}, S$
1: **Initialize:** $\rho_0^0 = 0$
2: **for** $s = 0, \ldots, S$ **do**
3:     Compute $\tau_0^s = \left(L_{XY} + L_G^2 \tilde{\sigma} \cdot 2^{\frac{s}{2}}\right)^{-1}, \sigma_0^s = \tilde{\sigma} \cdot 2^{\frac{s}{2}}$
4:     $(\bar{\mathbf{x}}^{s+1}, \bar{\mathbf{y}}^{s+1}, \rho_0^{s+1}) \leftarrow \text{APDPI}(\tau_0^s, \sigma_0^s, \bar{\mathbf{x}}^s, \bar{\mathbf{y}}^s, \rho_0^s, s)$
5: **end for**
6: **Output:** $\bar{\mathbf{x}}^{S+1}, \bar{\mathbf{y}}^{S+1}$
7: **procedure** $\text{APDPI}(\tau_0^s, \sigma_0^s, \mathbf{x}_0, \mathbf{y}_0, \rho_0^s, s)$
8:     **Initialize:** $(\mathbf{x}_{-1}, \mathbf{y}_{-1}) \leftarrow (\mathbf{x}_0, \mathbf{y}_0), \bar{\mathbf{x}}_0 = \mathbf{x}_0, k = 0, N_s = \infty, \Delta_{XY} = \frac{1}{2\tau_0^s} D_X^2 + \frac{1}{2\sigma_0^s} D_Y^2$
9:     **while** $k < N_s$ **do**
10:         $\mathbf{z}_k \leftarrow 2G(\mathbf{x}_k) - G(\mathbf{x}_{k-1})$
11:         $\mathbf{y}_{k+1} \leftarrow \arg\min_{\mathbf{y} \in \mathcal{Y}} \|\mathbf{y} - (\mathbf{y}_k + \sigma_k \mathbf{z}_k)\|^2$
12:         $\mathbf{x}_{k+1} \leftarrow \text{prox}_{f,\mathcal{X}}(\mathbf{x}_k - \tau_0^s \nabla G(\mathbf{x}_k)\mathbf{y}_{k+1}, \tau_0^s)$
13:         $\bar{\mathbf{x}}_{k+1} \leftarrow (k\bar{\mathbf{x}}_k + \mathbf{x}_{k+1})/(k+1),$
14:         $\rho_{k+1}^s \leftarrow \text{IMPROVE}(\mathbf{x}_k, \bar{\mathbf{x}}_k, \frac{1}{2}D_X^2, \frac{\Delta_{XY}}{k}, \rho_k^s)$
15:         Compute $N_s = \lceil \max\{\frac{4}{\rho_{k+1}^s \tau_0^s}, \frac{D_Y^2}{\rho_{k+1}^s \sigma_0^s D_X^2} \cdot 2^{s+1}\}\rceil$
16:         $k \leftarrow k + 1$
17:     **end while**
18:     **return** $\bar{\mathbf{x}}_{N_s}, \bar{\mathbf{y}}_{N_s}, \rho_k^s$
19: **end procedure**

---

*Proof.* The stepsize $\tau_k^s = \tau_0^s, \sigma_k^s = \sigma_0^s$ are unchanged at one epoch, which implies that $\rho_{k+1} = 0$, i.e., (37) are satisfied. By the definition of $\tau_0^s$ and $\sigma_0^s$, we have $(\tau_0^s)^{-1} = L_{XY} + L_G^2 \tilde{\sigma}\sqrt{2}^s = L_{XY} + L_G^2 \sigma_0^s$, which means equality holds at the first term in (37).

Since $g_i(\mathbf{x})$ is a strongly convex function with modulus $\mu_i$, then we have

$$\mathcal{L}(\bar{\mathbf{x}}_K, \mathbf{y}^*) \geq \mathcal{L}(\mathbf{x}^*, \mathbf{y}^*) + \frac{(\mathbf{y}^*)^\top \boldsymbol{\mu}}{2}\|\bar{\mathbf{x}}_K - \mathbf{x}^*\|^2, \quad \mathcal{L}(\mathbf{x}^*, \mathbf{y}^*) \geq \mathcal{L}(\mathbf{x}^*, \bar{\mathbf{y}}_K).$$

Summing up the two inequalities above, we can get

$$\mathcal{L}(\bar{\mathbf{x}}_K, \mathbf{y}^*) - \mathcal{L}(\mathbf{x}^*, \bar{\mathbf{y}}_K) \geq \frac{(\mathbf{y}^*)^\top \boldsymbol{\mu}}{2}\|\bar{\mathbf{x}}_K - \mathbf{x}^*\|^2. \tag{80}$$

Combining (79) and (80), we can obtain the second term of (79). □

We show msAPD obtains an $\mathcal{O}(1/\sqrt{\varepsilon})$ convergence rate, which matches the complexity of APDPro.

**Theorem 6.** *Let $\{\bar{\mathbf{x}}_0^s\}$ be the sequence computed by* msAPD. *Then, we have*

$$\|\bar{\mathbf{x}}_0^s - \mathbf{x}^*\|^2 \le \Delta_s \equiv D_X^2 \cdot 2^{-s}, \quad \forall s \ge 0. \tag{81}$$

*For any $\varepsilon \in (0, D_X^2)$,* msAPD *will find a solution $\bar{\mathbf{x}}_0^s \in \mathcal{X}$ such that $\|\bar{\mathbf{x}}_0^s - \mathbf{x}^*\|^2 \le \varepsilon$ in at most $\lceil \log_2 D_X^2/\varepsilon \rceil$ epochs. Moreover, the overall iteration number performed by* msAPD *to find such a solution is bounded by*

$$T_\varepsilon = \left( \frac{8L_{XY}}{\rho_{N_0}^0} + 2 \right) \left\lceil \log_2 \frac{D_X}{\sqrt{\varepsilon}} + 1 \right\rceil + (2 + \sqrt{2}) \left( \tilde{\sigma} L_G^2 + \frac{2D_Y^2}{\rho_{N_0}^0 \tilde{\sigma} D_X^2} \right) \frac{D_X}{\sqrt{\varepsilon}}.$$

*Proof.* We first show that (81) holds by induction. It is easy to verify that (81) holds for $s = 0$. Assume $\|\bar{\mathbf{x}}_0^s - \mathbf{x}^*\|^2 \le \Delta_s = D_X^2 \cdot 2^{-s}$ holds for $s = 0, \ldots, S - 1$. By Theorem 5, we have

$$\|\bar{\mathbf{x}}_0^S - \mathbf{x}^*\|^2 \le \frac{1}{(\mathbf{y}^*)^\top \boldsymbol{\mu} N_{S-1}} \left( \frac{2}{\tau_0^{S-1}} \Delta_S + \frac{1}{\sigma_0^{S-1}} D_Y^2 \right).$$

As the algorithm sets $N_{S-1} = \lceil \max \{ 4/(\rho_{N_{S-1}}^{S-1} \tau_0^{S-1}), 2D_Y^2/(\rho_{N_{S-1}}^{S-1} \sigma_0^{S-1} \Delta_S) \} \rceil$, the following inequalities hold:

$$2\big((\mathbf{y}^*)^\top \boldsymbol{\mu} N_{S-1} \tau_0^{S-1}\big)^{-1} \le 2\big(\rho_{N_{S-1}}^{S-1} N_{S-1} \tau_0^{S-1}\big)^{-1} \le \tfrac{1}{2},$$

$$D_Y^2 \big((\mathbf{y}^*)^\top \boldsymbol{\mu} N_{S-1} \sigma_0^{S-1}\big)^{-1} \le D_Y^2 \big(\rho_{N_{S-1}}^{S-1} N_{S-1} \sigma_0^{S-1}\big)^{-1} \le \tfrac{1}{2} \Delta_S.$$

Putting these pieces together, we have $\|\bar{\mathbf{x}}_S - \mathbf{x}^*\|^2 \le \tfrac{1}{2}\Delta_S + \tfrac{1}{2}\Delta_S = \Delta_S$. Suppose the algorithm runs for $S$ epochs to achieve the desired accuracy $\varepsilon$, i.e., $\|\mathbf{x}_0^S - \mathbf{x}^*\|^2 \le D_X^2 \cdot 2^{-S} \le \varepsilon$. Then the overall iteration number can be bounded by

$$\sum_{s=0}^S N_s \overset{(a)}{\le} \sum_{s=0}^S \left\{ \frac{4}{\rho_{N_0}^0 \tau_0^{S-1}} + \frac{2D_Y^2}{\rho_{N_0}^0 \sigma_0^{S-1} \Delta_S} + 1 \right\}$$

$$\overset{(b)}{\le} \sum_{s=0}^S \left\{ \left( \frac{4L_{XY}}{\rho_{N_0}^0} + 1 \right) + \left( \tilde{\sigma} L_G^2 + \frac{2D_Y^2}{\rho_{N_0}^0 \tilde{\sigma} D_X^2} \right) \sqrt{2}^s \right\}$$

$$\le \left( \frac{8L_{XY}}{\rho_{N_0}^0} + 2 \right) \left\lceil \log_2 \frac{D_X}{\sqrt{\varepsilon}} + 1 \right\rceil + (2 + \sqrt{2}) \left( \tilde{\sigma} L_G^2 + \frac{2D_Y^2}{\rho_{N_0}^0 \tilde{\sigma} D_X^2} \right) \frac{D_X}{\sqrt{\varepsilon}},$$

where $(a)$ holds by $\rho_{N_S}^s \ge \rho_{N_0}^0, \forall s \ge 0$, $(b)$ follows from the definition of $\tau_0^s$ and $\sigma_0^s$. □

**Remark 11.** *Theorem 6 shows that* msAPD *obtains a worst-case complexity of $\mathcal{O}\big( \log(D_X/\sqrt{\varepsilon}) + (D_X + D_Y^2/D_X)/\sqrt{\varepsilon} \big)$, which is an upper bound of the complexity of* rAPDPro *(see Theorem 2). The complexities of* msAPD *and* rAPDPro *match when $D_X = \Omega(1)D_Y$. Otherwise,* rAPDPro *appears to be much better in terms of dependence on $D_X/\sqrt{\varepsilon}$. On the other hand,* msAPD *has a simpler subproblem, which does not involve an additional cut constraint on the dual update.*

## I    Experiment details

We examine the empirical performance for solving sparse Personalized PageRank. Let $G = (V, E)$ be a connected undirected graph with $n$ vertices. Denote the adjacency matrix of $G$ by $A$, that is, $A_{i,j} = 1$ if $i \sim j$ and 0 otherwise. Let $D = \text{diag}(d_1, \ldots, d_n)$ be the matrix with the degrees $\{d_i\}_{i=1}^n$ in its diagonal. Then the constrained form of Personalized PageRank can be written as follows:

$$\min_{\mathbf{x} \in \mathbb{R}^n} \|D^{1/2}\mathbf{x}\|_1 \text{ s.t. } \tfrac{1}{2} \langle \mathbf{x}, Q\mathbf{x} \rangle - \alpha \langle \mathbf{s}, D^{-1/2}\mathbf{x} \rangle \le b, \tag{82}$$

where $Q = D^{-1/2}\big(D - \frac{1-\alpha}{2}(D + A)\big)D^{-1/2}$, $\alpha \in (0, 1)$, $\mathbf{s} \in \Delta^n$ is a teleportation distribution over the nodes of the graph $G$ and $b$ is a pre-specific target level.

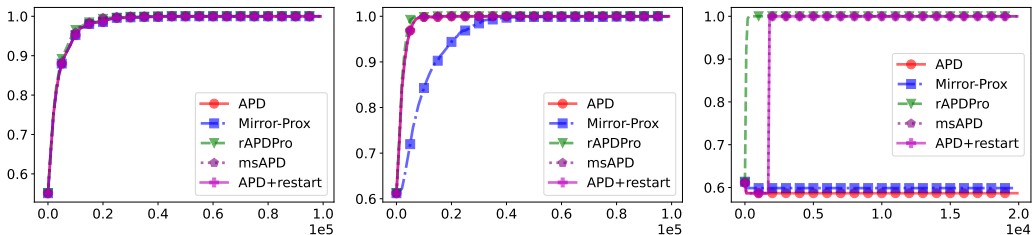

Figure 4: The experimental results on active-set identification. Datasets (Left-Right order) correspond to DD68, DD242 and peking-1. The $x$-axis reports the iteration number and the $y$-axis reports accuracy in active-set identification.

**Datasets** We selected 6 small-to-median scale datasets from various domains in the Network Datasets [28]. We skip large-scale networks as MOSEK struggles to achieve the optimal solution, making it unsuitable for subsequent comparison of the optimality gap. We briefly describe these datasets in Table 1. For more details, please refer to the network repository.

**Parameter tuning** For all experiments, we set $r = \min_{i \in [n]} |d_i|$, $\underline{\mu} = \lambda_{\min}(Q)$ and $L_X = \lambda_{\max}(Q)$, with $\lambda_{\min}(\cdot), \lambda_{\max}(\cdot)$ denoting the smallest and largest eigenvalue, respectively. For msAPD, we have made additional parameter adjustments. Based on our observations, due to a small estimated strongly convex coefficient, msAPD could not switch to the next cycle $s$ early enough. To prevent msAPD from degrading to APD, we iterate according to the predefined number of sub-iterations and manually switch to the next set of parameters. We divide $\tau$ by $\sqrt{2}$, multiply $\sigma$ by $\sqrt{2}$, and increase the number of sub-iterations in the next period by a factor of $\sqrt{2}$. For all experiments, we tune the stepsize $\tau, \sigma, \gamma$ from $\{0.0001, 0.0005, 0.001, 0.005, 0.01\}$, where $\tau, \sigma$ are the initial stepsizes of rAPDPro, msAPD and APD, $\gamma$ is the constant stepsize of Mirror-Prox. All algorithms start with the primal variables initialized as zero vectors and the dual variables initialized as ones.

**Additional experiment results** Figure 3 and Figure 4 describe the convergence performance and active set identification results on the last three datasets: DD68, DD242 and peking-1. Furthermore, we report the time consumption for the Personalized PageRank problem in Table 2. The table indicates that, although rAPDPro and msAPD require moderately complex computations to determine the lower bound of the strong convexity parameter, the two methods still accelerate the algorithm's convergence and can significantly reduce the overall convergence time.

Table 2: Time summary when $\max\{|f(\mathbf{x}) - f(\mathbf{x}^*)|/|f(\mathbf{x}^*)|, \max\{G(\mathbf{x}), 0\}\} \leq 10^{-3}$. All experiments were conducted five times, and the results are reported as mean (standard deviation). $*$ means that upon completion of all iterations, the algorithms still fails to meet the criteria for both error measures.

| dataset | APD | APD+restart | rAPDPro | Mirror-Prox | msAPD | mosek |
|---------|-----|-------------|---------|-------------|-------|-------|
| bio-CE-HT | 187.15 (0.86)* | 115.95 (1.04) | 136.92 (0.92) | 370.50 (1.80)* | **77.21** (0.67) | 0.21 |
| bio-CE-LC | 2.58 (0.16)* | 0.65 (0.01) | **0.44** (0.01) | 4.74 (0.33)* | 0.65 (0.03) | 0.1 |
| econ-beaflw | 72.28 (0.59)* | 87.12 (0.43)* | **18.42** (0.44) | 116.13 (1.15)* | 66.70 (0.76) | 0.16 |
| DD242 | 43.29 (1.20)* | 10.27 (0.39) | **6.30** (0.08) | 79.16 (0.60)* | 10.33 (0.62) | 0.16 |
| DD68 | 36.55 (0.42)* | 19.07 (0.66) | 22.35 (0.75) | 67.73 (1.39)* | **15.69** (0.37) | 0.24 |
| peking-1 | 122.37 (2.99)* | 11.55 (0.69) | **4.86** (0.09) | 243.45 (7.20)* | 11.24 (0.15) | 0.21 |

Nonetheless, we observe that Mosek achieves significantly faster computational efficiency for small-scale problems than our algorithm. Therefore, we test the efficiency of rAPDPro on some large-scale instances. For large-scale instances, we consider the following problem $\min_{\mathbf{x} \in \mathbb{R}^n} \|\mathbf{x} - \mathbf{1}\|_1$ s.t. $\frac{1}{2}\mathbf{x}^\top Q_i \mathbf{x} + c_i^\top \mathbf{x} + d_i \leq 0, i = 1, \ldots, m$, where $Q_i$ are dense and positive definite matrix and generated randomly and $c_i$ are generated randomly. Furthermore, we set proper $d_i$ to make the feasible region is non-empty. When $n = 5000$ and $m > 10$, MOSEK crashes on our computer, which means we can not get $\mathbf{x}^*$ for calculating the optimality gap. Therefore, we report the time required

for the algorithm to satisfy $\max\{|f(\mathbf{x}) - f(\mathbf{x}^*)|/|f(\mathbf{x}^*)|, \max\{G(\mathbf{x}), 0\}\} \leq 10^{-3}$ and the time taken by the algorithm to complete 10,000 iterations. On this problem, results from small datasets indicate that the performance of the 10,000-step algorithm should be sufficient to meet our specified termination criteria.

Table 3: Comparison of computational time in seconds between rAPDPro and MOSEK

| m | rAPDPro | MOSEK |
|---|---------|-------|
| 8 | 24.612 | 50.38 |
| 10 | 53.997 | 67.99 |
| 12 | 392 | - |

