# OpenReview forum: "Faster Accelerated First-order Methods for Convex Optimization with Strongly Convex Function Constraints"
_NeurIPS.cc/2024/Conference — NeurIPS 2024 poster_

### Official Review · Reviewer_h3VH · 2024-06-27

**Soundness:** 3
**Presentation:** 2
**Contribution:** 2
**Rating:** 5
**Confidence:** 5

**Summary:**

The authors introduce faster accelerated primal-dual algorithms for minimizing a convex function subject to strongly convex function constraints.

**Strengths:**

1. The authors address the theoretical questions about strongly convex-constrained optimization and the application of sparse optimization.

2. The authors present a new accelerated primal-dual algorithm with progressive strong convexity estimation (APDPro) for solving the problem (1).

3. The authors present a new restart algorithm (rAPDPro) which calls APDPro repeatedly with the input parameters properly changing over time.

**Weaknesses:**

1. What is the 2-norm (see line 116)? what is the meaning of $\bot$?

2. The conditions of Theorem 1 are too strict and limited, which hinders the application of APDPro. There are also similar issues with the corollary 1.

3. As illustrated by formulation (2), when $f$ is convex and $g_i$ is strongly convex, $L(\cdot, y)$ is strongly convex and the convergence rate of gradient descent is linear, which is a well-known conclusion in optimization. However, the convergence rate of your method is sublinear [1, 2].

4. This paper lacks related work. There are several studies exploring optimization with the objective of minimizing a convex function subject to strongly convex function constraints.

5. The improvement in the convergence rate comes at the cost of increased computational effort per step.

6. Convex optimization with strongly convex function constraints has been explored by [1, 2], but the authors don’t cite these references. Moreover, Nesterov's Accelerated Gradient Method [3] can also achieve a complexity of $1/\sqrt{\epsilon}$. The authors did not improve the convergence rate compared to previous works.


[1] Jorge Nocedal, Stephen J. Wright: Numerical Optimization. Springer 1999, ISBN 978-0-387-98793-4, pp. 1-634

[2] Yurii E. Nesterov: Introductory Lectures on Convex Optimization - A Basic Course. Applied Optimization 87, Springer 2004, ISBN 978-1-4613-4691-3, pp. 1-236

[3] A Differential Equation for Modeling Nesterov's Accelerated Gradient Method: Theory and Insights. J. Mach. Learn. Res. 17: 153:1-153:43 (2016)

**Questions:**

See Weaknesses

---

> ### Author Rebuttal · Authors · 2024-08-05
>
> Thanks for the careful reading and valuable suggestions!
>
> **Q1:What is $l _2$ norm and $\perp$?** We apologize for any confusion caused by the lack of clear definitions. $l _2$ norm for a vector x is defined as $(\sum _{i=1}^{n}|x _{(i)}|^2)^{1/2}$, where $x _{(i)}$ is the $i$-th element of x. The definition of the $l _2$ norm in the paper is consistent with that of the $l _q$ norm when $q=2$. "$0\le y^*\perp -G(x^*)\ge 0$" means $y _{(i)}^*\ge 0,g _i(x^*)\le 0$ and $\sum _{i=1}^{m}y _{(i)}^* g _i(x^*) = 0$.
>
> **Q2:The conditions of Thm1 and Cor1 are too limited.** We slightly disagree with your comments. In Thm 1, we demonstrate that the algorithm can converge if certain parameter relationships are satisfied. In Cor 1, we provide a specific parameter setting and verify that these settings satisfy the relationships in Thm 1. For example, in the parameter settings outlined in Cor 1, all parameter relationships are derived recursively. The key requirement is that the initial step size, $\tau _0$ and $\sigma _0$, satisfy $\tau _0^{-1}\ge L _{XY}+L _G^2\sigma _0/\delta$. Given specified $\delta$, like $\delta=1$, this relationship can be satisfied if $\tau _0$ is chosen to be sufficiently small. We apologize that our original writing may have given the impression that the parameter settings are overly complex. Following the suggestions of Reviewer 8Vg8, we will rewrite these conditions to enhance the readability.
>
> **Q3:$L(\cdot,y)$ is strongly convex and the convergence rate is linear. However, your result is sublinear.**
> We respectfully disagree that the sublinear convergence rate is bad. When y is strictly greater than 0, $L(\cdot,y)$ is strongly convex w.r.t. x. In this context, if y is fixed, the problem becomes an unconstrained strongly convex optimization. The best convergence rate is indeed linear. However, when a constrained optimization problem is expressed in the form of a Lagrangian function, it essentially becomes a minimax problem. Under certain constraint qualifications, like Slater's condition, the feasible region for the dual variable is bounded. Consequently, the lower bound on the convergence rate for first-order algorithms is sublinear [5]. Furthermore, we agree that the algorithm proposed in [2, Section 2.3.5] has achieved a linear convergence rate, which we guess is the algorithm referred to by the reviewer. Note that this algorithm requires both f and g to be strongly convex while ours only need strong convexity of g. More seriously, in each iteration, the algorithm needs to solve a much more difficult quadratic program with quadratic inequality constraints.
> $$
> f(x)+\langle f'(\bar{x}),x-\bar{x} \rangle + \frac{\mu}{2}\\|x-\bar{x}\\|^2 s.t. \ g _i(\bar{x})+\langle g _i'(\bar{x}),x-\bar{x} \rangle + \frac{\mu}{2}\\|x-\bar{x}\\|^2 \le 0.
> $$
> The complexity is particularly emphasized at the end of Section 3.5, on page 110 in [2]. Hence, comparing with this algorithm in terms of iteration complexity doesn't seem fair.
>
> **Q4:lacks related work.** We acknowledge that we had previously overlooked relevant literature on achieving linear convergence using a stronger oracle. Our literature review primarily focused on comparisons with first-order algorithms. We observed that most algorithms, such as [4] and [6], assume the objective function to be strongly convex. For the case where the objective function is convex and the constraints are strongly convex, a class of Frank-Wolfe algorithms has been studied. Due to space limitations, we have included this discussion in Appendix B.
>
> **Q5: The improvement comes at the cost of increased computational effort per step.**
> We agree with the reviewer on the additional computational efforts, however, those additional costs are still manageable.
> - The "improve procedure" to estimate the strong convexity computes the norm of the Jacobian matrix, which can be efficiently solved using the Power method or Lanczos method. Additionally, we can significantly reduce computation costs by using a warm start from previous iterates.
> - The dual update of APDPro involves a linear inequality constraint and nonnegative constraints.  Note that this subproblem is easy to solve in our sparsity-constrained problem, one can obtain a closed-form solution. In general, one can enumerate the active constraint and obtain the closed-form solution in each case. This is highly efficient and can be done in parallel when the constraint number is small (say, $\le 5$, refer to Appendix in [7]). Due to the space limit, we don’t plan to dive further into the details, but if the reviewer is interested, we are glad to explain further. To address more general cases, we developed the msAPD, which leverages the lower bound of the strong convexity to check the stopping condition for the inner loop, thus avoiding complicated dual updates. We believe the msAPD mitigates this issue effectively.
>
> **Q6: Lack references[2],[1] and NAG can also achieve $1/\sqrt{\epsilon}$. The authors did not improve rate**
> We respectfully disagree with your comments that our results are not novel. The previous result [3] is focused on unconstrained optimization. We think the comparison is unfair. Regarding the absence of citations to relevant literature, please see our responses to Q3 and Q4.
>
> [4] Yangyang Xu. Iteration complexity of inexact augmented lagrangian methods for constrained convex pro-
> gramming. Mathematical Programming, 2021.
>
> [5] Yuyuan Ouyang and Yangyang Xu. Lower complexity bounds of first-order methods for convex-concave
> bilinear saddle-point problems. Mathematical Programming, 2021.
>
> [6]Erfan Yazdandoost Hamedani and Necdet Serhat Aybat. A primal-dual algorithm with line search for
> general convex-concave saddle point problems. SIAM Journal on Optimization, 2021.
>
> [7]Yunmei Chen, Guanghui Lan, Yuyuan Ouyang, and Wei Zhang. Fast bundle-level methods for unconstrained
> and ball-constrained convex optimization. Computational Optimization and Applications, 2019.

---

> ### Comment · Reviewer_h3VH · 2024-08-09
> **Reply to the Authors**
>
> While I am partially satisfied with the author's response, I still have a questions:
>
> According to Figure 1, the algorithm presented by the authors exhibits a linear or superlinear convergence rate, which is faster than the sublinear convergence claimed in the paper. Why do the experimental results significantly exceed the theoretical predictions?
>
> Furthermore, numerous datasets exist for convex and strongly convex functions. Conducting additional experiments on well-known function sets would be beneficial in validating the effectiveness of the proposed theories.
>
> If the authors make a satisfactory response, I may consider raising my score.

---

> > ### Author Response · Authors · 2024-08-11
> > **Additional experiments**
> >
> > **Linear convergence in Figure 1?**
> > Your comments are insightful, and we believe we now understand your suggestions better. After examining the dual sequence, we observed that within the first thousand iterations, the constraint violation is small and the dual variables had already converged to the optimal value. Although we only establish an asymptotic convergence result for the dual variables, in these specific cases, the convergence was indeed notably rapid. However, we note that such cases do not always happen. As shown in the next additional experiment, our algorithms only achieve sublinear convergences to an accuracy of $O(10^{-3})$. In such a case, due to low feasibility accuracy, the dual is steadily increasing and converges slower than in the previous test case. This phenomenon raises interesting questions about when and how to identify the phase of linear convergence, which we would like to explore in future work.
> >
> > **Conducting additional experiments?**
> > Thanks for your advice, we are considering the following problem additionally. We use the dataset from [8] to test our algorithm.
> >
> > $$min \\|w\\|_1 \text{s.t.}\ \  \frac{1}{n}\sum _{i=1}^{n} \log(1+\exp(y_i \cdot x _i ^\top w)) + \frac{1}{2}\\|w\\|^2 \leq 1,$$
> >
> > where $x_i$ is feature and $y_i\in\\{-1,1\\}$. Use `max{optimality gap, feasibility gap}` $\leq 10^{-3}$ as terminal criteria, then we summarize the iteration number needed as follows:
> >
> > | method | rapdpro | msapd | apd | adp+restart | mirror-prox |
> > | --- | --- | --- | --- | --- | --- |
> > | iterations | 29100 | 19700 | 50000 | 32300 | 50000 |
> >
> > This problem is more challenging compared to the previous one, as it requires a much greater number of iterations. Nevertheless, our first-order algorithm still demonstrates a substantial advantage over other algorithms.
> > In addition, following Reviewer ecU7's suggestions, we conducted some large-scale experiments to further demonstrate the advantages of our algorithms.
> >
> > [8] Guyon, Isabelle, Gunn, Steve, Ben-Hur, Asa, and D. Gideon. Arcene. UCI Machine Learning Repository, 2008.
> > DOI: https://doi.org/10.24432/C58P55.

---

> > > ### Comment · Reviewer_h3VH · 2024-08-11
> > > **Raising Score**
> > >
> > > Thanks to the authors for their response. The authors have partially addressed our concerns. In recognition of their efforts, I plan to increase the score to 5.

---

### Official Review · Reviewer_ecU7 · 2024-07-13

**Soundness:** 3
**Presentation:** 3
**Contribution:** 3
**Rating:** 7
**Confidence:** 3

**Summary:**

This method proposes new acceleration methods to solve the convex optimization problem with convex constraints. To do this, the authors propose to iteratively improve the lower bounds of the strong convexity parameter of the associated Lagrangian. In turn, this lower bound is used to create cutting planes for the domain of the dual variables. By using this new technique, the authors is able to improve the running time complexity of the base method APD from $O(1/k)$ to $O(1/k^2)$, where $k$ is the number of iterations. The authors verify the claim of running time improve empirically through several datasets.

**Strengths:**

The core techniques (iteratively improve the lower bound of the strong convexity parameter and use it to add cutting planes to the domain of the dual variables) seem to be novel. The improvement on computational complexity is nontrivial and matches the lower bound for this class.

The writing is clear in terms of highlighting the high-level ideas and emphasizing which part is the authors' contribution.

The code is also available as part of the supplementary materials.

**Weaknesses:**

1. For Figure 2 on the active-set identification experiment, I am not seeing a big difference between APD (baseline) and rAPDPro (proposed method). So I am guessing the core empirical contribution is on improving the curve of optimality gap vs number of iterations. It would be nice to show the curve of optimality gap vs wall-clock running time.

**Questions:**

1. For the experimental section, you only show optimality gap vs number of iterations. Could you also report optimality gap vs wall-clock running time? You can report these results in table format during rebuttal.

2. For the baseline method APD, is restart also applied? For fair comparison, I think you should compare rAPDPro with APD + restart.

3. The authors obtain the optimal solutions via MOSEK. Could you report the optimality gap of the MOSEK solver together with what are shown in Figure 1?

4. For Figure 3 and Figure 4 in the appendix, can you explain why the baselines (mirror-P and APD) fail and perform so poorly on the rightmost plot?

**Limitations:**

The authors have already pointed out the limitations in this submission:

1. The proposed method requires knowing an upper bound on $\lVert \mathbf{y}^* \rVert$.

2. If the evaluation of the proximal operator of $f$ is inexpensive, this method can be rightly applied; if not, this method will incur additional computational cost.

---

> ### Author Rebuttal · Authors · 2024-08-05
>
> We greatly appreciate your comments. We provide the following clarifications to address your concerns. We include Table1 summarizing the time required for the optimal gap and infeasibility to decrease to $10^{-3}$. Additionally, as per your suggestion, we will compare our results with APD+restart. As shown in the table, comparing from the perspective of time, we observe that rAPDPro and msAPD demonstrate faster convergence speeds across different datasets. Regarding your inquiry about the relationship between MOSEK iterations and the optimality gap, since MOSEK employs a second-order algorithm, its computational cost per iteration differs significantly from the first-order method. Therefore, comparing them to the same figure may not be fair. The poor performance of Mirror-Prox and APD in Figure 3 and 4 is primarily due to their theoretical convergence being based on the average sequence. If the initial point is bad, it can adversely affect convergence and active set identification. Finally, we agree with your observations regarding the limitations of our paper, which we have also mentioned in the paper, and these will be the focus of our future direction.
>
>
> Table1: Time summary when max{optimality gap, feasibility gap} ≤ $10^{−3}$(* means that upon
> completion of all iterations, the algorithm still fails to meet the criteria for both error measures.)
> | Dataset     | APD                 | APD+restart       | rAPDPro          | Mirror-Prox        | msAPD            |
> |-------------|---------------------|-------------------|------------------|--------------------|------------------|
> | bio-CE-HT   | 187.15 (0.86)*      | 115.95 (1.04)     | 136.92 (0.92)    | 370.50 (1.80)*     | **77.21** (0.67) |
> | bio-CE-LC   | 2.58 (0.16)*        | 0.65 (0.01)       | **0.44** (0.01)  | 4.74 (0.33)*       | 0.65 (0.03)      |
> | econ-beaflw | 72.28 (0.59)*       | 87.12 (0.43)*     | **18.42** (0.44) | 116.13 (1.15)*     | 66.70 (0.76)     |
> | DD242       | 43.29 (1.20)*       | 10.27 (0.39)      | **6.30** (0.08)  | 79.16 (0.60)*      | 10.33 (0.62)     |
> | DD68        | 36.55 (0.42)*       | 19.07 (0.66)      | 22.35 (0.75)     | 67.73 (1.39)*      | **15.69** (0.37) |
> | peking-1    | 122.37 (2.99)*      | 11.55 (0.69)      | **4.86** (0.09)  | 243.45 (7.20)*     | 11.24 (0.15)     |

---

> ### Comment · Reviewer_ecU7 · 2024-08-09
>
> I thank the authors for the response.
>
> 1. I still want to insist on receiving the results from MOSEK. I know MOSEK uses the second-order method. I know the difference between the first- and second-order methods. I will keep the difference in mind when I evaluate this submission. Please report the optimality gap vs wall-clock running time of MOSEK.
>
> 2. To continue, if there is a chance MOSEK could beat you, maybe you could make a larger-scale dataset to show your algorithm is more scalable while MOSEK would run for a very very long time. After all, this is where the first-order method beats the second-order methods, in terms of handling larger datasets and/or more constraints.
>
> 3. Maybe I didn't describe it clearly enough when I said optimality vs. wall-clock running time. I was asking for Figure 1 but instead of number of iterations in the x-axis, use wall-clock running time. Could you report the results in table formats I just described again? You can report the results in three metrics - optimality gap, iteration number, wall-clock running time. You can basically use the code which produced Figure 1 but save and report the running time at the iteration checkpoints you marked in Figure 1.

---

> > ### Author Response · Authors · 2024-08-11
> > **Compare with MOSEK**
> >
> > We appreciate your insight and understanding of the difference between first-order and second-order methods. Upon your request, we conducted more comparisons with MOSEK. Below, we have recorded MOSEK's solving time for the tested datasets in the following table. We observe that MOSEK performs well with these medium-scale datasets, which have a dimensionality of around 2-3 thousand, solving them within seconds.
> >
> > | dataset | apd | apd_restart | apdpro | mirror | msapd | mosek |
> > | --- | --- | --- | --- | --- | --- | --- |
> > | bio-CE-HT | 187.15 (0.86)* | 115.95 (1.04) | 136.92 (0.92) | 370.50 (1.80)* | 77.21 (0.67) | 0.21 |
> > | bio-CE-LC | 2.58 (0.16)* | 0.65 (0.01) | 0.44 (0.01) | 4.74 (0.33)* | 0.65 (0.03) | 0.10 |
> > | econ-beaflw | 72.28 (0.59)* | 87.12 (0.43)* | 18.42 (0.44) | 116.13 (1.15)* | 66.70 (0.76) | 0.16 |
> > | DD242 | 43.29 (1.20)* | 10.27 (0.39) | 6.30 (0.08) | 79.16 (0.60)* | 10.33 (0.62) | 0.16 |
> > | DD68 | 36.55 (0.42)* | 19.07 (0.66) | 22.35 (0.75) | 67.73 (1.39)* | 15.69 (0.37) | 0.24 |
> > | peking-1 | 122.37 (2.99)* | 11.55 (0.69) | 4.86 (0.09) | 243.45 (7.20)* | 11.24 (0.15) | 0.21 |
> >
> > Therefore, according to your suggestion, we test the efficacy of the first-order algorithm on some large-scale instances, which indeed suggests the advantage of our methods against second-order solver. It is also important to note that our algorithm is only implemented in Python. Improved implementation in C or Julia can potentially leads to even more significant speed up.
> > For large-scale instances, we consider the following problem
> >
> > $$
> > \min_{x \in \mathbb{R}^n} \\|x-1\\|_1\ \ \text{s.t.}\ \  0.5 * x^T Q_i x + c_i^T x + d_i \le 0, i = 1, \ldots, m,
> > $$
> > where $Q_i$ are dense and positive definite matrix and generated randomly, and $c_i$ are generated randomly. Furthermore, we set proper $d_i$ to make the feasible region is non-empty. We only compared MOSEK with rAPDPro. When the problem dimension is $n = 5000$ and $m>10$, MOSEK crashes on our computer (Mac mini M2 Pro, 32GB.). However, the first-order algorithm has significantly lower memory dependence compared to these second-order methods, allowing it to continue solving. We report the time required for the algorithm to satisfy 'max{optimality gap, feasibility gap} $\leq 10^{-3}$'. When  $m>10$, we report the time taken by the algorithm to complete 10,000 iterations. On this problem, results from small datasets indicate that the performance of the 10,000-step algorithm should be sufficient to meet our specified termination criteria.
> >
> > | dataset | rapdpro（s） | mosek（s） |
> > | --- | --- | --- |
> > | m = 8 | 24.612 | 50.38 |
> > | m = 10 | 53.997 | 67.99 |
> > | m = 12 | 392 (10000 iteration) | - |
> >
> > You also would like to present various errors measures v.s. wall-clock time in the form of a table. Based on our understanding, you seem to be requesting the output log from MOSEK. Below, we have provided the relevant logs from MOSEK, along with a portion of the output from our methods. Since our output is quite lengthy, we have only included a segment of it. we hope these results meet your expectations.
> >
> > ```
> > log for $m=8$:
> > ITE PFEAS    DFEAS    GFEAS    PRSTATUS   POBJ              DOBJ              MU       TIME
> > 0   1.0e+00  2.0e+01  9.0e+00  0.00e+00   4.992000000e+03   5.000000000e+03   1.0e+00  36.50
> > 1   2.8e-01  5.5e+00  3.7e+00  -8.22e-01  8.369708190e+03   8.374145145e+03   2.8e-01  40.55
> > 2   1.6e-01  3.3e+00  1.5e+00  3.60e-01   7.233137244e+03   7.235917729e+03   1.6e-01  42.45
> > 3   3.9e-02  7.8e-01  1.2e-01  9.17e-01   5.514658049e+03   5.515358832e+03   3.9e-02  44.27
> > 4   9.2e-05  1.8e-03  8.1e-06  1.08e+00   4.942689048e+03   4.942690794e+03   9.2e-05  46.43
> > 5   4.5e-08  9.0e-07  8.8e-11  1.00e+00   4.939417987e+03   4.939417988e+03   4.5e-08  48.60
> > 6   1.2e-13  1.5e-11  2.1e-16  1.00e+00   4.939416383e+03   4.939416383e+03   1.2e-13  50.38
> > ```
> > ```
> > rapdpro log for $m=8$:
> > epoch obj         constrVio   dual_var    t
> > 0     4999.506511 0           2.825598698 0.09
> > 100   4984.100415 0           2.40219241  2.678
> > 200   4980.007982 0           1.975805107 5.113
> > 300   4975.166504 0           1.572708227 7.571
> > 400   4968.367763 0           1.207459909 10.034
> > 500   4958.888395 0           0.905529645 12.526
> > 600   4947.165431 0           0.701566182 14.943
> > 700   4937.057397 0.049015279 0.623223009 17.281
> > 800   4934.033091 0.107150985 0.648725996 19.701
> > 900   4936.837886 0.049825702 0.687484058 22.216
> > 1000  4939.84047  0           0.690012634 24.612
> > 1100  4940.75177  0           0.673186538 26.999
> > 1200  4940.22536  0           0.656075569 29.434
> > ```

---

> > > ### Comment · Reviewer_ecU7 · 2024-08-14
> > >
> > > Thank you very much for your providing the requested experimental results. This is what I expected.
> > >
> > > I am maintaining my score as it is already in the accept status.
> > >
> > > During revision, I strongly recommend you incorporate the results from MOSEK. I do think this makes your work more rounded and comprehensive. I think when you revise the writing, you can highlight your method is more advantageous than the second-order method when dealing with huge-scale datasets. Additionally, it would be great if you can find a real-world dataset corresponding to this huge-scale, not just the synthetic dataset. If you can do this, your work will also become more attractive to optimization practitioners.

---

> > > > ### Author Response · Authors · 2024-08-14
> > > >
> > > > Thank you very much for your insightful suggestions. We fully agree that incorporating the results from MOSEK will enhance the comprehensiveness of our work. In the revision, we will highlight the advantages of our method over the second-order method, particularly in the context of large-scale datasets. Additionally, as you suggested, we will make every effort to identify real-world examples to further demonstrate the efficiency of our algorithm compared to second-order methods.

---

### Official Review · Reviewer_rJ7a · 2024-07-13

**Soundness:** 3
**Presentation:** 3
**Contribution:** 3
**Rating:** 6
**Confidence:** 3

**Summary:**

This paper introduces accelerated primal-dual algorithms for minimizing a convex function subject to strongly convex constraints. Currently, the best complextiy bound for these problems is $\mathcal{O}(1/\epsilon)$, even when the constraints are strongly convex. However, this work develops a technique to progressively estimate the strong convexity of the Lagrangian function, and thereby establishes an improved, and optimal, complexity bound of $\mathcal{O}(1/\sqrt{\epsilon})$. Further, a restarted version of the methods can identify the sparsity pattern of the optimal solution within a finite number of steps.

**Strengths:**

The paper was well written and the problem is of interest to the wider research community.
The paper establishes convergence of their method, and exploits the strong convexity of the constraint functions to obtain an improved complexity of $\mathcal{O}(1/\sqrt{\epsilon})$.
That the restarted version of the algorithm can identify the sparsity pattern of the optimal solution in a finite number of steps, which is independently interesting.

**Weaknesses:**

The claim of "optimal rate" does assume that $\tilde \rho_K$ is small. If it is order epsilon, then the `usual' complexity bound holds (the authors are upfront about this as it is noted in Remark 2). This does weaken the "optimal" claim somewhat.

**Questions:**

1. In Definition 1, does the last condition (orthogonality of the vectors) imply complementarity? Does one not need $\mathbf{y^*}_i g_i(\mathbf{x^*}) = 0$ for all $i$?
2. On line 140, the word "closeness" is used. Do you mean "closedness"?

---

> ### Author Rebuttal · Authors · 2024-08-05
>
> Thanks for the careful reading and valuable feedback! We hope that the following can resolve your concerns and questions.
> Firstly, you are correct that we can not accelerate our algorithm if $\tilde{\rho} _{K}$  is at order $\mathcal{O}(\epsilon)$. Indeed, when $\tilde{\rho} _{K}$ is of the order of $\mathcal{O}(\epsilon)$, as we mentioned in Remark 2, our algorithm reverts to handling convex objective functions and convex constraints. In this case, our algorithm cannot fully utilize the strong convexity of the constraints and can obtain rate $\mathcal{O}(1/\varepsilon)$.
> Secondly, we are sorry for the confusing notations in Definition 1. The last condition indeed implies complementarity. Following Reviewer 8Vg8's suggestion, we are preparing to modify Definition 1 equivalently as follows: **Definition 1(KKT condition).** We say that $x^*$ satisfies the KKT condition of (1) if there exists a Lagrangian multiplier vector $y^{*} \in \mathbb{R} _+ ^m$ such that
>
> $$0 \in \partial _{x} L(x^*,y^*),G(x^*) \leq 0,\langle \mathbf{y} ^*,G(\mathbf{x} ^*) \rangle = 0.$$
>
> These conditions can actually imply $\\mathbf{y}^*_{(i)} g _i(\mathbf{x}^*)=0, \forall i$. We completely agree with your observation.
> Finally, you are right, it should be "closedness".

---

### Official Review · Reviewer_8Vg8 · 2024-07-19

**Soundness:** 2
**Presentation:** 1
**Contribution:** 3
**Rating:** 6
**Confidence:** 4

**Summary:**

Overall, this paper introduces a new idea and new result which is that we can accelerate constraint optimization as soon as the constraint sets are strongly convex. This result is very interesting to the community, but the paper’s writing is really bad as it is and needs a huge improvement to be publishable in my opinion.

**Strengths:**

This paper addresses an important problem, constrained optimization, and shows that we can achieve a Nesterov-like acceleration over such problems.

**Weaknesses:**

Here I list all the remarks that I would like to address, going from typo to concern. They are not all of equal importance, I list them conserving the order of the paper:

- l.87: Can the authors be more precise on which upper bound it is? Is this some known upper bound that we need to use in the algorithm? Or $||y_0 - y_*||$ would work? This is not clear from here.

- l.125: Authors should specify the derivative. Which variables are we deriving against?
- l.125: $y\geq 0$ is redundant with $y\in\mathbb{R}_+^m$.
- l.125: I suggest to the authors to avoid such a sequence of different types of relations and instead write "$G(x^*)\leq 0$ and $<y^*, G(x^*)> = 0$".

- l.131: « easily verifiable ». Can the authors explicitly say how? I understand in the particular case they mention right after, but in general, it would require finding all minimizers of $f$ and check. But no practical algorithm is guaranteed to provide an exact solution to unconstraint minimization of $f$, and even less to provide all of them.
- My personal point of view on Assumption 2 is that, while Problem 1 would be equivalent to an unconstrained minimization problem if this assumption is not verified, we would like to have an algorithm that covers both cases to avoid the need for determining in which case we are. Except if, as authors pretend, Assumption 2 is easily verifiable. It is fair to make an assumption if the most general case if hard to solve, but I think the authors should not underestimate the importance of this assumption in their text.

- Section D1 eq 19: Authors should not use $\Longrightarrow$ instead of « then » or « thus » or « it follows ». $A \Longrightarrow B$ has a very precise signification. It means that if A, then B, but without knowing if A is verified. Which is different from stating A as a true statement and concluding with B.
- Section D1 eq 19: The last $\leq \bar{c}$ should be replaced by an equality by definition of $\bar{c}$.

- l.141: Is the closeness of the subdifferential set of proper convex functions a result originally dated back to 2017? Otherwise, I think a more relevant reference should be preferred. This applied to other references in this paper. Authors should prefer original ones except a more recent one brings some very clear and new explanation to the claim.

- Authors should add a sentence after the different statements they make with a clickable link to the place where we can find the proof in their paper/appendix.
- l.141: « we derive a subdifferential separation result ». Where is it? From assumption 2, I understand that $d(\partial f(x^*), 0) > 0$. So if $r$ depends on $f$, the statement is trivial, if it is uniform over all the problems, then it needs proof.

- After stating Assumption 1, the authors should define $\tilde{x}$ as a generic notation for the existing strictly feasible point so that it can be used later. Otherwise, the $\tilde{x}$ is not defined in Propositions 1 and 3.

- In general, I suggest the authors motivate their lemmas/propositions/theorems before stating them (e.g. Proposition 3). Otherwise, a reader does not know where all this is going.

- Proposition 3: Why add a $\zeta$? Since $\tilde{x}$ is in the interior of $\mathcal{X}_G$, the first inequality replacing $x_1$ and $x_2$ by respectively $\tilde{x}$ and $x^*$ should be strict, and therefore $\zeta$ can be set to 0.

- eq.7: At this stage, the authors explain that we can straightforwardly prove some bound but we do not know why it is useful, nor what it means. More motivation and discussions are needed to improve the writing of this paper.

- l.164: For better readability, I suggest the author use classical tools. What they define as $prox_{f, \mathcal{X}}(x, z, \eta)$ is simply the prox of $f$ at a different point: $prox_{f, \mathcal{X}}(x - \eta z, \eta)$. Otherwise one can think that the authors assume access to many different oracles, the proximal operators of many different functions.

- l.178: $N_\mathcal{X} \rightarrow \mathcal{N}_\mathcal{X}$


Algo 1:
- l.4: « Compute $\theta_k$ ». How? Eq.11?
- $\theta_k$ is only used in l.5 which defines $z_k$, only used in l.6. I suggest authors merge those lines into a single one.
- l.8: 3 updates in one line. Authors should at least use « \qquad » to separate them and increase readability.
- l.10: Update how? In any way as soon as Thm1 assumptions are verified? $\gamma$ is not defined in Thm1. Should we look at Cor 1? It is then defined as $\sigma / \tau$. Why do we update it after updating $\sigma$ and $\tau$? It is not even called in Algo 1.

Thm1:
- Why assume that $t_{k+1} / \sigma_{k+1} \leq t_{k} / \sigma_{k}$ (second inequality of eq.11) while in Algo 1, $t_k = \sigma_k / \sigma_0$? The assumed inequality is necessarily true and even an equality.
- Concerning the third assumption in eq.11, we discussed it above: the expression of $\theta$ should be replaced in the algorithm for better readability.
- Replacing $\theta_k$ by its expression, one sees that the second inequality of eq.12 simply reads $\delta \leq 1$.
In summary, the second and third assumptions of eq.11 and the second assumption of eq.12 should be removed and replaced by the sole assumption $\delta\leq 1$. Moreover, since $\delta$ is used nowhere in the algorithm, nor in the consequence eq.13 of Theorem 1, but only mentioned in the first inequality of eq.12, we can replace it with its largest possible value, i.e. 1.
Finally, in conclusion, authors should replace eq.11 and 12 by
$\tau_k(1/\tau_{k+1} -  \rho_{k+1}) \leq \sigma_k / \sigma_{k+1}$ and $L_{XY} + L_G^2\sigma_k \leq 1/\tau_k$, making the statement much clearer.

Cor1:
- Again, $\delta$ seems to be used only in the first assumption and neither the algorithm nor the guarantee depends on the choice of $\delta$, so why not just fix it to the most permissive value, i.e. 1?
- $\theta$ is defined for a single used, so why not avoiding a new definition?
- $t_k$ is finally fixed as in the algorithm. If the authors want to be more permissive in the theorem, $t_k$ must not be fixed in the algorithm.
- A new notation arrived: $\gamma$. Why? This is not used in the algorithm.
- Moreover, all those assumptions can be simplified a lot. First, the last equation of the second line simply writes $\tau_k^2\gamma_k$ is constant. Removing $\gamma_k$ and writing all this in terms of used quantities, we have that $\sigma_k\tau_k$ is. constant. In conclusion, the algorithm is parametrized by 3 sequences:  $t$, $\tau$ and $\sigma$, with $t\propto\sigma\propto 1/\tau $,  which can be greatly simplified using only 1 sequence.
- The two only useful assumptions of theorem 1 now write (with this new proportionality assumption of Cor.1, introducing $c_0 = \tau_k\sigma_k$):
    - $\sigma_k / c_0 \geq L_{XY} + L_G^2\sigma_k$ and using that. $\sigma_k$ is increasing as pointed out in Remark 3, it suffices to verify this inequality for $k=0$, explaining the first inequality following l.194 (to be taken for $\delta=1$): $c_0 \leq \sigma_0 / (L_{XY} + L_G^2\sigma_0)$.
    - $c_0 \rho_k \geq (\sigma_k^2 - \sigma_{k-1}^2) / \sigma_k$. Note that it shows $c_0$ should be taken as large as possible and then the previous bound on $c_0$ has to be taken as an equality. Finally, $\rho_k$ is upper bounded, so is $(\sigma_k^2 - \sigma_{k-1}^2) / \sigma_k$, showing that $\sigma$ can only increase up to a certain speed, similar to the classical parameters of NAG algorithm. This clearly explains the fact that $t_k$ (as well as $\sigma_k$ and $1/\tau_k$) grows as $k$ and $T_k$ as $k^2$, giving eq.14. Remark 3 explains why we obtain at least a $1/k$ guarantee, but it not more complicated to see that we can actually accelerate. In my view, the algorithm, the thm, and the corollary have all been overcomplicated with many notations and assumptions and a lack of explanation. and intuition that would have made all this straightforward to understand.


In summary, I would say that the method is sound as soon as we have some $\mu_{min}$ (main assumption of this paper) and some $r$ (eq.4) coming from the assumption that no minimizer of $f$ can be a solution to our problem. Note we also need to have access to such a $r$, which does not seem trivial in general.


Overall, this paper introduces a new idea and new result which is that we can accelerate constraint optimization as soon as the constraint sets are strongly convex. This result is very interesting to the community, but the paper’s writing is really bad as it is and needs a huge improvement to be publishable in my opinion.

**Questions:**

- l.7-8: « Our approach, for the first time, effectively leverages the constraint strong convexity, obtaining an improved complexity of $O(1/\sqrt{\varepsilon})$ ».
- l.27-28: « When the objective is strongly convex, the complexity can be further improved to $O(1/\sqrt{\varepsilon})$ [cite refs]».
- l.42-43: « Specifically, direct applications of previously discussed algorithms yield an $O(1/\varepsilon)$ complexity ».

Is this acceleration result novel?
While I understand that the authors mention a 2-loops vs single loop procedure, this must be clear from the beginning. I find the abstract a bit overselling if there actually is some method, even using 2 loops, that achieves this accelerated rate. I suggest the authors be more specific in their abstract if this is the case.

**Limitations:**

No limitations other than the claimed assumptions.

---

> ### Author Rebuttal · Authors · 2024-08-05
>
> We sincerely appreciate your feedback and are grateful for the thorough reading and valuable suggestions for our paper. Due to space limits, we respond to the main technical questions.
>
> **l87:** We establish two upper bounds for two error measures. Since new point $y^+$ is needed for optimality and feasibility gap (see Cor1), while this is not needed for $\\|x_k - x^*\\| \le \varepsilon$. We clarify that our algorithms do not require the $\\|x-x^*\\|$ and $\\|y-y^*\\|$. Instead, it relies on $D_X$ and $D_Y$ to replace $\beta$ in Prop4 for estimating $\mu_{min}$.
>
> **l125:** We plan to revise KKT condition as follows: We say $x^*$ satisfies the KKT condition of (1) if there exists a Lagrangian multiplier vector $y^* \in \mathbb{R}_{+}^{m}$ such that $0\in \partial _{x} L(x^*,y^*)$, $G(x^*)\le 0$ and $\langle y^*,G(x^*) \rangle = 0$.
>
> **Is Assu 2 easily verifiable?:** We agree that the term may not be entirely appropriate. Assu 2 would require finding all minimizers of f and check, and solving this issue would be challenging. This indeed motivates the l1 loss, where Assu 2 can be verified.
>
> **Can an algorithm cover both cases?** While it is challenging in general, we feel verifying Assu 2 is possible when projection on the level set of $f(x)$ is easy. First, we compute an $\epsilon$-solution $\hat{x}$ of $\min f(x)$ efficiently, (e.g. in $O(1/\sqrt{\epsilon})$ using Nesterov's acceleration). We switch the objective and constraint and consider the following problem: $\tau^*=\min _x \max _i g_i(x), \text{ s.t. } f(x)\le f(\hat{x}).$ When the projection is easy to solve, we can find an approximate solution ($\bar{x}$) and value $\bar{\tau}$ satisfying $\bar{\tau}\le \tau^*+\epsilon$ in $O(1/\sqrt{\epsilon})$ by using accelerated gradient method and smoothing technique. If $\bar{\tau}>\epsilon$, then we have $\tau^*>0$ and hence Assu 2 holds. Otherwise, we have $\tau^*\le \bar{\tau}\le \epsilon$, then the solution $\bar{x}$ naturally becomes an $\epsilon$-solution of the original problem. We hope this can partially resolve the reviewer's concern.
>
> **closedness** Upon reviewing the literature, we found in [1], Chapter 23, (2nd paragraph, page 215), the statement: "Obviously $\partial f(x)$ is a closed convex set, since ..." However, this is not a formal theorem. This book is one of the earlier sources on "The closedness of the subdifferential set," which we intend to cite.
>
> **L141 derive:** We apologize the statement here is unclear. As you say, it is impossible to determine r for general function. We aim to modify it as follows. In view of Assu 2, Prop 2, we know that $dist(\partial f(x^*))>0$. Furthermore, we make the following assumption: Throughout the paper, a lower bound $r\in (0,dist(\partial f(x^*))]$ is known. We give some important examples for which the lower bound r can be estimated.
>
> **More motivations:** Previously, due to space limitations, we omitted much of the preparatory material. In the revision, we will detail the motivation. Regarding Prop 3, we will add introductory statements: "When considering the Lipschitz continuity of functions in $\mathbb{R}^n$, even quadratic functions are not Lipschitz continuous. However, the Lipschitz continuity of $g_i(x)$ is crucial for algorithm convergence. Therefore, we define the bounded feasible region in the following proposition." We hope this will improve the understanding of Prop 3.
>
> **$\zeta$ Prop 3:** We deeply appreciate your rigorous analysis of the inequalities: $\\|x^*-x_i^*\\|^2\le \frac{-2g_i(x_i^*)}{\mu_i}$ and $\\|\tilde{x}-x_i^*\\|^2 < \frac{-2g_i(x_i^*)}{\mu_i}$. We will revise the definition of $\mathcal{X}$.
>
> **Motivation for (7):** $L_{XY}$ is necessary for step size setting in algorithms. We will add the following statement for readability: "The Lipschitz smoothness of the Lagrangian function with respect to the primal variable x is crucial for the convergence of algorithms. Given that the dual variable y is bounded from above, and considering the smoothness of the constraint functions, we can derive the smoothness of the Lagrangian function. Combining (5) and ...".
>
> **All problems in Alg1, Thm1 and Cor1.** We apologize for any lack of conciseness and readability. Our goal was to show the flexibility of algorithm parameters. Thm1 shows that the algorithm can converge if the parameters meet certain conditions, and Cor1 specifies these parameters. Introducing many auxiliary sequences reduced readability and did not aid calculations. You suggested that our pseudocode should clearly present the calculation methods for parameters. Using only one sequence, $\sigma_k$, and setting $\delta$ to 1 improves readability. Your understanding of our algorithm is commendable. However, to simplify subsequent proofs, we will keep the two step size sequences, $\tau_k$ and $\sigma_k$. We have included the pseudocode of our APDPro in the rebuttal PDF. As you suggested, we revise equation (12) to  $\tau_k(\tau_{k+1}^{-1}-\rho_{k+1})\le\sigma_k/\sigma_{k+1}, L_{XY}+L_G^2\sigma_k\le1/\tau_k$ and the precondition in Cor1 to $1/\tau_0\ge L_{XY}+L_G^2\sigma_0$. Furthermore, after eliminating the auxiliary sequence $\gamma_k$, Lemma 3 requires corresponding modifications. Due to the space limitation, please review our comments for detailed explanations of our revisions.
>
> **2 loop vs single loop:** The reviewer raises an interesting question of whether a method, even one using 2 loops, has achieved or can potentially reach this accelerated rate. While we have not observed any methods aiming to leverage the strong convexity of constraint functions, we believe that further improvement is quite promising based on our technique. Typically, our technique can benefit a wide range of algorithms using Lagrangian multipliers, such as IALM ([32] in paper), which involve 1st-order methods to solve their subproblems.
>
> [1] R Tyrrell Rockafellar. Convex analysis, volume 18. Princeton university press, 1970.

---

> ### Author Response · Authors · 2024-08-05
> **Revisions of Theorem 1, Corollary 1 and Lemma 3**
>
> **Theorem 1.** Suppose for any $x^*\in \mathcal{Y}^*$, $(x^*)^\top \boldsymbol{\mu}\geq \rho_0$ holds, and there exist sequences $\{\tau_k,\sigma_k\}$ satisfies
> $$\tau_k(\tau_{k+1}^{-1}-\rho_{k+1})\leq \sigma_k/\sigma_{k+1}, L_{XY}+L_G^2\sigma_k\leq \tau_k^{-1}.$$
> Then, the set $\mathcal{Y} _k$ is nonempty and $\mathcal{Y}^*\subseteq \mathcal{Y}_k$.  Let $\Delta(x, y):=\frac{1}{2\tau_0}\\|x-x_0\\|^2+\frac{1}{2\sigma_0}\\|y-y_0\\|^2$,$\bar{y}_K=T_K^{-1}\sum _{s=0} ^{K-1}t_s y_s$. The sequence $\{\bar{x}_k,x_k,\bar{y}_k\}$ generated by APDPro satisfies
>
> $$\frac{t_{K-1}\tau _{K-1} ^{-1}}{2T _{K}}\\|x^*-x_K\\|^{2}+L(\bar{x}_K,y^*)-L(x^{*},\bar{y}_K)\leq\Delta(x^ *,  y^ *)/ T _K.$$
>
> **Corollary 1.** Suppose that $\sigma_k,\tau_k$ satisfy:
> $\tau_0 ^{-1}\geq L_{XY}+L_G ^2\sigma_0$, then we have
>
> $$f(\bar{x}_{K})-f(x^*)\leq  \frac{6}{6+\tau _{0} \tilde{\rho} _{K} (K+1)K} (\frac{1}{2\tau_0}\\|x_0-x^*\\|^2+\frac{D_Y^2}{2\sigma_0} ),
> $$
>
> $$\\|[G(\bar{x} _{K})] _{+}\\| \leq \frac{6}{c ^*(6+\tau _{0}\tilde{\rho} _{K} (K+1)K)}(\frac{1}{2\tau_0}\\|x_0-x^*\|^2+\frac{D_Y^2}{2\sigma_0}),$$
>
> $$\frac{1}{2}\\|x _{K}-x ^*\\|^{2} \leq \frac{3 \sigma _{0}}{\tilde{\rho} _{K} ^{2} \tau _{0} ^{2}K ^{2}+9\gamma _{0}}\Delta(x ^*,y ^*),$$
>
> where $c^*:=(f(x^*)-\min_{x}f(x))/{\min_{i\in[m]}\{-g_{i}(\tilde{x})\}}>0$, $\tilde{\rho} _k = 2\sum _{s=0} ^{k}\hat{\rho} _s s/({k(k+1)})$ and $\hat{\rho} _k$ satisfy the following condition, $\hat{\rho} _{k+1}:= \sqrt{\hat{\rho} _k ^2 k^2 + (3 \rho _{k+1} \hat{\rho} _k)k}/(k+1), \hat{\rho} _1 = 3\sqrt{\rho _1/\tau _0}$.
>
> **Lemma 3.** Let $\hat{\rho} _{k+1}:=\frac{\sqrt{\hat{\rho} _{k}^{2}k^{2}+(3\rho _{k+1}\hat{\rho} _{k})k}}{k+1}$
> for $k\geq1$ and $\hat{\rho} _{1}=3\sqrt{\frac{\rho _{1}}{\tau _{0}}}$.
> Suppose $\sigma _{k},\tau _{k}$ satisfy:
>
> $$\tau_{0}^{-1}\geq L_{XY}+L_{G}^{2}\sigma_{0},\ \ \tau_{k+1}=\tau_{k}(1+\rho_{k+1}\tau_{k})^{-\frac{1}{2}},\ \ \sigma_{k+1}=\frac{\tau_{k}\sigma_{k}}{\tau_{k+1}}.$$
>
> Then we have
>
> $$\frac{1}{\tau_{k} ^{2}}\geq\frac{\hat{\rho} _{k}^{2}}{9}k^{2}+\frac{1}{\tau _{0}^{2}},T _{k}\geq 1+\frac{\tau _{0}}{6}\tilde{\rho} _{k}(k+1)k,\ \ \hat{\rho} _{k}\geq\min\\{\rho _{1},\hat{\rho} _{1}\\},$$
>
> where $\tilde{\rho} _{k}=2\sum _{s=0} ^{k}\frac{\hat{\rho} _{s}s}{k(k+1)}$
> for $k\geq1$. Moreover, suppose $\bar{\rho}\tau _{0}\leq2$, where
> $\bar{\rho}=\bar{c}\cdot\bar{\mu}$, then we have $\sigma _{k}^{2}\leq\sigma _{0}^{2}(k+1)^{2}.$
>
> **Proof.** We first use induction to show that $\frac{1}{\tau _{k}^{2}}\geq\frac{\hat{\rho} _{k}^{2}}{9}k^{2}+\frac{1}{\tau _{0}^{2}}$.
> It is easy to see that $\frac{1}{\tau _{k}^{2}}\geq\frac{\hat{\rho} _{k}^{2}}{9}k^{2}+\frac{1}{\tau _{0}^{2}}$
> holds for $k=1$ by the definition $\hat{\rho} _{1}=3\sqrt{\rho _{1}/\tau _{0}}$
> and $\tau _{1}=\tau _{0}(1+\rho _{1}\tau _{0})^{-\frac{1}{2}}$. Assume
> $\frac{1}{\tau _{k}^{2}}\geq\frac{\hat{\rho} _{k}^{2}}{9}k^{2}+\frac{1}{\tau _{0}^{2}}$
> holds for all $k=0,\ldots,K$, then we have
>
> $$\frac{1}{\tau _{K+1}^{2}} =\frac{1}{\tau _{K}^{2}}+\frac{\rho _{K+1}}{\tau _{K}}
>   \geq\frac{\hat{\rho} _{K}^{2}}{9}K^{2}+\frac{1}{\tau _{0}^{2}}+\rho _{K+1}\sqrt{\frac{\hat{\rho} _{K}^{2}}{9}K^{2}+\frac{1}{\tau _{0}^{2}}}
>   \geq\frac{\hat{\rho} _{K}^{2}}{9}K^{2}+\frac{1}{\tau _{0}^{2}}+\frac{\rho _{K+1}\hat{\rho} _{K}K}{3}
>   \geq\frac{\hat{\rho} _{K+1}^{2}}{9}(K+1)^{2}+\frac{1}{\tau _{0}^{2}},$$
>
> which completes our induction. It follows from $\frac{1}{\tau _{k}^{2}}\geq\frac{\hat{\rho} _{k}^{2}}{9}k^{2}+\frac{1}{\tau _{0}^{2}}$
> and the relation among $T _{k},t _{k},\sigma _{k},\tau _{k}$ that, for
> any $k\geq1$
> $$T _{k}=\sum _{s=0}^{k-1}t _{s}=1+\sum _{s=1}^{k-1}t _{s}\geq1+\sum _{s=1}^{k-1}\frac{\sigma _{s}}{\sigma _{0}}=1+\sum _{s=1}^{k-1}\frac{\tau _{0}}{\tau _{s}}\geq1+\tau _{0}\sum _{s=1}^{k-1}\sqrt{\frac{\hat{\rho} _{s}^{2}s^{2}}{9}+\frac{1}{\tau _{0}^{2}}}>1+\tau _{0}\sum _{s=1}^{k-1}\frac{\hat{\rho} _{s}s}{3}=1+\frac{\tau _{0}}{6}\tilde{\rho} _{k}(k+1)k.$$
>
> Similarly, we use induction to prove
>
> $$\hat{\rho} _{k}\geq\min\{\rho _{1},\hat{\rho} _{1}\},\forall k\geq1.$$
>
> It is easy to find that $\hat{\rho} _{1}\geq\min\{\rho _{1},\hat{\rho} _{1}\}$ (the same to paper).
>  Moreover, we use induction to show
> $\sigma _{k}^{2}\leq\sigma _{0}^{2}(k+1)^{2}$. It is obvious that the
> inequality holds for $k=0$. Assume the inequality holds for all $k=0,\ldots,K,$
> then we have
> $$\sigma _{K+1}^{2}  =\sigma _{K}^{2}(1+\rho _{K+1}\frac{\tau _{0}\sigma _{0}}{\sigma _{K}})
>   =\sigma _{K}^{2}+\rho _{K+1}\tau _{0}\sigma _{0}\sigma _{K}
>   \leq\sigma _{0}^{2}\left((K+1)^{2}+\rho _{K+1}\tau _{0}(K+1)\right)
>   \leq\sigma _{0}^{2}(K+2)^{2},
> $$
> where the last inequality use the relation $\rho _{k}\leq\bar{\rho},\forall k$,
> and $\bar{\rho}\tau _{0}\leq2$.

---

> ### Author Response · Authors · 2024-08-12
> **Looking forward to your response**
>
> Dear Reviewer 8Vg8,
>
> We are deeply grateful for your insightful comments and the valuable feedback you have provided. We have carefully addressed your concerns and would greatly appreciate hearing from you if you have any further questions or need clarification. As the rebuttal deadline is approaching, we remain at your disposal and are ready to promptly address any additional concerns you may have.
>
> Thank you once again for your invaluable input and consideration.

---

> > ### Comment · Reviewer_8Vg8 · 2024-08-12
> > **Thank you for the detailed rebuttal**
> >
> > First of all, I thank the authors for their efforts to address most of my concerns.
> > While the Neurips review system does not allow the submission of a new version of the paper, the authors expressed the changes they wanted to make and I appreciate it. Bellow, I answer their rebuttal with a few remaining concerns.
> >
> >
> >
> > l.87: My point was that when you say « where $D_y$ is an upper bound of … », it might be understood in two different ways: « where $D_y$ is a quantity we define later in … and that has the property to upper bound … » and « where $D_y$ is any known upper bound of … ». I think you probably meant the second option, but this should be cleared in the text in line 87.
> >
> > l.125: great.
> >
> > Assumption 2: I agree about the particular case of the l1 norm. My point is that saying « easily verifiable » in the text minimizes the importance of the assumption. Here you made an assumption, which is perfectly fine, but you need to be clear: « We present assumption 2, which is essential for our analysis. ». You may eventually add « In some cases, we can verify it beforehand as in the example ….[of the l1 loss] ».
> >
> > Can an algorithm cover both cases?
> > - Your paper tackles a general convex problem with strongly convex constraints. Assumption 2 is the point here. Here you mention how we can overcome its need by assuming that we can easily project onto the level set of $f$ which also is a huge assumption. One replaces another, but my question was « keeping your setting, not having to verify the assumption 2, is it possible to make your idea work ».
> > - « $O(1/\sqrt{\epsilon})$ by using accelerated gradient method and smoothing technique ». Can you be more specific on the type of smoothing technique you use here to achieve the accelerated rate under constraint, assuming access to the projection operator?
> >
> > l.141 derive: Great! I think the Assumption 2 should be introduced the same way. « We make the following assumption … and provide cases where it can be verified ». But please avoid « 31 Assumption 2 is indeed a mild condition and easily verifiable ».
> >
> > More motivations, $\zeta$ Prop 3, and Motivation for (7): Great!
> >
> > All problems in Alg1, Thm1, and Cor1: Actually, I am not eager to see fewer variables. I even think you need them all. Let me explain. Thm 1 is very general w.r.t. the values of the parameters. It provides conditions for the latter. Then Cor 1 fixes the values to optimize the algorithm. Which is great. My point was that in Alg1, many parameters are also fixed. So I pointed out some inconsistencies in the presentation of all that. If they are fixed in the algorithm, you should not have a thm stating that « if they verify something that they clearly verify in your algo, then … ». So, in order to give sense to thm 1, all the free parameters (or updates of the parameters) should be inputs of your algorithm. Then Thm 1 states conditions on the inputs. And Cor1 states what are the best inputs. Am I clear enough?
> >
> > 2 loop vs single loop: I do not understand the response of the authors here. In lines 25-26 of the paper, it is said: « When both $f(x)$ and $g_i(x)$ are convex and smooth (or composite), it has been found 26 that these double-loop algorithms can attain an iteration complexity of $O(1/\varepsilon)$ ». So this rate seems to be known in an even more general setting where the constraints are not strongly convex. Am I misunderstanding something?
> >
> >
> > Proposed rewriting of Alg1/thm1/cor1: This is more readable as is, but I still have 2 concerns:
> > - In thm1, first line: instead of « , and there exist sequences … verifying», please prefer « . Let … 2 sequences verifying …». This way, not only do they exist, but also the notations stick to those sequences.
> > - tau and sigma. are still defined in algo1. They should be inputs of a generic algorithm algo0. Then Thm1 states the conditions they must verify. And finally, Cor1 proposes one specific value. And the well-tuned algorithm 1 is a particular case of the general Algo 0. Otherwise, this does not give sense to thm 1.
> > - In algo1 as it is: « $\tau_{k+1} = \tau_k \times \sqrt …$ ». Shouldn’t it be « $\tau_{k+1} = \tau_k / \sqrt …$ » instead?

---

> > > ### Author Response · Authors · 2024-08-13
> > >
> > > **Alg1, Thm1, and Cor1:** Thank you very much for your suggestions. In the original paper, we indeed fixed certain sequences, such as $t_k$ in Alg1, but in Thm1, we assumed that these sequences satisfy certain conditions, which may have led to some inconsistency in the description. Specifically, the sequence $t_k$ is fixed in Alg1 but free in Thm1, and $\theta_k$ is variable in Alg1 and fixed in Thm1. To summarize your proposal, you suggest modifying the algorithm as follows:
> > >
> > > - Input $\{x_0,y_0,\sigma_0>0,\tau_0>0,\rho_0\ge 0, N>0\}$
> > > - Initialize: $(x _{-1},y _{-1}) \leftarrow(x _0,y _0),\bar{x} _0 \leftarrow x _0,t _{-1} \leftarrow t _{0}$
> > > - delete $\theta _k$ and change the corresponding symbol as $t _{k-1}/t _k$
> > > - modify the line 8 of Alg1 to 'Compute $t _k,\ \ \bar{x} _{k+1}\leftarrow (T _k\bar{x} _k + t _k x _{k+1})/(T _k + t _k),\ \ T _{k+1}\leftarrow T _k+t _k$'
> > > - delete sequence $\gamma _k$, and modify line 10 as ‘update $\tau _{k+1}$ and $\sigma _{k+1}$ depending on $\rho _{k+1}$’
> > >
> > > In the revised version of Alg1, the sequence $\sigma_k,\tau_k$, and $t_k$ are not fixed. Furthermore, we remove the sequence $\theta_k$ and $\gamma_k$ for readability. And the iterative condition among them in Thm1 can be simplified by deleting $\theta_k$ and letting $\delta=1$.  Finally, we then provide a detailed calculation method for these parameters in Cor1.
> > >
> > > **Rewriting of Alg1/thm1/cor1:**
> > >
> > > - We will revise Thm1 as follows based on your suggestion.
> > > Thm1: Suppse for any ..., and let the sequence $\{\tau_k,\sigma_k,t_k,\rho_{k+1}\}$ generated by Alg1 satisfy:
> > > $t_{k+1}(\tau_{k+1}^{-1}-\rho_{k+1})\le t_k \tau_k^{-1},t_{k+1}/\sigma_{k+1}\le t_k/\sigma_k,L_{XY}+L_G^2\sigma_k\le 1/\tau_k$.
> > > - We will maintain the general form of the algorithm to ensure the validity of Theorem 1, and the specific parameter settings will be provided in Corollary 1.
> > > - We sincerely apologize for this typo. You are right, it should be $\tau_{k+1}=\tau_{k}/\sqrt{1+\rho_{k+1}\tau_k}$.
> > >
> > >
> > > **A misunderstanding between us.**
> > >
> > > We apologize for any confusion caused by our previous response. The reviewer initially commented, “I find the abstract a bit overselling if there actually is some method, even using 2 loops, that achieves this accelerated rate.” We understood this to mean that you were concerned that double-loop algorithms may have already achieved a convergence rate of  $O(1/\sqrt{\epsilon})$, potentially diminishing the novelty of our results. We would like to clarify that our technique for estimating the strong convexity coefficient can also be applied to certain double-loop algorithms, potentially enhancing their convergence performance.
> > >
> > > In your new reply, we understand your comment as, “the $O(1/\epsilon)$ rate in 2 loop algorithms, where constraints are not strongly convex,  seems to be in a more general setting than that of our paper, as f(x) and g(x) can be both convex and smooth (or composite)”.
> > >
> > > We want to make a correction to line 25-26, a more appropriate statement should be “When  f(x) is convex and smooth (or composite), and g(x) is convex and smooth, it has been found that these double-loop algorithms can attain an iteration complexity of $O(1/\epsilon)$ ”
> > >
> > > Typically, in double-loop algorithms, to obtain the $O(1/\epsilon)$ rate, a smoothness assumption on the constraint function is required. Without this, the penalty problem, which involves a composition of a penalty function and g(x), becomes non-differentiable and challenging to analyze.
> > >
> > > From a technical level, the key requirement to our analysis is the lower bound $r$. However, for general $f(x)$, computing such an $r$ may be difficult. Therefore, we have simplified our assumption by only considering proximal-friendly objectives.

---

> > > > ### Comment · Reviewer_8Vg8 · 2024-08-13
> > > > **Thank you for your reply**
> > > >
> > > > I thank the authors for their reply.
> > > > I think those modifications will improve the readability of this paper.
> > > > On my side, I will increase my score.
> > > >
> > > > One last remark in their answer about "$O(1/\sqrt{\epsilon})$ by using some accelerated and smoothing technique?":
> > > >
> > > > The authors describe how to minimize max(g_i) with acceleration methods using smoothing, and I agree with all they said. But the previous answer was about minimizing it under the constraint to stay in a certain sublevel set of $f$. Even if we assume the knowledge of the projection operator on the latter, I do not see how we can reach an acceleration while projecting.

---

> ### Author Response · Authors · 2024-08-13
> **Thank you for the detailed comments**
>
> **l87** As you mentioned, it is crucial to specify a known $D_y$. Here, we need to add further details to avoid any misunderstanding that the existence of $D_y$ is sufficient. We later present a method for calculating the value of $D_y$ when the Slater's point is known. If $D_y$ is not known in advance, it is possible to establish an unknown upper bound for the dual variables by choosing an appropriate step size. The existence of this upper bound is discussed in [3]. However, it is important to clarify that your comment is correct: in our setting, a known $D_y$ is necessary to estimate the lower bound of the strong convexity of Lagragian function. $D_y$ is used in the calculation of $\Delta_{XY}$ (line 9 in Alg1). Investigating how to achieve a convergence rate of $O(1/\sqrt{\epsilon})$ when $D_y$ is unknown is an interesting problem and will be the focus of our future work. Finally, we agree with your comment. We will emphasize at l87 that $D_y$ is a known value in revision.
>
> **Assumption 2: « We present assumption 2, which is essential for our analysis. ». You may eventually add « In some cases, we can verify it beforehand as in the example …. ».**
>
> We appreciate your suggestion to emphasize the role of Assumption 2, and we plan to revise it according to your suggestion.
>
> **"keeping your setting, not having to verify the assumption 2, is it possible to make your idea work”**
>
> This is indeed an interesting question. Unfortunately, without verifying assumption 2, it seems unlikely that our approach would be effective. Specifically, we need a positive $r$ to estimate the strong convexity of the Lagrangian function, which is crucial for achieving the accelerated rate. If $r$ is set to zero, then our algorithm will set all the rho to be zero and then it will be reduced to the standard APD algorithm with an $O(1/\epsilon)$ complexity. As a result, our algorithm is best suited for problems where feasibility is a significant challenge and a non-degenerate solution is anticipated.  We acknowledge the reviewer’s suggestion as an open question and plan to explore it in future work.
>
> $O(1/\sqrt{\epsilon})$ **by using some accelerated and smoothing technique?**
>
> To achieve a complexity of  $O(1/\sqrt{\epsilon})$  using accelerated and smoothing techniques, we can proceed as follows:
>
> We can write $\max_{i}\\{g_i(x)\\}$  as the sum of max-type function and quadratic function: $\max_{i}\\{g_i(x)-\frac{\mu_{\min}}{2}\\|x\\|^2\\}+ \frac{\mu_{\min}}{2}\\|x\\|^2$  and smooth out the max operator using the softmax operator. (Example 4.9, [2]).  After smoothing, we can apply an accelerated gradient method to solve the resulting strongly convex smooth problem. By choosing the smoothing parameter properly, we can obtain a complexity of $O(\sqrt{\epsilon}^{-1}\log(1/\epsilon))$, which interplates between the $\log(1/\epsilon)$ rate of smooth strongly convex optimization and $O(1/\epsilon)$ of nonsmooth strong convex optimization. To obtain the tightest possible rate $O(\sqrt{\epsilon}^{-1})$, one can employ an adaptive smoothing and regularization technique as described in [1]. It is worth noting that while [1] can apply to the max of linear functions, [4] extends this approach to handle the max of convex differentiable functions.
>
> [1] Allen-Zhu, Zeyuan, and Elad Hazan. "Optimal black-box reductions between optimization objectives." *Advances in Neural Information Processing Systems* 29 (2016).
>
> [2] Beck, Amir, and Marc Teboulle. "Smoothing and first order methods: A unified framework." *SIAM Journal on Optimization*22, no. 2 (2012): 557-580.
>
> [3] Erfan Yazdandoost Hamedani and Necdet Serhat Aybat. A primal-dual algorithm with line
> search for general convex-concave saddle point problems. SIAM Journal on Optimization,
> 31(2):1299–1329, 2021.
>
> [4] Lin, Qihang, Selvaprabu Nadarajah, and Negar Soheili. "A level-set method for convex optimization with a feasible solution path." *SIAM Journal on Optimization* 28, no. 4 (2018): 3290-3311.

---

> ### Author Response · Authors · 2024-08-14
> **Thank you for your comments**
>
> [1] demonstrate that combining the adaptive smoothing technique with accelerated gradient methods, such as Katyusha [5], can achieve a convergence rate of $O(1/\sqrt{\epsilon})$.  This result remains valid if the proximal operator is easy to compute, allowing the regularization term to be set as the indicator function $\psi(x) = I_{\mathcal{X}}(x)$. Thus, by combining adaptive smoothing with accelerated proximal gradient methods and setting $\psi(x) = I_{\mathcal{X}}(x)$ for cases where projection onto the constraint set is straightforward, the $O(1/\sqrt{\epsilon})$ rate can still be achieved. Additionally, [4] propose a new method for the case $x\in \mathcal{X}$, and if all $g_i$ are smooth, it should achieve the first conclusion of Theorem 4 in [4], which also results in a convergence rate of $O(1/\sqrt{\epsilon})$. The specific method for achieving this rate is described in Oracle 1, with the parameters detailed in Table 1 of [4].
>
> [5] Allen-Zhu, Zeyuan. "Katyusha: The first direct acceleration of stochastic gradient methods." Journal of Machine Learning Research 18, no. 221 (2018): 1-51.

---

> > ### Comment · Reviewer_8Vg8 · 2024-08-14
> >
> > I see, thank you for the answer.

---

### Author Rebuttal · Authors · 2024-08-05

We sincerely thank the PC, SAC, AC, and all the reviewers, especially the four reviewers. Their feedback has been invaluable, and we will carefully revise our manuscript to meet their standards. We have responded to each comment in the author rebuttal, aiming to resolve their concerns. Due to space limitations and the inability to attach figures in the author rebuttal, we will provide additional explanations for two reviewers' comments.

Reviewer 8Vg8 provided many suggestions on our writing. We have revised the manuscript according to the suggestions to enhance readability. Based on the feedback, the latest version of APDPro in the PDF clearly outlines the specific parameter settings.

Reviewer ecU7 requested a comparison between our algorithms and APD+restart, particularly regarding wall-clock time. Using the condition that both the optimality gap and feasibility gap are less than $10 ^{-3}$ as the stopping criterion, we recorded the required wall-clock time. Our algorithms (rAPDPro and msAPD) are generally faster and more stable compared to other algorithms, including APD+restart.

---

### Decision · Program_Chairs · 2024-09-25

**Decision:**

Accept (poster)

**Comment:**

The authors study the problem of convex optimization with strongly convex constraints. They present new primal-dual first-order methods that improve the complexity bound from $O(1/\epsilon)$ to $O(1/\sqrt{\epsilon})$. The reviewers are in agreement that the paper would be of interest to the community and should therefore be accepted, though please be sure to incorporate the suggested feedback to improve the presentation of the paper for the camera-ready version.